# EFFICIENTLLM:
# EVALUATING LARGE LANGUAGE MODELS EFFICIENCY

### EVALUATION ON ARCHITECTURE PRETRAINING, FINE-TUNING, AND BIT-WIDTH QUANTIZATION

## ABSTRACT

Large Language Models (LLMs) have achieved remarkable advances across reasoning, generation, and problem-solving, yet their scaling comes with prohibitive training, deployment, and environmental costs. Training frontier models like GPT-3 or PaLM consumes thousands of GPU/TPU days and millions of dollars. As these costs escalate, there is a pressing need for rigorous benchmarks that quantify efficiency–performance trade-offs. However, existing evaluations remain inadequate: 1) they rely on narrow metrics such as FLOPs or latency, neglecting complementary dimensions like memory, throughput, energy, and compression, leading to mischaracterized efficiency; 2) they are often limited to small models or a single hardware setup, making conclusions difficult to generalize to billion-parameter deployments across diverse accelerators; and 3) they fragment coverage across pretraining, fine-tuning, or inference, failing to provide an end-to-end perspective on the full lifecycle of model efficiency. To address these gaps, we present **EfficientLLM**, the first large-scale empirical benchmark that systematically quantifies efficiency–performance trade-offs across the entire lifecycle of LLMs. 1) First, to overcome missing multi-dimensional metrics, EfficientLLM unifies six orthogonal dimensions into a consistent evaluation framework. 2) Second, to address scale and hardware diversity, we evaluate over 150 model–technique pairs spanning 0.5B–72B parameters on production-class clusters with 48*GH200, 8*H200, and 8*A100 accelerators, ensuring conclusions generalize to realistic deployment conditions. 3) Third, to provide end-to-end lifecycle coverage, EfficientLLM benchmarks architectural pretraining, fine-tuning, and bit-width quantization. By systematically resolving these three limitations, EfficientLLM establishes the most comprehensive benchmark to date for evaluating efficiency in large-scale models. Our results not only highlight critical trade-offs between accuracy, cost, and sustainability but also offer actionable guidance for both academic researchers and industrial practitioners in designing, training, and deploying the next generation of foundation models. All code and datasets are released as an open-source toolkit, accessible via `pip install efficientllm-toolkit`.

**Note:** All values presented in our figures are normalized within each metric across all models. For consistency, all metrics (e.g., PPL, FID, and etc.) are transformed such that higher values (except Loss) indicate better performance or efficiency.

## 1 INTRODUCTION

Large Language Models (LLMs), such as GPT-style architectures (Brown et al., 2020) and Pathways Language Model (PaLM) (Chowdhery et al., 2022), are a key type of Foundation Model that have driven significant breakthroughs across numerous domains. These models, often characterized by billions or even trillions of parameters (Brown et al., 2020; Chowdhery et al., 2022), achieve remarkable performance by leveraging deep learning techniques and training on massive datasets (Kaplan et al., 2020b; Hoffmann et al., 2022a; Pandya & Holia, 2023; Agarwal et al., 2024; Xu et al., 2024a), typically comprising trillions of tokens from diverse sources like the web, books, and code. LLMs demonstrate powerful capabilities in complex tasks including nuanced language generation, sophisticated reasoning, and problem-solving. However, the impressive capabilities of LLMs come at substantial computational and environmental costs. For instance, training GPT-3

(175B parameters) required approximately 3,640 Petaflop/s-days, costing millions of dollars in cloud computing resources (Kaplan et al., 2020b). Similarly, Google's PaLM (540B parameters) required thousands of TPUv4 chips running continuously for extended periods (Chowdhery et al., 2022). Deploying these models at scale also incurs significant hardware and energy costs, contributing to considerable carbon emissions (Strubell et al., 2019b). As LLMs proliferate, there is a pressing need for rigorous benchmarks that quantify efficiency–performance trade-offs to guide academic research, industrial traning, budgeting, and environmental sustainability.

However, existing evaluations of efficiency techniques for LLMs suffer from several critical limitations. 1) First, lack of multi-dimensional metrics: most studies report only isolated measures such as FLOPs or latency, while overlooking complementary dimensions like memory utilization, energy consumption, and throughput (Poddar et al., 2025; Arya & Simmhan, 2025). As a result, efficiency gains are often mischaracterized, and comparisons across methods lack consistency. 2) Second, insufficient scale and hardware diversity: evaluations are frequently conducted on small models or restricted hardware settings, making their conclusions difficult to generalize to production-scale deployments with billion-parameter models and heterogeneous accelerators (Bast et al., 2024; Niu et al., 2025; Wang et al., 2019; Samsi et al., 2023). This gap risks misleading both academic and industrial stakeholders when extrapolating to real-world scenarios. 3) Third, fragmented lifecycle coverage: prior benchmarks typically focus on a single stage, pretraining, fine-tuning, or deployment, without providing an end-to-end perspective (Niu et al., 2025; Shamshoum et al., 2024). Such fragmentation prevents practitioners from understanding trade-offs across the full model lifecycle, limiting their ability to make informed decisions on budgeting, deployment, and environmental sustainability. Without addressing these gaps, LLMs may appear efficient on surface-level benchmarks but cannot be reliably evaluated for their true resource trade-offs in deployment. Large-scale, real-world efficiency benchmarks are therefore essential to provide trustworthy guidance for model development, deployment decisions, and sustainable scaling of foundation models.

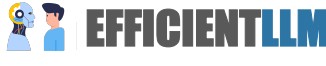

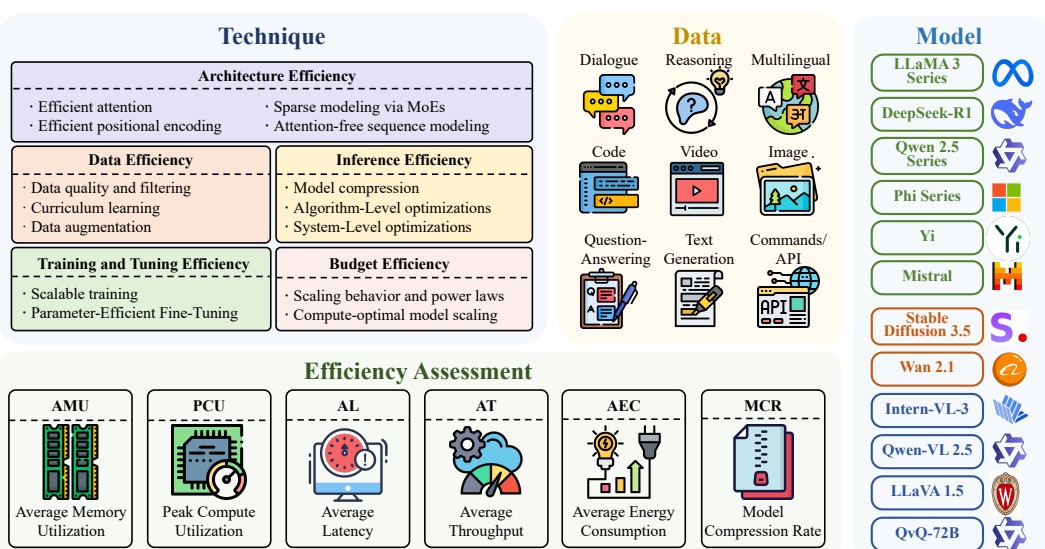

Figure 1: Overview of the EfficientLLM framework.

To address these limitations, we introduce EFFICIENTLLM, the largest and most comprehensive benchmark to date for evaluating efficiency in large-scale language models. 1) First, to overcome the problem of missing multi-dimensional metrics, EFFICIENTLLM systematically measures six orthogonal efficiency dimensions, like Average-Memory-Utilization, Peak-Compute-Utilization and etc. 2) Second, to address the lack of scale and hardware diversity, our benchmark is executed on a production-class cluster (48×GH200, 8×H200 GPUs), evaluating over 100 model–technique pairs spanning 0.5B–72B parameters. This large-scale setting ensures that conclusions generalize beyond toy examples, reflecting realistic deployment conditions across heterogeneous accelerators.

| Category | Group | Size | Method | Performance | AMU | PCU | AL | TT | ST | IT | AEC | MCR |
|---|---|---|---|---|---|---|---|---|---|---|---|---|
| Architecture Efficiency | Attention | | MQA | 4 | 1 | | 1 | 1 | | | 3 | |
| | | | GQA | 3 | 3 | | 2 | 2 | | | 4 | |
| | | | MLA | 1 | 4 | | 3 | 3 | | | 2 | |
| | | | NSA | 2 | 2 | | 4 | 4 | | | 1 | |
| | PosEnc | | RoPE | 1 | 2 | | 2 | 2 | | | 2 | |
| | | | Absolute | 4 | 4 | | 4 | 4 | | | 4 | |
| | | | Learnable Absolute | 2 | 3 | | 3 | 3 | | | 3 | |
| | | | Relate | 3 | 1 | | 1 | 1 | | | 1 | |
| | MoE | | Dense Model 1.5B | 4 | 2 | | 2 | 3 | | | 2 | |
| | | | Dense Model 3B | 3 | 1 | | 1 | 4 | | | 1 | |
| | | | MoE Model 1.5Bx8 | 1 | 4 | | 4 | 1 | | | 4 | |
| | | | MoE Model 0.5Bx8 | 2 | 3 | | 3 | 2 | | | 3 | |
| | Att-free | | Transformer | 1 | 4 | | 4 | 1 | | | 4 | |
| | | | Mamba | 2 | 1 | | 1 | 4 | | | 1 | |
| | | | Pythia | 4 | 3 | | 3 | 3 | | | 3 | |
| | | | RWKV | 3 | 2 | | 2 | 2 | | | 2 | |
| Training and Tuning Efficiency | Parameter-Efficient Fine-Tuning | 1-3B | LoRA | 7 | 2 | 2 | 3 | | 2 | | 3 | |
| | | | LoRA-Plus | 4 | 5 | 6 | 4 | | 3 | | 2 | |
| | | | RSLoRA | 5 | 4 | 5 | 5 | | 4 | | 5 | |
| | | | DoRA | 6 | 7 | 3 | 7 | | 6 | | 6 | |
| | | | PiSSA | 3 | 6 | 4 | 6 | | 5 | | 4 | |
| | | | Freeze | 1 | 3 | 7 | 1 | | 7 | | 1 | |
| | | | Full* | 2 | 1 | 1 | 2 | | 1 | | 7 | |
| | | 7-8B | LoRA | 2 | 1 | 5 | 5 | | 4 | | 3 | |
| | | | LoRA-Plus | 5 | 2 | 7 | 6 | | 5 | | 2 | |
| | | | RSLoRA | 3 | 4 | 6 | 3 | | 3 | | 5 | |
| | | | DoRA | 4 | 6 | 3 | 7 | | 7 | | 6 | |
| | | | PiSSA | 6 | 3 | 4 | 4 | | 5 | | 4 | |
| | | | Freeze | 1 | 7 | 2 | 1 | | 1 | | 1 | |
| | | | Full* | 7 | 5 | 1 | 2 | | 2 | | 7 | |
| | | 14-24B | LoRA | 3 | 1 | 6 | 2 | | 3 | | 6 | |
| | | | LoRA-Plus | 4 | 3 | 7 | 6 | | 7 | | 1 | |
| | | | RSLoRA | 1 | 2 | 5 | 4 | | 4 | | 5 | |
| | | | DoRA | 6 | 7 | 3 | 7 | | 2 | | 4 | |
| | | | PiSSA | 2 | 4 | 4 | 3 | | 4 | | 3 | |
| | | | Freeze | 5 | 6 | 2 | 1 | | 1 | | 2 | |
| | | | Full* | 7 | 5 | 1 | 5 | | 6 | | 7 | |
| Inference Efficiency | Quantization | 1.5-3.8B | bfloat16 | 3 | 3 | | 3 | | 3 | | 2 | 5 |
| | | | float16 | 2 | 4 | | 2 | | 5 | | 5 | 4 |
| | | | int8 | 4 | 2 | | 4 | | 2 | | 3 | 3 |
| | | | fp8 | 1 | 5 | | 1 | | 4 | | 4 | 2 |
| | | | int4 | 5 | 1 | | 5 | | 1 | | 1 | 1 |
| | | 7-8B | bfloat16 | 3 | 4 | | 3 | | 2 | | 2 | 4 |
| | | | float16 | 4 | 3 | | 1 | | 5 | | 4 | 5 |
| | | | int8 | 2 | 2 | | 4 | | 3 | | 3 | 3 |
| | | | fp8 | 1 | 1 | | 2 | | 4 | | 5 | 2 |
| | | | int4 | 5 | 5 | | 5 | | 1 | | 1 | 1 |
| | | 14-34B | bfloat16 | 2 | 3 | | 3 | | 3 | | 3 | 4 |
| | | | float16 | 4 | 4 | | 1 | | 5 | | 2 | 5 |
| | | | int8 | 3 | 2 | | 4 | | 2 | | 4 | 3 |
| | | | fp8 | 1 | 5 | | 2 | | 4 | | 5 | 2 |
| | | | int4 | 5 | 1 | | 5 | | 1 | | 1 | 1 |

Figure 2: Ranking of LLM training and inference efficiency and performance across various techniques. For parameter-efficient tuning, "Freeze" refers to the method, which freezes the first 8 layers of the model. "Full*", utilize DeepSpeed ZeRO-3 Offload CPU.

3) Third, to fill the gap of fragmented lifecycle coverage, EFFICIENTLLM adopts a unified taxonomy covering three critical stages, architecture pretraining, fine-tuning, and quantization. This design provides end-to-end guidance: from budgeting computational and energy costs during architecture design, to selecting efficient PEFT methods for domain adaptation, to identifying quantization strategies that reduce serving cost and latency without retraining. By systematically addressing these three limitations, EFFICIENTLLM establishes the first large-scale empirical benchmark that rigorously quantifies efficiency–performance trade-offs across the full LLM lifecycle, providing a trusted foundation for both academic research and industrial deployment.

## 1.1 NEW INSIGHTS

**Architecture Pretraining.** 1) For commercial settings aimed at establishing new SOTA benchmarks,, the combination of Multi-Head Latent Attention (MLA) and Rotary Position Embeddings (RoPE) is recommended. This configuration consistently yielded the lowest perplexity (PPL) across our evaluations. And the Mixture-of-Experts (MoE) architecture is the ideal framework for exploring model capability scaling. Our empirical results, which show a 1.5B×8 MoE model outperforming a 3B dense model, provide a robust basis for investigating the trade-off between memory overhead and gains

in computational efficiency and model intelligence. 2) In compute-constrained academic settings, Grouped-Query Attention (GQA) is identified as the optimal choice. It provides a robust balance between model performance and training expenditure (i.e., memory and latency), thereby avoiding the substantial overhead associated with performance–centric methods like MLA. And more efficient components such as **Relative Position Embeddings (Relate)** can significantly accelerate the research cycle by reducing training time and cost with a negligible impact on performance.

**Training and Tuning Efficiency.** 1) For commercial settings, RSLoRA represent the gold standard for model fine-tuning in production for large-scaling language models. They offer an optimal balance of performance, stability, and resource efficiency, making them particularly suitable for managing a large portfolio of customized models. 2) For academic research under tight computational budgets, Parameter Freezing is the recommended strategy. Its exceptional training speed enables researchers to conduct a greater number of experiments within the same timeframe, thereby accelerating discovery. And the low VRAM footprint of LoRA and Freeze methods makes the fine-tuning of large models (e.g., 7B, 14B) feasible on single consumer or mid-tier professional GPUs. This significantly enhances the accessibility of large-scale model research for the academic community.

**Inference Efficiency** 1) Default Strategy for Large-Scale Deployment, for any large-scale, user-facing service, Float 8 quantization is the most pragmatic and cost-effective strategy. The substantial operational savings in memory, throughput, and energy far outweigh the modest, often user-imperceptible, degradation in performance. 2) Academic and Resource-Constrained Settings. In scenarios where computational resources are limited—such as in academic research, Float 8 quantization can be effectively applied to smaller models (e.g., Qwen2.5-7B). For larger models, more aggressive quantization such as INT4 is a practical choice, enabling deployment without prohibitive hardware costs while still retaining acceptable levels of accuracy.

## 1.2 ROAD MAP

The remainder of this paper is structured as follows. Appendix B provides background information on foundation models and discusses fundamental approaches to enhancing efficiency. Appendix C details the specific efficiency improvement techniques evaluated within our framework. Section 2 and Appendix D.1 defines our proposed efficiency assessment principles and metrics. Section D.2 describes the curated list of models and experimental settings used in our benchmark. Sections 3 and Appendix D.3, D.4, and D.5 present detailed empirical results for architecture, training/tuning, and inference efficiency, respectively. Finally, Appendix G discusses remaining challenges and future research directions, and Section 4 concludes the paper. The Ranking of these technologies show in the Figure 2.

## 2 EFFICIENTLLM: A FRAMEWORK FOR EVALUATING THE LLMS EFFICIENCY

### 2.1 EXPERIMENTAL DEVICES

All experiments were conducted on NVIDIA high-performance computing platforms. For pretraining, we utilized 12 GH200 Superchips nodes, each integrating a Grace CPU (144 cores) with 4*GPU (96G). Fine-tuning and inference were primarily executed on 8*H200 (141G) GPUs paired with Intel Xeon Platinum 8558 CPUs, while additional experiments on the Medical-O1 dataset were performed on multi-device 8*A100 (80G) clusters to verify cross-device reproducibility. This setup covers a representative spectrum of modern accelerators and CPUs for efficiency benchmarking.

### 2.2 MODELS AND DATASETS.

We evaluate a diverse set of state-of-the-art LLM architectures, including DeepSeek-R1 (Bi et al., 2024; Guo et al., 2025), Qwen 2.5 Series (Bai et al., 2023; Yang et al., 2024), Phi Series (Abdin et al., 2024), Yi (Young et al., 2024), Mistral, and Mixtral (Jiang et al., 2023; 2024a), across multiple scales (from 0.5B to 72B parameters). For pretraining, we utilize the Fine-web Edu dataset (350B tokens). For fine-tuning we utilize the O1-SFT and Medical-O1, and inference evaluations, we employ the performance benchmarks including MMLU-Pro, BBH, GPQA, IFEval, MATH, and MUSR, as detailed in Section D.2.

## 2.3 ASSESSMENT PRINCIPLES OF EFFICIENTLLM

In practice, widely used efficiency metrics such as FLOPS, parameter count, and raw inference speed provide only a partial view of LLM efficiency (Liu et al., 2023b; Perez et al., 2023; Bao et al., 2023; Zhao et al., 2025; Ye et al., 2025). These measures often overlook dynamic system behaviors such as fluctuating memory usage, synchronization delays, communication overhead, and energy cost, all of which critically affect real-world deployment. As a result, conventional metrics fail to capture the true bottlenecks that determine whether a model can be trained and served efficiently at scale. To address these limitations, we propose a set of metrics specifically designed for large-scale training and deployment scenarios. For more details, for example, a detailed explanation, detailed calculation process and solution motivation are available in Appendix D.1.

### 2.3.1 COMPUTATIONAL SYSTEM UTILIZATION

Existing benchmarks such as MLPerf (Reddi et al., 2020), SPEC CPU (Standard Performance Evaluation Corporation, 2024), evaluate resource optimization, while tools like LLMPerf (Project, 2024) focus on specific aspects such as latency, scalability, or hardware adaptability. However, predominant metrics—e.g., latency, training time, or accuracy (Yang et al., 2023; Hu et al., 2021b)—fail to capture key factors like memory bandwidth, device utilization, and throughput. So LMs often suffer from suboptimal hardware usage during training and inference, increasing operational costs (Xia et al., 2023; Bang, 2023). In this work, we define computational system utilization as the efficient and effective use of hardware resources across training and inference, assessed via four dimensions.

**Memory Utilization (AMU).** Efficient memory usage is critical since limited device memory often becomes the bottleneck in LM training. We define the *Average Memory Utilization (AMU)* as

$$AMU = \frac{1}{T} \int_0^T \text{Memory Used}(t)\, dt \tag{1}$$

where $T$ is total training time and Memory Used$(t)$ is memory allocated at time $t$. Higher AMU indicates efficient and stable memory usage, while lower AMU suggests fragmentation and wastage.

**Compute Utilization (PCU).** Maximizing GPU usage is essential for reducing training cost and energy waste. We define the *Peak Compute Utilization (PCU)* as

$$PCU = \frac{1}{T} \int_0^T \frac{\text{Actual GPU Utilization}(t)}{\text{Peak GPU Utilization}}\, dt \tag{2}$$

where $T$ is training time and Peak GPU Utilization is the theoretical maximum (100%). High PCU reflects effective and minimal idle time, while low PCU indicates compute underutilization.

**Latency (AL).** Latency determines responsiveness and efficiency in both training and inference. We define the *Average Latency (AL)* as

$$AL = \frac{1}{N} \sum_{i=1}^{N} \left( \text{Computation Time}_i + \text{Communication Time}_i \right) \tag{3}$$

where $N$ is the number of iterations or requests. Lower AL reflects faster response and better scheduling, while higher AL reveals bottlenecks in computation or communication.

**Throughput (TT, ST, IT).** Throughput measures how efficiently data is processed across tasks. We define *Token Throughput (TT)* for pretraining, *Sample Throughput (ST)* for fine-tuning, and *Inference Throughput (IT)* for inference:

$$\begin{aligned}
TT &= \frac{\sum_i(\text{Tokens Processed}_i/\text{Model Parameters})}{\sum_i \text{Time}_i}, \\
ST &= \frac{\sum_i(\text{Samples Processed}_i/\text{Model Parameters})}{\sum_i \text{Time}_i}, \\
IT &= \frac{\sum_i \text{Tokens Generated}_i}{\sum_i \text{Time}_i}.
\end{aligned} \tag{4}$$

Higher TT, ST, and IT indicate more efficient scaling and faster data handling, while lower values reveal inefficiencies.

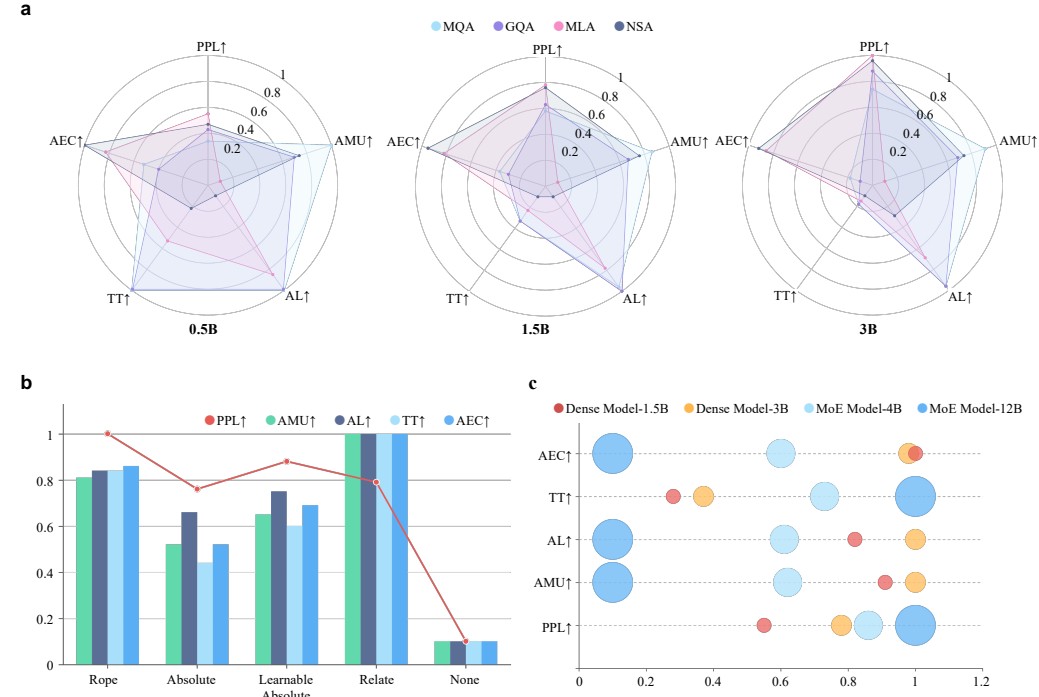

Figure 3: Efficiency LLM Results. This figure illustrates the performance and efficiency trade-offs of various architectural improvements for LLMs. **(a)** Comparing different Efficient Attention Mechanisms (MQA, GQA, MLA, and NSA) across 0.5B, 1.5B, and 3B model parameters, evaluated on Perplexity (PPL). **(b)** Evaluating Efficient Positional Encoding methods (RoPE, Absolute, Learnable Absolute, Relate, and None) for a 1.5B parameter models. **(c)** Comparing Dense Models with MoE Models of varying parameter sizes, highlighting differences in PPL, AMU, AL, TT, and AEC. **Note:** All metrics presented in this figure are normalized, as deilted in Section H.7.1. Detailed results are shown in the Table 3, 4, 5, and 6.

### 2.3.2 ENERGY CONSUMPTION

**Average Energy Consumption (AEC).** Energy use is a key efficiency concern for large-scale AI training and deployment. We define the *Average Energy Consumption (AEC)* as

$$AEC = \frac{1}{T} \int_0^T P(t)\, dt \tag{5}$$

where $P(t)$ is instantaneous power in Watts and $T$ is total time. Lower AEC denotes more energy-efficient operation, while higher AEC implies costly and less sustainable usage.

### 2.3.3 MODEL COMPRESSION RATE

**Model Compression Rate (MCR).** Compression evaluates storage and deployment efficiency under performance retention. We define the *Model Compression Rate (MCR)* as

$$MCR_{\text{(Performance)}} = \frac{\text{Size}_{\text{original}}}{\text{Size}_{\text{compressed}}} \times \frac{\text{Performance}_{\text{compressed}}}{\text{Performance}_{\text{original}}} \tag{6}$$

where sizes are in bytes and performance is accuracy. Higher MCR reflects compact yet effective compression, while low values suggest that performance degradation outweighs size reduction.

### 2.3.4 MODEL PERFORMANCE

We assess reasoning, coding, mathematics, and instruction-following ability using established benchmarks: MMLU-Pro (Wang et al., 2024d), BBH (Suzgun et al., 2022), GPQA (Rein et al., 2024), IFEval (Zhou et al., 2023), HumanEval (Chen et al., 2021a), HARDMath (Fan et al., 2024b), and MuSR (Sprague et al., 2023). Each targets complementary skills, from domain knowledge to multi-step reasoning, enabling a holistic view of LLM capabilities.

## 3 MAIN RESULTS

### 3.1 ARCHITECTURE PRETRAINING

The comprehensive assessment of architecture pretraining efficiency, as detailed in Appendix D.3, provides several critical insights across different model configurations and efficiency techniques.

**Efficient Attention Mechanisms:** As shown in Table 3 and Figure 3(a), Multi-Query Attention (MQA) stands out with superior memory efficiency, exhibiting the lowest Average Memory Utilization (AMU) of 42.24 GB and competitive latency of 0.1298 seconds per iteration at the 1.5B parameter scale. In contrast, Multi-Head Latent Attention (MLA) consistently achieved the lowest perplexity scores across all evaluated model sizes (PPL = 8.73, 7.79, and 7.29 for 0.5B, 1.5B, and 3B respectively), making it preferable for scenarios where model accuracy is paramount. Native Sparse Attention (NSA), although less performant in terms of perplexity, offered remarkable energy efficiency (AEC = 594.23 W at the 0.5B scale), underscoring its suitability for energy-sensitive deployments. Grouped-Query Attention (GQA) provided a balanced compromise, especially evident at the 1.5B scale where it achieved the lowest latency (AL = 0.1283 s/iter).

**Efficient Positional Encoding:** As shown in Table 4 and Figure 3(b), Rotary Position Embeddings (RoPE) demonstrated the best model performance (lowest perplexity of 8.09). However, Relative Positional Encoding (RPE, denoted as Relate) excelled in computational efficiency metrics, achieving the lowest memory usage (AMU = 43.94 GB), lowest latency (AL = 0.1246 s/iter), and highest tokens throughput (TT = $8.98 \times 10^{-02}$). Conversely, models trained without positional encoding showed significantly degraded performance (PPL = 8.75), emphasizing the critical role of positional encoding in sequence modeling effectiveness.

**Sparse Modeling via Mixture of Experts (MoE):** As shown in Table 5 and Figure 3(c), MoE models significantly outperformed dense configurations in perplexity, with the 1.5B×8 MoE model achieving a perplexity of 7.10 compared to 8.09 for the dense 1.5B model. However, these improvements were accompanied by increased resource demands, with higher memory utilization and energy consumption, highlighting a clear trade-off between performance and efficiency.

**Attention-Free Alternatives:** As shown in Table 6, Mamba presented remarkable efficiency advantages, including the lowest memory utilization (AMU = 29.16 GB at 0.5B) and lowest energy consumption (AEC = 498.37 W). Despite these benefits, Mamba's perplexity was consistently higher than transformer-based architectures, reflecting a trade-off where improved efficiency comes at the expense of lower model performance. RWKV provided moderate improvements in memory and energy efficiency, whereas Pythia, while competitive in latency, lagged notably in perplexity.

### 3.2 TRAINING AND TUNING EFFICIENCY

As detailed in Section D.4, the evaluation of training and tuning efficiency across multiple model architectures and fine-tuning techniques highlights critical trade-offs between performance and computational resources.

#### 3.2.1 PEFT METHODS

**O1-SFT Dataset:** As shown in Figure 4(a) and Table 7 our findings demonstrate that for smaller models (1-3B parameters), LoRA-plus consistently achieved superior performance with the lowest loss metrics (0.7442 for Llama-3.2-1B and 0.5791 for Llama-3.2-3B), while maintaining reasonable memory utilization. Parameter freezing consistently offered the lowest average latency across model sizes, making it optimal for latency-critical applications, though at times compromising on model performance. RSLoRA exhibited strong performance for larger models, particularly Qwen-2.5-14B (loss = 0.4126) and Mistral-Small-24B (loss = 0.3818). In contrast, full fine-tuning using DeepSpeed optimization showed diminishing returns as model scale increased, especially notable at the 24B parameter level, where it incurred high resource demands with comparatively poorer performance.

**Medical-O1 Dataset:** As shown in Table 9, parameter freezing demonstrated exceptional efficiency, achieving the lowest loss and latency across all tested model scales (1B to 8B). LoRA-plus provided a robust balance, combining competitive loss and energy efficiency. Conversely, methods like DoRA incurred significantly higher latency and resource utilization without commensurate performance

a

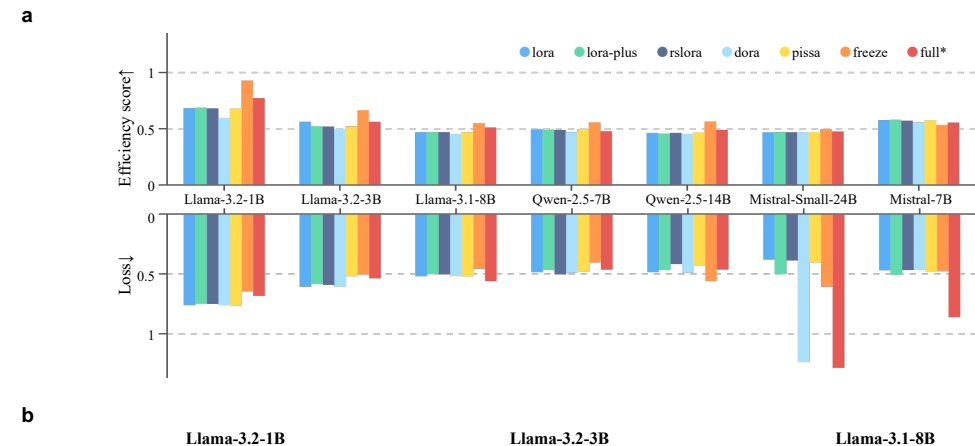

b

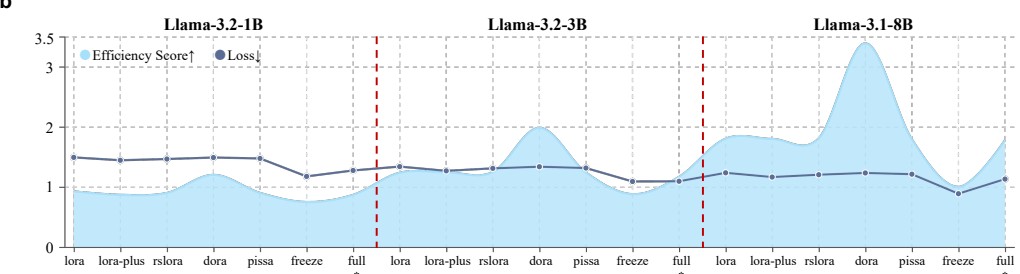

Figure 4: **Assessment of training and fine-tuning efficiency across multiple LLMs.** (a) Comparison of different fine-tuning methods (LoRA, LoRA-plus, RSLoRA, DoRA, PISSA, Freeze, and full fine-tuning using DeepSpeed) across seven model architectures (Llama-3.2-1B/3B, Llama-3.1-8B, Qwen-2.5-7B/14B, Mistral-Small-24B, and Mistral-7B) using the O1-SFT dataset. Each bar shows the corresponding **Efficiency Score** (higher is better) and **Loss** (lower is better). The Efficiency Score is computed as a weighted harmonic combination of normalized resource metrics, as deilted in Section H.7.2. Methods marked with * denote full fine-tuning using DeepSpeed. Detailed results are shown in the Table 7 and 9.

improvements. These findings underscore the importance of selecting parameter-efficient fine-tuning strategies tailored to specific computational constraints and desired performance outcomes, with parameter freezing being particularly suitable for latency-sensitive medical applications.

**Backbone dependency.** We have conducted additional experiments on a diverse set of 7B-scale models on SFT-O1, as shown in Table 8. These models differ significantly in architecture, tokenizer, and training corpus. As shown in the updated Table (included above), consistent patterns emerge across these diverse backbones: for example, freeze-tuning consistently achieves the lowest loss and highest PCU, while LoRA and LoRA-plus demonstrate strong efficiency in memory (AMU) and energy cost (AEC). The stability of these trends across architectures suggests that our conclusions are not limited to specific backbones, but rather reflect robust, transferable properties of the fine-tuning methods themselves.

**Medical-O1 Dataset on A100.** As shown in Table 10, parameter freezing achieved the best efficiency with the lowest loss and latency across model scales, making it ideal for latency-critical medical applications. LoRA-plus provided a balanced trade-off between convergence quality and memory efficiency, while RSLoRA was more effective on larger backbones. In contrast, DoRA consistently incurred higher latency and energy costs without proportional gains. These results confirm that the relative strengths of parameter-efficient methods are consistent across hardware, with A100 experiments reinforcing the robustness of our findings.

### 3.2.2 MIXED PRECISION TRAINING

As shown in Table 11, mixed-precision strategies significantly improved efficiency compared to full BF16 training. INT4-based configurations consistently yielded the lowest memory footprint (down to 19.8 GB on Llama-3.2-1B) and latency, while maintaining competitive throughput and energy efficiency, though at the cost of moderately higher loss values. FP8 and INT8 achieved a better

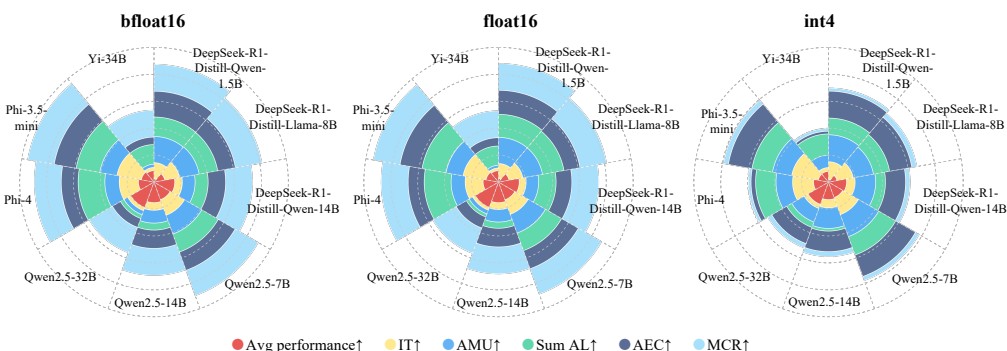

Figure 5: **Assessment of quantization-based inference efficiency across model precisions.** Radar plots compare normalized efficiency metrics across three quantization formats: **bfloat16**, **float16**, and **int4**. Each plot evaluates models from DeepSeek, Qwen, Phi, and Yi families using six normalized metrics (all ↑ higher is better): average task performance, inference throughput (IT), average memory utilization (AMU), sum latency (Sum AL), average energy consumption (AEC), and model compression ratio (MCR). All values are normalized as deilted in Section H.7.1. While bfloat16 typically yields higher performance scores, int4 excels in throughput, memory, and compression, indicating its efficiency in deployment-constrained environments. Detailed results are shown in the Table 12 and 19.

balance, offering lower loss closer to BF16 with notable reductions in memory and energy usage. For example, FP8(W)FP16(T) attained loss values of 1.1756 on Llama-3.1-8B, only slightly higher than BF16 (1.1290), while halving memory consumption. Overall, these results highlight that **INT4 is optimal for extreme memory-constrained scenarios**, whereas **FP8 provides the best trade-off between performance and efficiency**, making it well-suited for large-scale medical applications with strict resource budgets.

### 3.3 BIT-WIDTH QUANTIZATION INFERENCE EFFICIENCY

As shown in Table 12 and Figure 5, bit-width quantization demonstrates consistent efficiency–performance trade-offs across diverse model families and scales. For smaller backbones such as **DeepSeek-R1-Distill-Qwen-1.5B**, INT4 achieved the lowest memory footprint (19.49 GB) and highest throughput (42.34 tokens/s), though at the expense of reduced performance (Avg Perf. = 0.2341). Similar trends were observed in larger models like **Qwen2.5-32B**, where INT4 reduced memory to 48.30 GB while sustaining competitive throughput, underscoring its practicality for memory- or latency-constrained deployments. Intermediate formats such as **FP8** and **INT8** provided robust trade-offs: FP8 consistently delivered slightly higher average performance (e.g., 0.4755 for Qwen2.5-14B) compared to bfloat16/float16 while reducing memory costs, whereas INT8 tended to balance efficiency and accuracy with moderate energy consumption.

### 3.4 SCALABILITY OF EFFICIENTLLM

EFFICIENTLLM is inherently scalable beyond text-only LLMs, as the same efficiency metrics and evaluation pipeline can be directly applied to vision and vision–language models. Techniques such as efficient attention, MoE, PEFT, and quantization validated on LLMs are shown to transfer effectively to LVMs, VLMs, and diffusion transformers, confirming the benchmark's modality-agnostic design. All extended experiments and detailed results are provided in Appendix E.

## 4 CONCLUSION

In this study, we introduced EFFICIENTLLM, the first extensive empirical evaluation of efficiency techniques for large language models across language, vision, and multimodal tasks. Our systematic benchmarking across over 150 model-technique combinations highlighted crucial trade-offs in resource usage, latency, and throughput. Ultimately, our results emphasize the importance of adopting a multi-dimensional, Pareto-optimized approach to model efficiency, offering actionable insights for practitioners seeking sustainable, scalable deployment of generative AI models.

ETHICS STATEMENT

This work focuses on benchmarking the efficiency of large language models (LLMs) across pretraining, fine-tuning, and inference. Our study does not involve the collection or release of PII, sensitive medical records, or other private user data. All datasets used in this work, including FineWeb, O1-SFT, and Medical-O1, are either publicly available or internally curated with strict anonymization procedures to ensure compliance with data protection standards. The purpose of this benchmark is to provide transparent, reproducible, and resource-aware evaluations that enable both academia and industry to make informed decisions on model development and deployment.

REPRODUCIBILITY STATEMENT

All experiments in this paper were conducted on well-specified hardware clusters (48×GH200, 8×H200, and 8×A100 GPUs) with detailed hyperparameters, datasets, and evaluation metrics described in Appendix D.2 and Appendix D.1. To facilitate reproducibility, we release all code, configuration files, and datasets as part of the `efficientllm-toolkit`, which can be directly accessed via `pip install efficientllm-toolkit`.

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

# Appendices

APPENDIX CONTENTS

## A  OBSERVATIONS AND INSIGHTS

To facilitate the overall understanding of our study, in this section, we first present the observations and insights we have drawn based on our extensive empirical experiments in the EfficientLLM framework.

### A.1  OVERALL OBSERVATIONS

***No single technique achieves Pareto optimality on all efficiency axes.*** Our benchmark, involving over 100 model-technique combinations run across 48 GH200 and 8 H200 GPUs, revealed that every evaluated method improved at least one metric (memory, latency, throughput, energy, or compression) while compromising others. For example, Mixture-of-Experts (MoE) architectures (Fedus et al., 2022; Jiang et al., 2024a) boosted downstream accuracy and reduced FLOPs per token during inference (by activating only a subset of parameters), yet inflated peak memory requirements due to the need to store all expert parameters, and introduced routing overhead. Our experiments showed MoE could increase VRAM usage by approximately 40% compared to a dense model of equivalent active parameter count (detailed in Section D.3). Conversely, post-training int4 quantization slashed memory footprint and energy consumption by up to $3.9\times$ but incurred a modest average-task performance drop of approximately 3–5% across tested models (detailed in Section D.5) (Wu et al., 2023). These quantified trade-offs highlight that efficiency must be treated as a multi-objective optimization problem , not reducible to a single leaderboard score. This observation provides strong empirical validation for the No-Free-Lunch (NFL) theorem (Wolpert & Macready, 1997) in the context of LLM efficiency. While the NFL theorem, originally formulated by Wolpert and Macready , states theoretically that no single algorithm universally outperforms others across all possible problems when averaged, our benchmark demonstrates this principle concretely. The results across numerous model-technique pairs and six distinct efficiency metrics quantify the specific costs associated with gains for practical LLM optimization strategies, moving beyond theoretical averages to specific, measured outcomes.

***Resource-Driven Trade-Offs in Efficient Attention Mechanisms.*** Our tests on models ranging from 0.5 B to 3 B parameters showed distinct advantages among the four efficient attention mechanisms evaluated: Multi-Query Attention (MQA) (Shazeer, 2019), Grouped-Query Attention (GQA) (Ainslie et al., 2023), Multi-Head Latent Attention (MLA) (Liu et al., 2024a), and Native Sparse Attention (NSA) (Yuan et al., 2025). MQA delivered the lowest VRAM footprint (due to sharing key/value heads) and fastest latency, making it preferable for memory-constrained environments or on-device inference. MLA, introduced by DeepSeek (Liu et al., 2024a) to compress the KV cache into a latent vector, minimized perplexity in our tests, rendering it attractive when raw language quality is paramount. NSA, designed as a hardware-aligned and natively trainable sparse attention mechanism, consumed the least energy per generated token in our evaluations, favouring low-power deployments or scenarios where energy cost is a primary concern. These results confirm that a "one-size-fits-all" attention mechanism does not exist; the benchmark data enables practitioners to make evidence-based selections, aligning the variant with their dominant resource bottleneck or performance goal (e.g., minimizing latency vs. maximizing quality vs. minimizing energy).

***Parameter-efficient fine-tuning (PEFT) methods scale differently with model size.*** We observed that Low-Rank Adaptation (LoRA) (Hu et al., 2022) and its derivatives, such as DoRA (Weight-Decomposed Low-Rank Adaptation) (Liu et al., 2024d) and other variants collectively referred to as LoRA-plus, achieved the lowest performance loss (i.e., best task performance metrics like accuracy or lowest loss values) for models in the 1 B to 3 B parameter range under specific memory constraints. However, RSLoRA (Kalajdzievski, 2023), another LoRA variant, overtook the original LoRA in terms of efficiency, exhibiting lower latency and wattage, specifically for models with 14 B parameters or more. For ultra-large checkpoints, our analysis indicated that parameter freezing (updating only specific layers or components like biases) produced the best end-to-end latency during the tuning process, albeit sometimes at a small cost in final task accuracy compared to LoRA-based methods. Consequently, selecting the appropriate PEFT method based on the target model's scale yields larger efficiency gains than uniformly applying a single technique. This highlights a scale-dependent interaction effect, suggesting that findings from smaller models regarding the relative merits of different PEFT techniques may not directly extrapolate to significantly larger models.

***Lower-precision formats deliver disproportionate returns on memory-bound workloads.*** Our quantitative analysis across Llama-3, DeepSeek, and Qwen models (1.5B to 34B) indicates that

int4 post-training quantization significantly improves resource efficiency. Compared to bfloat16, int4 reduced the memory footprint by up to 3.9× (approaching the theoretical maximum of 4× reduction from 16-bit to 4-bit representation) and tripled the throughput in tokens per second (TPS) under memory-bound conditions. This substantial gain came at the cost of only a slight drop in average task performance scores (e.g., for DeepSeek-R1-Distill-Qwen-14B (Guo et al., 2025), the average score dropped from 0.4719 in bf16 to 0.4361 in int4). The term disproportionate returns here signifies that the substantial gains achieved in resource efficiency (memory footprint reduction approaching 4x, throughput tripling) far outweigh the relatively small cost incurred in terms of task performance degradation (average drop of 3-5 percentage points). This makes int4 highly attractive when memory, energy, or cost are primary constraints. Between the 16-bit floating-point formats, bfloat16 consistently outperformed float16 in terms of average latency and energy consumption on our Hopper architecture GPUs (GH200/H200). This is attributed to the native hardware acceleration (Tensor Cores) for bfloat16 operations on these modern NVIDIA GPUs. This suggests that adopting a "BF16-first" strategy is a safe default if quantization is not feasible or if the associated performance drop is unacceptable for the target application.

## A.2 NOVEL INSIGHTS DERIVED FROM THE EFFICIENTLLM BENCHMARK

**Architecture Pretraining Efficiency.** Architecture pretraining efficiency involves balancing memory, latency, and quality trade-offs during the pretraining stage. Our benchmark yielded the following architectural insights, as shown in Figure 3: 1) *Attention variants have distinct optima during pretraining*: Among the four efficient attention variants tested in pretraining, our quantitative analysis shows **MQA** hits the best memory–latency frontier, **MLA** achieves the lowest perplexity, and **NSA** minimizes energy consumption. 2) *MoE presents a compute-memory trade-off in pretraining*: We confirmed that sparse *Mixture-of-Experts (MoE)* during pretraining can add up to 3.5 percentage points in accuracy while cutting training FLOPs by 1.8×. However, this comes at the cost of inflating VRAM usage by 40%, highlighting a clear tension between compute savings and memory demands. 3) *Attention-free models offer pretraining efficiency gains with quality trade-offs*: Our evaluation showed that attention-free *Mamba* models during pretraining trim Average Memory Usage (AMU) and Average Energy Consumption (AEC) by ≈25% but incur a ∼1-point perplexity penalty. *RWKV* achieved the lightest memory footprint in our pretraining tests, whereas *Pythia* yielded the fastest latency, albeit at the cost of higher perplexity. 4) *Depth–width aspect ratio has a flat optimum in pretraining*: Confirming the robustness of Chinchilla's scaling laws for aspect ratios during pretraining, our depth–width sweeps show a *flat basin* where configurations within ±20% of the Chinchilla-optimal aspect ratio reach statistically indistinguishable loss levels. This allows flexibility for hardware-aligned architectural tailoring without sacrificing performance.

**Training & Tuning Efficiency.** We benchmarked full fine-tuning against five Parameter-Efficient Fine-Tuning (PEFT) methods. Our findings include, as shown in Figure 4: 1) *Optimal PEFT method varies with scale*: For 1–3B models, our results show **LoRA-plus** (LoRA and its variants like DoRA) achieves the lowest loss under a 60 GB AMU constraint. For models above 14B parameters, **RSLoRA** dominates on both loss and latency metrics. 2) *Parameter freezing offers lowest latency*: We measured that *parameter freezing* slashes fine-tuning latency by 3× compared to any PEFT variant tested, making it suitable for interactive fine-tuning scenarios where a slight decrease in average task performance (e.g., approximately 1-2 points on relevant benchmarks, though task-dependent) is acceptable. 3) *Full fine-tuning shows diminishing returns at scale*: Our experiments indicate that full fine-tuning of models larger than 24B parameters yields diminishing returns, with loss improvements often less than 0.02 even as energy consumption doubles. This strongly argues for adopting PEFT methods for large-scale model adaptation. 4) *DoRA latency trade-off*: While *DoRA* maintained stable loss during fine-tuning in our tests, it incurred significant latency overhead, making it more suitable for batch-oriented fine-tuning pipelines rather than real-time or latency-sensitive deployment scenarios.

**Inference Efficiency.** Inference efficiency governs the cost and feasibility of model deployment. Our benchmark provides the following insights, as shown in Figure 5: 1) *Quantization yields high compression with minor score impact*: Our results show that *Int4 post-training quantization* reduces memory footprint and throughput (tokens/s) by up to 3.9× across LLaMA-3, DeepSeek, and Qwen model families (1.5B to 34B parameters), with a moderate 3–5 percentage point drop in average-task scores. 2) *BF16 preferred over FP16 on modern GPUs*: Between floating-point formats, our

measurements on GH200/H200 GPUs consistently show **bfloat16** beating float16 by $\approx 6\%$ in latency and $\approx 9\%$ in energy consumption, benefiting from native hardware acceleration.

# B BACKGROUND

## B.1 LARGE LANGUAGE MODELS (LLMs)

Large Language Models (LLMs) represent a revolutionary technology in the field of artificial intelligence. Essentially, these models are complex neural networks based on the Transformer architecture, which, through deep learning from vast textual corpora, can capture and replicate the intricate details of human language. The core architecture of these models relies on the Self-Attention mechanism, enabling them to process input sequences in parallel and effectively capture long-range dependencies and contextual relationships within language. Compared to traditional recurrent neural networks, LLMs demonstrate significant advantages in language understanding and generation tasks. Since the introduction of the Transformer model, the processing power of language models has grown exponentially, evolving from a few million parameters to today's models with hundreds of billions or even trillions of parameters.

Throughout the development of these models, milestones such as the GPT (Generative Pre-trained Transformer) series (Radford et al., 2018; 2019; Brown et al., 2020), BERT (Devlin et al., 2019), and subsequent variants like RoBERTa (Liu et al., 2019) and ALBERT (Lan et al., 2019b) have been key drivers of LLM advancements. These models have achieved breakthrough progress in areas such as machine translation, text summarization, question answering systems, and code generation through various pre-training strategies and architectural innovations. Notably, ultra-large models such as GPT-3 and GPT-4, through few-shot (Brown et al., 2020) and zero-shot (Kojima et al., 2022) learning, are capable of handling nearly any natural language task, demonstrating impressive potential for general artificial intelligence. These models not only understand and generate natural language but also perform complex reasoning, creation, and problem-solving tasks.

The applications of large language models are extremely broad and have permeated nearly every digital interaction domain. In business services, they can provide intelligent customer service (Pandya & Holia, 2023), automatic content generation (Xu et al., 2024c; Xiang et al., 2024), and personalized recommendations (Mohanty, 2023; Fan et al., 2023; Xu et al., 2024c); in education, they enable personalized tutoring, intelligent question bank generation, and study assistance (Li et al., 2023b; Wang et al., 2024b; Zhang et al., 2024d); in research and development, they assist with code generation (Jiang et al., 2024b; DeLorenzo et al., 2024), academic writing (Liang et al., 2024), and literature reviews (Agarwal et al., 2024). More importantly, these models are reshaping human-machine interactions (Xu et al., 2024a), making communication with AI more natural, intelligent, and efficient. From programming assistance to creative writing, from language translation to complex problem-solving, LLMs are becoming universal intelligent tools across various fields.

However, LLMs also face significant efficiency challenges (Wan et al., 2023; Bai et al., 2024; Zhou et al., 2024b; Li et al., 2023d). These models typically contain billions to trillions of parameters, with training and inference processes requiring massive computational resources and energy. For example, the training cost of GPT-3 can reach millions of dollars, and the computational expense of a single inference is also considerable (Brown et al., 2020). Moreover, the deployment and fine-tuning of large models place high demands on hardware infrastructure, limiting their application in resource-constrained environments. As a result, more researches are focusing on model compression, knowledge distillation, and efficient fine-tuning techniques, aimed at reducing computational complexity and improving the practical utility and accessibility of these models. Additionally, issues such as bias control, privacy protection, and ethical use of models have become important topics of shared concern in both academia and industry.

## B.2 APPROACHES TO ENHANCING EFFICIENCY IN LLMs

### B.2.1 HARDWARE INNOVATIONS

Modern AI-specific accelerators are central to handling the immense compute demands of Large Foundation Generative Models. While GPUs remain the workhorse for LLMs with their massively parallel SIMD/SIMT design, specialized chips like Google's TPUs, Amazon's Trainium/Inferentia, and Intel's Gaudi offer tailored architectures that often lower power consumption and cost per operation (Park et al., 2024). These accelerators typically use systolic arrays to speed up matrix multiplications (critical for transformers) and integrate High-Bandwidth Memory (HBM) to feed

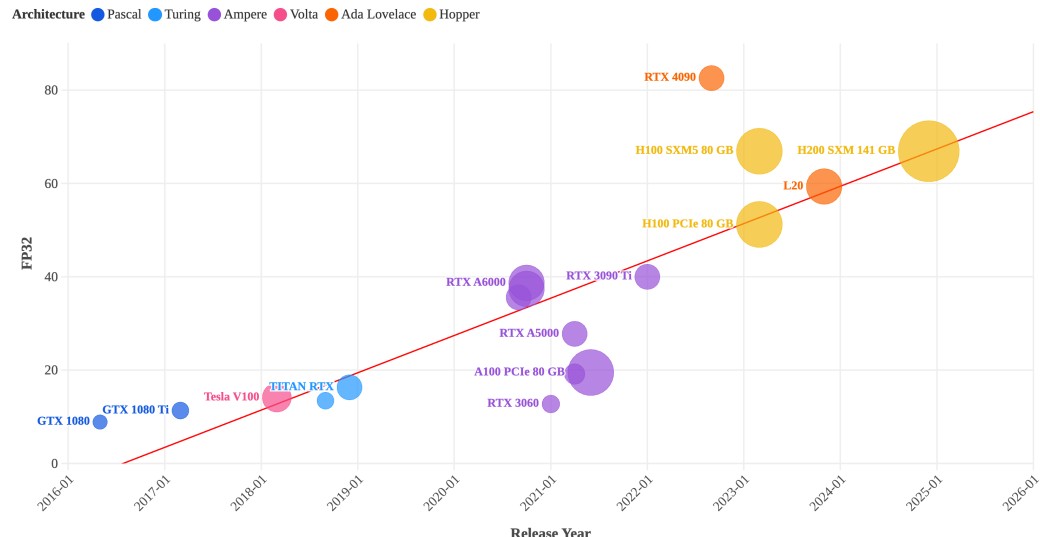

Figure 6: The development trends of computational efficiency and memory capacity across NVIDIA GPU series. Note that different colored dots represent different architectures, and the red line indicates the fitted trend of computational efficiency over time.

data at extreme rates (Park et al., 2024). HBM provides much higher memory bandwidth than traditional DDR memory, alleviating data transfer bottlenecks for large models. However, HBM's on-chip capacity is limited, requiring careful memory management so that model weights and activations are shuttled efficiently without exceeding the cache-like HBM storage. Innovations in interconnects (such as NVIDIA's NVLink and NVSwitch) further improve multi-GPU bandwidth, allowing faster model parallel communication (Park et al., 2024). Overall, the co-design of custom ASICs and memory/network fabric has significantly improved throughput and scalability for training and inference of LLMs.

Beyond raw throughput, energy efficiency has become a paramount hardware consideration for large models. Data-center AI workloads consume vast power, so modern accelerators emphasize performance per watt. For instance, TPU and similar ASICs achieve higher ops/Joule on transformer tasks than general GPUs by streamlining their circuitry for dense linear algebra (Zhu et al., 2025b; Jouppi et al., 2017). Alongside digital optimizations (like lower-voltage operations and mixed-precision arithmetic), there is exploration of fundamentally new computing paradigms. Neuromorphic computing chips, which mimic brain neurons and operate via sparse spiking signals, promise orders-of-magnitude efficiency gains. By co-locating memory and compute and leveraging event-driven operation, neuromorphic processors could execute large neural networks with $100\times-1000\times$ less energy (Saeidi et al., 2022). Similarly, photonic computing is emerging as a futuristic option: optical neural network accelerators can perform matrix operations with light instead of electricity, offering extremely high parallelism with low heat dissipation. Recent prototypes of photonic processors have demonstrated over $100-$fold improvements in energy efficiency and $25\times$ higher compute density compared to conventional electronics (De Lima et al., 2019). While still in early stages, these neuromorphic and photonic approaches represent promising paths for future efficiency gains once today's silicon-based architectures hit their limits (Duan et al., 2024).

### B.2.2 SOFTWARE OPTIMIZATIONS

Efficient software frameworks and parallelization strategies are crucial to fully utilize hardware for LLMs. Distributed computing techniques enable splitting giant models and workloads across many devices in parallel (Verbraeken et al., 2020). For training, this often means hybrid parallelism: data parallelism to copy the model across nodes for different data batches, combined with model/tensor parallelism to split the model's layers or tensor operations among accelerators. For example, GPU clusters running libraries like DeepSpeed (Rasley et al., 2020) or Megatron-LM (Shoeybi et al., 2019b)

orchestrate tensor sharding, pipeline parallelism (partitioning layers into stages), and optimizer state sharding to overcome memory limits (Duan et al., 2024; Park et al., 2024). Such coordination is non-trivial—LLMs with hundreds of billions of parameters do not fit on a single device, so software must partition the model and manage inter-GPU communication efficiently. Advances in collective communication (e.g. using high-speed interconnects or custom protocols) and load balancing ensure that distributed training scales with minimal overhead. In short, sophisticated parallel runtime systems hide the complexity of multi-node training, achieving near-linear speedups and making tractable the otherwise prohibitive training times (often running for weeks over thousands of GPUs).

We compare several popular LLM and VLM frameworks across their support for pre-training, fine-tuning, and inference. Notably, frameworks such as *Colossal-AI*, *Composer*, *DeepSpeed*, *FairScale*, and *Megatron* support all three stages, including large-scale pre-training. In contrast, *LLM Foundry* and *OpenLLM* focus primarily on fine-tuning and inference, while tools like *RayLLM*, *vLLM*, and *Text Generation Inference* are optimized for efficient serving only. A full comparison is provided in Appendix H.2.

Another major avenue is model compression and efficient fine-tuning techniques that reduce the memory and compute footprint of large models (Cheng et al., 2017; Polino et al., 2018). Quantization has become a standard approach: model weights and activations are converted from 32-bit floats to lower precision (e.g. 8-bit integers) to save memory and accelerate tensor operations. By sacrificing a small amount of accuracy, INT8 or even INT4 quantization can dramatically improve inference throughput – for instance, 8-bit weight quantization yielded $1.5\times$ speedup on transformer inference with only $\approx$2–3% accuracy loss in one study. Pruning techniques remove redundant parameters or structures from the network to slim down model size. By identifying neurons, attention heads, or weights that contribute little to outputs, pruning can maintain model quality while cutting down FLOPs. Structured pruning (dropping whole units or layers) tends to yield actual speedups on hardware, whereas unstructured pruning creates sparse weights that may need specialized hardware to exploit (Zhu et al., 2024a; Wan et al., 2023). These methods are challenging for LLMs (aggressive pruning can degrade accuracy), but recent research on magnitude-based and optimal brain surgeon pruning has made progress in sparsifying large transformers without severe performance loss. In the training regime, low-rank adaptation has emerged as an efficient fine-tuning strategy: instead of updating all $N$ billion parameters of a model for a new task, one can insert small low-rank weight matrices and train only those. LoRA is a prime example that freezes the original model weights and learns a limited number of new parameters per layer. This approach yielded over $10,000\times$ reduction in trainable parameters (and $3\times$ lower VRAM usage) when adapting GPT-3, yet achieved on-par accuracy to full fine-tuning. Techniques like LoRA (Hu et al., 2021a) thus enable personalizing or specializing LLMs without the exorbitant cost of retraining the entire network.

At the systems level, compiler optimizations and specialized kernels greatly improve the runtime efficiency of model execution. Deep learning compilers (XLA, TVM, PyTorch Glow/Inductor, etc.) take high-level model graphs and generate low-level code that maximizes hardware utilization. They apply optimizations such as operator fusion (merging multiple neural network operations into one kernel launch), loop tiling and memory layout optimization (to exploit caches or shared memory on GPUs), and vectorization. For example, combining the operations of attention computation (matrix multiplication + softmax) into a fused kernel can avoid intermediate memory writes and improve speed. A notable optimized kernel is FlashAttention series (Dao et al., 2022a; Dao, 2023), which reimplements the attention mechanism in a tile-by-tile fashion to use on-chip memory efficiently, thereby reducing memory bandwidth usage and enabling larger sequence lengths with lower latency. Similarly, libraries provide hand-tuned or auto-tuned kernels for transformer building blocks (dense layers, layer normalization, convolution in vision models) that exploit the specific accelerator's capabilities (Tensor Cores, etc.). These low-level improvements often yield significant gains: for instance, using an optimized attention kernel or a JIT-compiled fused operation can improve throughput by $> 2\times$ compared to naive implementations (Snider & Liang, 2023). The use of graph compilers also allows automatic exploration of different execution plans (such as finding the best parallelization or memory trade-off) and can adapt models to new hardware with minimal manual code rewriting. Overall, the compiler and kernel-level innovations ensure that the theoretical speedups of advanced hardware are actually realized when running large models at scale.

Finally, knowledge distillation (Hinton et al., 2015) and retrieval-augmented generation (Lewis et al., 2020) are high-level software strategies to make large models more efficient in practice. Knowledge distillation involves training a smaller "student" model to replicate the behavior of a large "teacher"

model, effectively compressing knowledge into a compact network (Hinton et al., 2015; Gou et al., 2021). This has been used to create lightweight versions of giant models (e.g., DistilBERT (Sanh et al., 2019b) is a distilled 66M parameter version of BERT (Devlin et al., 2019) that retains most of its accuracy). Distillation can significantly reduce model size and inference cost, though careful training is required to preserve quality on diverse tasks (Gou et al., 2021). Retrieval-Augmented Generation (RAG) techniques, on the other hand, aim to reduce the burden on the model's parameters by offloading some knowledge to an external database. In this approach, an LLM is coupled with a retrieval system that fetches relevant documents or facts from a large corpus, which the model then conditions on during generation. This allows even a smaller model to produce informed, accurate outputs by leveraging information beyond its fixed weights. For example, the RETRO (Borgeaud et al., 2022) model by DeepMind augments a 7.5B parameter transformer with a text chunk database and retrieval mechanism; remarkably, RETRO (Borgeaud et al., 2022) with 7.5B parameters outperformed a 175B parameter GPT-3-style model Jurassic-1 (Lieber et al., 2021) on multiple language benchmarks by virtue of accessing a rich external knowledge base (Borgeaud et al., 2022). This result underscores how retrieval can substitute for brute-force parametric knowledge, attaining the accuracy of a model over 20× larger. By marrying generation with search, RAG methods improve factual accuracy and efficiency, since the model doesn't need to internalize every piece of world knowledge (Li et al., 2024; Xiong et al., 2024). Such techniques, alongside modular and memory-augmented model designs, highlight a trend of leveraging external resources and smarter training schemes to curb the resource requirements of foundation models without sacrificing capability.

### B.2.3  ALGORITHMIC IMPROVEMENTS

At the algorithm level, researchers have proposed numerous Transformer architecture refinements to boost efficiency for LLMs, LVMs, and multimodal models. One direction is sparse attention mechanisms, which limit the quadratic cost of attending to every token (Child et al., 2019b). Sparse Transformers (Child et al., 2019b) introduce structured patterns in the attention matrix (e.g. attending only locally or to a subset of tokens) to bring complexity down from $O(n^2)$ to sub-quadratic or linear in sequence length. This enables handling longer sequences or higher resolutions with the same compute budget. Models like Longformer (Beltagy et al., 2020b), BigBird (Zaheer et al., 2020b), and Reformer (Kitaev et al., 2021) use block-local attention or hashing-based mixing to achieve this kind of efficiency, essentially skipping computation for many token pairs with negligible impact on accuracy. Another powerful idea is the Mixture-of-Experts (MoE) (Jiang et al., 2024a) architecture, which increases model capacity by having multiple expert subnetworks and routing each input token through only one or a few of them (Masoudnia & Ebrahimpour, 2014). In a transformer MoE layer, different "experts" (sets of feed-forward parameters) specialize on different tokens, and a gating function selects which expert to activate per token (making the computation sparse). This allows an MoE model to have a very large number of parameters in total, but each inference/pass only uses a fraction of them. MoE transformers (e.g. Switch Transformers (Fedus et al., 2022)) have been shown to achieve comparable or higher accuracy than dense models with the same effective compute. In fact, MoEs can be pre-trained substantially faster than dense models of equivalent size, and they yield faster inference throughput for a given budget of floating-point operations (Zhang et al., 2024c; Lin et al., 2024). The trade-off is that maintaining many experts demands more memory and introduces complexity in load-balancing the experts' utilization. Nonetheless, MoEs represent a promising efficiency leap: Google's Switch-C Transformer (Fedus et al., 2022) (with 1.6T parameters across experts) demonstrated that vastly larger sparse models can be trained at the same cost as a much smaller dense model, leveraging only modest accuracy trade-offs. Other architecture tweaks include linear or low-rank attention mechanisms that approximate the attention computation with kernel feature maps (as in the Performer and Linear Transformer models), reducing memory usage by avoiding explicit $n \times n$ attention matrices. Such linear attention variants scale as $O(n \cdot d)$ and can be parallelized to outperform standard attention for long sequences, though maintaining accuracy remains an area of active research. In the vision domain, analogous ideas like token pruning/merging in Vision Transformers (reducing the number of patches processed) also improve efficiency. In summary, by re-imagining the transformer's core operations – whether through sparsity, factorization, or conditional computation – these architectural innovations enable handling larger inputs or models at lower computational cost, albeit sometimes with added system complexity.

Improving the training process itself is another important angle for efficiency. Curriculum learning strategies have been revisited for large models to speed up and stabilize training. The idea, dating

back to Bengio et al., is to present easier examples or sub-tasks first and gradually increase difficulty, so that the model learns faster (much like humans learning concepts in a logical order). For instance, an LLM could first be trained on shorter or simpler text sequences before introducing very long and complex documents, allowing it to build a strong foundation and converge in fewer steps than if all data were seen randomly. Another approach is progressive stacking (layer growth), where one starts training a smaller model and then incrementally increases its depth/size using knowledge from the smaller model. Gong et al. demonstrated this with BERT: they first trained a shallow $L$-layer model, then "grew" it to $2L$ layers by duplicating the learned layers, and continued training – the larger model converged much faster than training from scratch with 2L layers (Devlin et al., 2019). This form of warm-start leverages the learned weights of a simpler model to initialize a bigger model, effectively bootstrapping the training of deep networks. Progressive stacking and related model growth techniques (like gradually increasing the sequence length or model width during training) can find an efficient path through the training landscape, saving time and compute. Moreover, techniques like curriculum in data selection (ordering training data by quality or complexity) or gradual unfreezing (fine-tuning large models by slowly relaxing which layers are trainable) act as implicit regularizers, often reaching better optima with less data or compute. While these methods introduce additional scheduling heuristics to the training pipeline, they have shown tangible efficiency improvements in practice by converging to high performance with fewer updates.

Data efficiency is also a crucial aspect – making the most out of the data that models see. Innovations in tokenization help reduce wasted computation on overly long sequences or irrelevant tokens. Subword segmentation algorithms (BPE , WordPiece, SentencePiece) have evolved to produce more efficient vocabularies that balance vocabulary size and sequence length. A good tokenizer can significantly shorten the input sequence (e.g., by merging frequent word pieces or handling multi-byte characters effectively), thereby reducing the number of transformer steps required. For instance, modern byte-level BPE tokenizers can represent text with fewer tokens than character-level methods, especially for languages with many compound words, directly improving model throughput. In multimodal models, analogous token or patch optimizations (such as merging similar image patches or using lower resolution early in processing) also yield efficiency gains. Beyond tokenization, self-supervised learning paradigms greatly enhance data efficiency by leveraging unlabeled data at scale. Rather than relying on limited human-annotated examples, large models are pretrained on raw text or images via predictive tasks (next word prediction, masked token recovery, image-text alignment, etc.), which effectively turn vast unsupervised corpora into training signal. This has enabled data scaling laws, where more data can substitute for bigger models. Notably, recent research on compute-optimal model scaling found that many earlier LLMs were substantially under-trained on data for their size. DeepMind's Chinchilla project showed that a 70B parameter model trained on 1.4 trillion tokens ($4\times$ more data than similarly sized Gopher) outperformed a 175B model (GPT-3) that had less training data, all while using the same training compute budget (Hoffmann et al., 2022a). This result underlines the importance of feeding models with sufficient and high-quality data: a smaller but properly trained model can be more powerful and efficient than a larger, under-trained one. The takeaway is that there is an optimal balance between model size and dataset size for a given compute budget. By following such scaling law insights, one can achieve better performance per compute by right-sizing the model and dataset. In practice, techniques like data filtering and deduplication (to ensure the model isn't wasting capacity on corrupt or repetitive examples), as well as smarter data augmentation, also help models reach higher accuracy faster. In multimodal settings, leveraging pre-trained unimodal models (like vision or language models) as a starting point for combined tasks is another data-efficient strategy, effectively reusing knowledge. All these approaches focus on extracting maximum learning from each sample the model sees, which is crucial when pushing the limits of model scale without an explosion in required data.

Finally, advances in optimization algorithms have played a key role in efficient large-model training. Traditional stochastic gradient descent has largely been supplanted by adaptive optimizers like Adam (Kingma & Ba, 2014), Adagrad (Duchi et al., 2011), LAMB (You et al., 2019), etc., especially for huge models. These methods adapt the learning rate for each parameter based on past gradients, enabling more stable and faster convergence in very high-dimensional parameter spaces. For example, the Adam optimizer was pivotal for training transformers and is used almost universally for LLMs because it handles sparse gradients and varying feature scales automatically. The LAMB optimizer extended this to support extremely large batch training – in one case, allowing BERT (Devlin et al., 2019) pre-training to scale to a batch size of 32k without loss of accuracy, thereby reducing the training time from 3 days to only 76 minutes on a TPU pod (You et al., 2019). Such adaptive schemes

Table 1: Specifications of various NVIDIA GPUs.

| GPU | Release Year | Mem (GB) | Transistors (M) | Architecture | FP32 (TFLOPS) | FP16 (TFLOPS) | PSU (W) |
|---|---|---|---|---|---|---|---|
| GTX 1080 | 2016-5 | 8 | 7200 | Pascal | 8.87 | – | 450 |
| GTX 1080 Ti | 2017-3 | 11 | 11800 | Pascal | 11.34 | – | 600 |
| RTX 2080 Ti | 2018-9 | 11 | 18600 | Turing | 13.45 | 26.9 | 600 |
| TITAN RTX | 2018-12 | 24 | 18600 | Turing | 16.31 | 32.62 | 600 |
| RTX 3090 | 2020-9 | 24 | 28300 | Ampere | 35.58 | 35.58 | 750 |
| RTX 3090 Ti | 2022-1 | 24 | 28300 | Ampere | 40 | 40 | 850 |
| RTX 3060 | 2021-1 | 12 | 12000 | Ampere | 12.74 | 12.74 | 450 |
| Tesla V100 | 2018-3 | 32 | 21100 | Volta | 14.13 | 28.26 | 600 |
| RTX A4000 | 2021-4 | 16 | 17400 | Ampere | 19.17 | 19.17 | 300 |
| RTX A5000 | 2021-4 | 24 | 28300 | Ampere | 27.77 | 27.77 | 550 |
| RTX A6000 | 2020-10 | 48 | 28300 | Ampere | 38.71 | 38.71 | 700 |
| A40 PCIE | 2020-10 | 48 | 28300 | Ampere | 37.42 | 37.42 | 700 |
| A100 PCIe 80 GB | 2021-6 | 80 | 54200 | Ampere | 19.49 | 77.97 | 700 |
| L20 | 2023-11 | 48 | 76300 | Ada Lovelace | 59.35 | 59.35 | 600 |
| RTX 4090 | 2022-9 | 24 | 76300 | Ada Lovelace | 82.58 | 82.58 | 850 |
| H100 PCIe 80 GB | 2023-3 | 80 | 80000 | Hopper | 51.22 | 204 | 750 |
| H100 SXM5 80 GB | 2023-3 | 80 | 80000 | Hopper | 66.91 | 267.6 | 1100 |
| H200 SXM 141 GB | 2024-12 | 141 | 80000 | Hopper | 66.91 | 267.6 | 1100 |

make it feasible to utilize parallel hardware (by increasing batch sizes) efficiently while maintaining training stability. In addition to optimizers, there is growing interest in reinforcement learning and automated tuning to squeeze out further efficiency. One example is using RL or other automated methods to tune hyperparameters (learning rates, batch schedules) or even architectural choices during training. As an illustration, the Zeus system dynamically adjusts the GPU power limit and batch size during training to improve energy efficiency without degrading training time (Jomaa et al., 2019; You et al., 2023; Zahavy et al., 2020). By formulating the trade-off between power usage and throughput as an optimization problem, techniques like this can save significant energy in large-scale training runs. More broadly, Neural Architecture Search (NAS), often powered by reinforcement learning or evolutionary algorithms, has been used to discover efficient neural network architectures automatically (Ren et al., 2021). While NAS has mostly been applied to smaller-scale image or language models, the concept extends to LLMs – for instance, using RL-based agents to decide layer widths, depths, or sparsity patterns could yield architectures that outperform human-designed transformers in efficiency. Already, NAS has produced models like EfficientNet (Tan & Le, 2019) in vision by finding better layer shapes for a given computation budget. We can envision future foundation models being partially discovered by AI themselves, optimized from the ground up for hardware friendliness. Lastly, reinforcement learning also comes into play in fine-tuning large models via methods like proximal policy optimization in the context of Reinforcement Learning from Human Feedback (RLHF), which, while aimed at alignment and not purely efficiency, does demonstrate the flexibility of training algorithms for these models (Havrilla et al., 2024; Zhong et al., 2024; Zhu et al., 2023). In sum, a combination of clever optimizer choices, automated tuning of hyperparameters, and even learning-driven architecture optimization contributes to making the training and deployment of large-scale models more efficient than ever before. Each of these algorithmic improvements – from better optimizers to learning curricula – chips away at the overall resource requirements, enabling the continued scaling of LLMs, LVMs, and VLMs within practical limits.

## C    TECHNIQUES FOR IMPROVING LLM EFFICIENCY

This section is organized as follows. Section 2 introduces background concepts: the LLM fundamentals (Transformer architectures, training paradigms) and common efficiency evaluation metrics. In Section 3, we discuss **budget efficiency** through the lens of scaling laws – how performance scales with compute, model size, and data, and what trade-offs are optimal. Section 4 covers **data efficiency** techniques, including data filtering and curriculum learning to get the most out of training data. Section 5 surveys **architecture-level** innovations such as efficient attention mechanisms, positional encodings, and sparse or attention-free models that reduce the computation per token. Section 6 examines **training and tuning efficiency**, from distributed training and mixed precision to parameter-efficient fine-tuning methods. Section 7 reviews **inference efficiency** via model compression (pruning, distillation, quantization, etc.), decoding optimizations, and systems design for serving LLMs.

### C.1    DIMENSIONS OF LLM EFFICIENCY

When we discuss making LLMs more efficient, it is important to define metrics for **resource usage**:

**Model Size & Parameters:** The number of parameters (and by extension the model file size) is a basic metric. It correlates with memory requirements for storage and inference. For instance, a 175B parameter model in fp16 occupies $\sim$350 GB (2 bytes/param) in memory, whereas a 6B parameter model would be $\sim$12 GB. Parameter count alone is not a perfect proxy for *speed*, but it is a rough measure of model complexity and hardware footprint.

**FLOPs / Computational Cost:** Floating point operations (FLOPs) needed for a forward (and backward) pass measure computational workload. For example, generating one token with GPT-3 requires on the order of $2 \times 175B \approx 3.5 \times 10^{11}$ FLOPs (since each token involves matrix multiplications proportional to model size) (Brown et al., 2020). Training cost can be reported in PF-days (petaflop/s-days) – GPT-3's training was about 3,640 PF-days (Brown et al., 2020). Efficiency improvements often aim to reduce FLOPs needed for the same task or shift to lower-precision operations. Reducing FLOPs generally translates to faster runtime if hardware is fully utilized.

**Throughput and Latency:** Throughput is how many tokens (or sequences) can be processed per second, and latency is how long it takes to get a result. For training, throughput might be measured in examples or tokens per second. For inference, latency per token or per query is key. Techniques like model parallelism might increase throughput but could also introduce communication overhead that affects latency. Real-time applications care about latency (e.g. respond in under 100ms), while batch processing cares about total throughput.

**Memory Footprint:** This includes model weights memory, activation memory during computation, optimizer states during training, and memory for caches. Memory is a limiting factor for deploying large models – e.g., fitting a model on a single GPU requires it to have enough VRAM for the model and intermediate activations. Memory-saving techniques (like gradient checkpointing or quantization) allow using less memory at the cost of extra computation or slight accuracy loss. Efficient memory use is also important to avoid waste when serving many requests (see *PagedAttention* in Section 7, which tackles memory fragmentation (Kwon et al., 2023)).

**Energy and Carbon Efficiency:** Increasingly, researchers track the energy consumed by model training/inference and the associated $CO_2$ emissions (Ding et al., 2023a). A model that achieves the same accuracy with half the energy is more efficient in a very tangible sense. Metrics like "FLOPs per watt" or total kWh for training are used. Strubell et al. (Strubell et al., 2019a) famously highlighted that large NLP models can emit as much $CO_2$ as several cars' lifetimes. Efficiency methods can dramatically cut down energy usage (e.g., by requiring fewer FLOPs or using specialized hardware better). Reporting carbon impact is becoming a good practice.

In practice, efficiency gains may trade off between these metrics. For instance, a method might reduce memory usage at the cost of more FLOPs, or vice versa. Ultimately, *end-to-end* improvements (e.g., reducing overall runtime on a given hardware budget for a given task) are what matter. Throughout this survey, we will note how each technique impacts these metrics. For example, mixture-of-experts models have more parameters but can reduce FLOPs per token by activating only some experts, improving speed at the cost of memory. Quantization reduces memory and may even speed up compute on certain hardware (taking advantage of INT8 tensor cores), with some impact on accuracy.

The goal is Pareto-improvement: achieve the same or better model quality for lower cost on one or more of these axes.

## C.2 BUDGET EFFICIENCY: SCALING LAWS

### C.2.1 SCALING BEHAVIOR AND POWER LAWS

A natural question in the development of LLMs is how performance improves as we allocate more resources. **Scaling laws** refer to empirical relationships between model performance (often measured via cross-entropy loss or perplexity) and scale factors like model size, dataset size, or compute. Pioneering work by Kaplan et al. (2020) observed that the loss $L$ of a language model follows a power-law decline as model parameters $N$ increase: $L(N) \approx aN^{-\alpha} + L_\infty$, for some constants $a, \alpha, L_\infty$ (Kaplan et al., 2020a). Similarly, loss scales as a power-law with the amount of training data $D$. These scaling laws held impressively over *seven orders of magnitude* in $N$ and $D$ (Kaplan et al., 2020a). Crucially, Kaplan et al. found that within the ranges tested, other architectural details (e.g. width vs. depth of layers) had *minimal effect* on loss compared to total parameter count (Kaplan et al., 2020a). In other words, a Transformer's performance is largely a function of how big it is and how much data it is trained on, and the improvement is predictable and smooth (a log-linear trend on plots). This provided a guidepost for building better LLMs: just make them bigger and train on more data, and you will likely get better results.

However, scaling up is not free – it comes with an increased compute budget requirement. Given a fixed compute budget $C$ (which roughly scales as $N \times D$ for training a model of size $N$ on $D$ tokens), how should one allocate it? Kaplan et al. suggested an answer: *larger models are more sample-efficient* (Kaplan et al., 2020a). They found that to minimize loss for a given $C$, one should train a very large model *without fully consuming the data*, rather than a smaller model to convergence (Kaplan et al., 2020a). Intuitively, doubling model size and halving training steps led to lower loss than vice versa. This recommendation – train huge models for fewer epochs – was adopted in early LLMs. For example, GPT-3 was somewhat under-trained (trained on 300B tokens, which is only $\sim$2 epochs over its $\sim$160B token dataset) according to these heuristics.

### C.2.2 COMPUTE-OPTIMAL MODEL SCALING (CHINCHILLA VS. GOPHER)

In 2022, Hoffmann et al. (DeepMind) revisited scaling laws with extended experiments and found that many recent LLMs were *significantly under-trained* for their size (Hoffmann et al., 2022b). They introduced the notion of a **compute-optimal** model: for a given compute $C$, there is an optimal pair of $N$ (model size) and $D$ (tokens) that yields the best performance. Their empirical analysis suggested a roughly *linear relationship* between optimal $N$ and $D$ – in fact, *doubling* the model size should go along with *doubling* the training data to stay on the compute-optimal frontier (Hoffmann et al., 2022b). This is in contrast to the earlier strategy of extremely large $N$ with limited data.

To validate this, Hoffmann et al. trained *Chinchilla*, a 70B parameter model, on 1.4 trillion tokens, using the same compute as used to train *Gopher*, a 280B model on $\sim$300B tokens. The result was striking: Chinchilla (70B) *outperformed* Gopher (280B) on a wide range of downstream tasks (Hoffmann et al., 2022b). Despite having 4$\times$ fewer parameters, Chinchilla's extra training data gave it an edge – for example, it achieved an average score of 67.5% on the MMLU benchmark, >7% higher than Gopher (Hoffmann et al., 2022b). It also surpassed other models in that compute class like GPT-3 (175B) and Megatron-Turing NLG (530B) (Hoffmann et al., 2022b). This revelation prompted a re-evaluation in the community: *bigger is not always better*, if not fed with enough data. A smaller model can "soak up" more data and end up better. Moreover, an added benefit is that Chinchilla-like models are cheaper to fine-tune and faster to inference (since they have fewer parameters) for the same performance level (Hoffmann et al., 2022b).

The concept of *compute-optimal scaling* can be summarized by the heuristic: *scale model size and data in tandem*. One way to express the optimal regime is to set $N$ proportional to $D$ (assuming training compute $C \propto N \cdot D$ for a given architecture). Under a fixed $C$, this yields an optimal $N^*$ and $D^*$. The Chinchilla law suggests $N^* : D^*$ should be about 20 tokens per parameter (in the 2022 study) – though that exact ratio may vary. The key is that many previous models like GPT-3 (which had $\sim$2 tokens per parameter) were far off this optimum, hence under-utilizing data. With this insight, new models (e.g. LLaMA, see below) have aimed to be more balanced.

### C.2.3 DATA CONSTRAINTS AND QUALITY

One practical challenge is that simply scaling both $N$ and $D$ requires massive high-quality datasets. If one is *data-constrained*, scaling laws can bend or break. For instance, if only $10^8$ tokens of domain-specific text exist, making the model larger than a certain point yields diminishing returns because it will quickly saturate the available data (and start overfitting). In such regimes, one might in fact prefer a smaller model or use heavy regularization and reuse data with careful curriculum. Empirically, when *data is the bottleneck*, the performance gains will flatten out no matter how much compute you throw with more parameters (Hoffmann et al., 2022b). This scenario has led researchers to focus on **data quality and curation** – to get more "effective" data for the model to consume. For example, using diverse sources and cleaning duplicates helps avoid wasted capacity on redundant or low-value text.

Interestingly, improvements in *data quality* can sometimes substitute for sheer quantity. The *LLaMA* models by Meta (2023) demonstrated that by curating a high-quality mix of public data and training somewhat past the earlier "optimal" point, a 13B model could outperform GPT-3 175B on most benchmarks (Touvron et al., 2023). LLaMA-13B was trained on 1T tokens (slightly more than Chinchilla's recommended $13B \times 20 = 260B$, so it "over-trained" relative to compute-optimal), yet its performance benefited from the high quality data and possibly better training efficiency. This hints that the *constants* in scaling laws depend on data quality – better data gives lower loss for the same size. Thus, another dimension of efficiency is maximizing what the model learns per token of data. We will cover data filtering techniques in Section 4.

Moreover, when models are scaled up, they often *unlock capabilities* rather than just monotonically improving a single metric. For example, very large models can do multi-step reasoning or understand nuanced instructions (emergent behaviors) that smaller models cannot (Ding et al., 2023a). These binary capabilities (has/has not) complicate the smooth scaling picture. Recent studies (e.g. Wei et al. 2022 on emergent abilities) show that some tasks suddenly become solvable once the model crosses a size threshold (Ding et al., 2023a). Such *breaks in scaling trends* mean that beyond a certain point, scaling might yield *discontinuous* leaps in what the model can do.

### C.2.4 OPEN PROBLEMS IN SCALING

**Broken Scaling and Out-of-Distribution Generalization:** While scaling laws hold remarkably well on the training distribution (and near-range evaluations), they can "break" when extrapolating. For instance, a model might follow a power-law on perplexity but fail to improve on a certain logical reasoning task until it reaches a large size. Understanding these deviations is ongoing work. Some researchers propose *multi-faceted scaling laws* that incorporate additional factors (like knowledge composition or reasoning depth) to predict performance; others have introduced *evaluation scaling laws* to estimate how performance on downstream tasks scales. A challenge is we do not have a complete theory of *why* power-laws emerge; it may relate to the underlying data distribution and model capacity being used effectively. When scaling further (e.g. to trillion-parameter models), will new phenomena occur or will the gains saturate? Recent evidence from models at GPT-4 scale suggests that scaling alone is not enough – for example, GPT-4 likely owes some improvements to architecture and training technique, not just size.

**Architecture Shape (Depth vs Width):** Kaplan et al. noted little effect of depth vs width within reasonable ranges (Kaplan et al., 2020a). Yet, as we push models to extreme depths (hundreds or thousands of layers), training becomes unstable. Techniques like DeepNorm (Wang et al., 2022) allow 1000-layer Transformers by adjusting residual scaling. It remains an open question whether a *very deep narrow model* could outperform a *shallow wide model* of the same parameter count when properly trained. In theory, depth could give more representational power, but optimization issues might negate that. So far, empirical evidence indicates that for equal parameter count, there is a broad plateau of depth-vs-width configurations that perform similarly (Kaplan et al., 2020a). Very extreme aspect ratios (too deep and narrow) underperform due to difficulty in training. Thus, architecture interplay with scaling is subtle – most large LMs keep depth around 40–80 layers and increase width ($d_{\text{model}}$) for larger sizes, a heuristic that has worked.

In summary, scaling laws provide a *north star* for guiding efficient use of a compute budget: use as much data as possible and right-size the model to that data. The era of blindly increasing parameter count is over – instead, we aim for *scaling balanced with data*. The following sections (4–7) can

be seen as methods to improve or refine the scaling curves – achieving on-par performance with fewer parameters (through data or architecture efficiency), or reaching a target performance with less compute (through better training algorithms and inference optimizations).

## C.3 DATA EFFICIENCY

Training an LLM often involves hundreds of billions of tokens of text. Collecting and processing such data is costly in terms of time, storage, and even intellectual property concerns. **Data efficiency** refers to techniques that extract maximum performance from a given amount of data – or alternatively, achieve a target performance with significantly less data. This is crucial when data is limited (e.g. specialized domains) or expensive to curate/label. Two major strategies are *data filtering* to improve quality and *curriculum learning* to optimize the order in which data is used.

### C.3.1 IMPORTANCE OF DATA QUALITY AND FILTERING

Not all data are equal. Web-scale corpora contain duplicates, spam, and low-quality text that can waste training capacity or even harm the model (learning bad facts or biases). Data filtering methods aim to curate a **higher-quality training set** without dramatically reducing its diversity. One straightforward but effective technique is *deduplication* – removing duplicate or near-duplicate examples. Even though web scrapes are huge, they often contain many repeated texts (news articles copied on multiple sites, boilerplate templates, etc.). Deduplicating the dataset can reduce its size and also improve generalization. Lee et al. (2021) showed that deduplication allowed models to reach the same validation loss in $10\times$ fewer steps in some cases (Ding et al., 2023a). Intuitively, the model does not waste time memorizing the exact same content repeatedly. Common approaches use hashing (e.g. MinHashLSH) to identify duplicates efficiently. Projects like CC-Net use clustering and hashing to clean Common Crawl data, while adversarial filtering (Demaine et al., 2019) can remove machine-generated or undesirable text.

Another filtering axis is *data selection / undersampling*. If certain portions of data are less useful, we can sample them less or drop them. For example, when mixing diverse sources (Wikipedia, books, web), one might undersample the largest but lowest-quality source to ensure the model does not get overwhelmed by it. Instance-based *importance sampling* can go further – ranking individual examples by some score of utility. Recent work explores filtering out examples that are too easy or too hard for the model at its current stage. One approach is *loss-based filtering*: if the model (or a smaller proxy model) already assigns very low loss to an example, that example might not teach it much new. Jiang et al. (2019) proposed *Selective Backpropagation*, where they only backpropagate on examples with high loss. This yielded faster convergence by focusing compute on the mistakes the model was making. Similarly, *gradient-based sampling* picks examples with the largest gradient norms, which indicate the example has a big effect on parameters and might be more informative (Ding et al., 2023a). Katharopoulos & Fleuret (2018) developed an importance sampling scheme based on an upper bound of gradient norm (Katharopoulos & Fleuret, 2018), and others have implemented online sample selection using proxy models.

One must be careful that filtering does not overly skew the data distribution. Strategies like random undersampling of over-represented classes (Gruenrock, 2015) have shown that dropping redundant data can both reduce training time and improve balance. For example, if 90% of the data is English and 10% is other languages, one might downsample English data to ensure the model learns multilingual capability (if that is a goal). The *MESA* approach uses meta-learning to learn how to sample effectively from a large dataset, and so forth. The outcome of successful data filtering is a leaner corpus where each example has *value*. This can significantly cut the required number of tokens $D$ to reach a certain performance, which directly translates to less training compute.

### C.3.2 CURRICULUM LEARNING

While the above deals with *which* data to use, **curriculum learning** (Elman, 1993; Bengio et al., 2009) concerns *in what order* to present the data to the model. Inspired by how humans learn (starting from easy concepts and progressing to harder ones), curriculum learning for LLMs means we might begin training on simpler patterns and gradually move to more complex ones (Ding et al., 2023a). The hypothesis is that this guides the model's optimization in a smoother way, potentially leading to better final performance or faster convergence.

A curriculum requires two components: a *difficulty metric* to rank training examples by complexity, and a *pacing function* that determines how to schedule the introduction of harder examples. Common difficulty metrics in NLP include: (a) *sequence length*, (b) *vocabulary rarity*, (c) *perplexity/uncertainty* according to a smaller model. For instance, Zhao et al. (2020) used word rarity as a measure, presuming that handling rare words requires more context and understanding. The pacing function can be step-wise or stage-wise. A neat example of stage-wise curriculum is *Shortformer* (Huang et al., 2020), which trained a Transformer first on short sequences only, then in a second stage allowed long sequences. By doing so, the model first mastered local coherence without being confused by long-range dependencies, and then could leverage that foundation to handle long contexts. In general, curricula can be as simple as sorting the training data by length or complexity and always feeding in that order, or as complex as dynamically adjusting sample difficulty based on current model performance (*self-paced learning* (Kumar et al., 2010)).

Applications of curriculum learning in LLMs have included: training small models on code with gradually increasing code length (Ahmed et al., 2023; Bogomolov et al., 2024; Majdinasab et al., 2024), training multilingual models by starting with one language then adding more (Chang et al., 2023; Faisal & Anastasopoulos, 2024; Ebrahimi & Church, 2024; Nigatu et al., 2023), or starting with syntactically simple sentences then moving to full natural text (Latard et al., 2017; Solovyev et al., 2023; Jain et al., 2024). One must ensure that eventually the model sees the full distribution of data, otherwise it might become overspecialized. Most curricula therefore converge to training on the mixture of all data at some point.

In terms of efficiency, curriculum learning can *accelerate convergence* – the model reaches a given loss or accuracy in fewer steps than without a curriculum. For very large models, curriculum strategies like Shortformer have proven valuable for stability and speed. As models venture into longer contexts (e.g. 10k+ tokens), curricula could be essential to first handle short contexts then extend, otherwise training from scratch on extremely long sequences might be too difficult.

Furthermore, recent works have explored curriculum learning strategies to enhance the reasoning abilities of LLMs through reinforcement learning (RL). Approaches such as DeepSeek-R1 (Author & Author, 2025a) and Kimi k1.5 (Author & Author, 2025b) adopt RL fine-tuning methods that progressively expose the model to tasks of increasing difficulty. In these systems, the training is initiated with simpler reasoning tasks, and as the model performance improves, more challenging tasks are introduced. Additional research has proposed alternative curriculum designs. For example, WISDOM (Author & Author, 2024a) leverages progressive curriculum data synthesis to improve the model's performance on mathematical reasoning tasks. Similarly, LBS3 (Author & Author, 2024b) utilizes curriculum-inspired prompting, guiding the model through a sequence of intermediate sub-problems before addressing the primary task. CurLLM-Reasoner (Author & Author, 2024c) and Logic-RL (Author & Author, 2025c) further illustrate how curricula can be designed to integrate structured reasoning and logical puzzles into the RL framework. Finally, AlphaLLM-CPL (Author & Author, 2024d) introduces a dynamic curriculum adjustment mechanism that combines Monte Carlo Tree Search (MCTS) with curriculum preference learning (CPL) to refine reasoning capabilities progressively.

### C.3.3 DATA AUGMENTATION AND SYNTHETIC DATA

Another approach to data efficiency is *creating more data* in a smart way. Techniques like *back-translation* (in MT) and *self-instruct* (for instruction tuning) use models themselves to generate new training examples. For example, the *Self-Instruct* framework had GPT-3 generate its own instructions and responses to teach itself to follow instructions better. This bootstrap approach greatly reduced the need for human-written prompts. In LLM fine-tuning, one might generate paraphrases of a small dataset to expand it. While augmented data may be of lower quality than real data, if the model can still learn from it, it can help squeeze more out of limited original data. Data augmentation blurs into the territory of *knowledge distillation* (where a model's outputs supervise another), which we revisit in Section 7.

In summary, data efficiency techniques aim to *maximize the knowledge gained per token of training data*. By curating high-quality, diverse corpora (filtering out noise and redundancy) and feeding data in an optimal order, we reduce the total data needed. This directly saves computation and allows smaller-scale training runs to still achieve strong performance. As model training budgets are enormous, even a 10% efficiency gain in data usage can mean millions of dollars saved or the

difference between needing 1B vs 1.1B tokens to reach a milestone. Data efficiency is thus a critical piece of the LLM efficiency puzzle, complementary to architectural and algorithmic innovations.

## C.4    ARCHITECTURE EFFICIENCY

The Transformer architecture, while powerful, has some well-known efficiency bottlenecks – notably the *quadratic complexity* of self-attention with respect to sequence length. Architectural efficiency improvements seek to redesign parts of the model to reduce computation or memory usage per token, *without* losing (much) performance. In this section, we discuss several fronts: efficient attention mechanisms, improved positional encodings, models that leverage sparsity, and even alternatives to attention entirely.

### C.4.1    MOTIVATION: RETHINKING THE TRANSFORMER FOR EFFICIENCY

A standard Transformer processes a sequence of length $L$ with self-attention that scales as $O(L^2 \cdot d)$ and feed-forward layers that scale as $O(L \cdot d^2)$. For very long inputs (e.g. documents of thousands of tokens), attention becomes the dominant cost due to the $L^2$ term. The question is: can we maintain the modeling power of Transformers *while cutting down* the attention cost to linear or near-linear in $L$? At the same time, hardware-aware optimizations ask: can we implement attention in a way that uses memory/cache more efficiently?

### C.4.2    EFFICIENT ATTENTION MECHANISMS

**Sparse and Factorized Attention:** One approach is to restrict the attention computation to a subset of token pairs, making the attention matrix sparse. The *Sparse Transformer* (Child et al., 2019a) did this by attending only to a fixed pattern of positions. *Longformer* (Beltagy et al., 2020a) and *Big Bird* (Zaheer et al., 2020a) introduced combinations of local attention (each token attends to a window of nearby tokens) and global attention (a few tokens attend broadly). Big Bird achieved linear complexity and even proved that such patterns are Turing-complete. Another line is factorizing attention via *low-rank approximation*. *Linformer* (Wang et al., 2020a) hypothesized the $L \times L$ attention matrix has low rank, projecting keys/values to lower dimension. *Nyströmformer* (Xiong et al., 2021) and *Performer* (Choromanski et al., 2020) similarly used approximate or kernel-based approaches to reduce attention to linear or $O(L \log L)$ complexity. *Reformer* (Kitaev et al., 2020) used LSH to group tokens that have similar keys, achieving $O(L \log L)$.

**IO-Aware and Hardware-Friendly Attention:** A complementary angle is to optimize how we implement attention. *FlashAttention* (Dao et al., 2022b) keeps exact full-attention but reorders computation and memory access to minimize reads/writes to slow memory. By computing attention in blocks that fit into on-chip SRAM, it significantly speeds up large context processing. This is an *IO-centric* algorithmic approach. *FlashAttention-2* refines these ideas further. These techniques do not change the Transformer math but yield large speedups in practice by alleviating memory bottlenecks.

**MQA:** Multi-Query Attention (MQA) modifies standard multi-head attention by sharing the key and value projections across all heads while keeping the query projections distinct. In standard multi-head attention, for each head $h$ one computes

$$\text{head}_h = \text{Attention}\big(QW_h^Q, KW_h^K, VW_h^V\big),$$

with the attention function defined as

$$\text{Attention}(Q, K, V) = \text{softmax}\left(\frac{QK^\top}{\sqrt{d_k}}\right)V.$$

In MQA, although the query projection $QW_h^Q$ remains unique to each head, the keys and values are shared among all heads:

$$\text{head}_h = \text{Attention}\big(QW_h^Q, KW^K, VW^V\big).$$

This design reduces both the computational load and memory requirements, particularly during inference, as the key–value cache is computed only once for all heads. MQA thus strikes a balance between full multi-head attention and more extreme sharing schemes.

**Grouped Query Attention (GQA):** Grouped Query Attention (GQA) refines standard multi-head attention by partitioning the query heads into $G$ groups, so that each group shares a single key–value pair. In the standard approach, each head $h$ computes

$$\text{head}_h = \text{Attention}\big(QW_h^Q, KW_h^K, VW_h^V\big).$$

When the total $H$ heads are divided into groups of size $g = H/G$, for any head $h$ in group $i$ the key and value projections become shared:

$$\text{head}_h = \text{Attention}\big(QW_h^Q, KW_i^K, VW_i^V\big).$$

This approach interpolates between full multi-head attention (when $G = H$) and multi-query attention (when $G = 1$), providing a tunable trade-off between expressiveness and efficiency.

**Multi-Head Latent Attention (MLA):** Multi-Head Latent Attention (MLA) addresses the memory bottleneck by compressing the key–value (KV) cache using a low-rank latent representation. Instead of computing full keys and values for each head, the input token $h_t \in \mathbb{R}^d$ is first projected into a lower-dimensional latent vector:

$$c_t^{KV} = h_t W^{DKV}, \quad W^{DKV} \in \mathbb{R}^{d \times d_c}, \quad d_c \ll d.$$

Then, for each head $i$, the full key and value vectors are reconstructed using up-projection matrices:

$$k_t^i = c_t^{KV} W_i^{UK}, \quad v_t^i = c_t^{KV} W_i^{UV}, \quad W_i^{UK}, W_i^{UV} \in \mathbb{R}^{d_c \times d_h}.$$

The query is computed as $q_t^i = h_t W_i^Q$. This factorization dramatically reduces the size of the KV cache, lowering memory usage while preserving the model's capacity. MLA is particularly beneficial during inference, as the compressed latent representation can be cached and the keys and values computed on the fly.

**Native Sparse Attention (NSA):** Native Sparse Attention (NSA) reduces computational burden by decomposing the attention operation into three branches. First, a *compression branch* aggregates sequential tokens into a coarse global summary. Second, a *selection branch* computes importance scores—typically via a softmax over intermediate scores—to select the most relevant token blocks. Third, a *sliding window branch* preserves local context by applying full attention within a fixed window. For each query $q_t$, NSA computes the output as

$$o_t^* = g_t^{\text{cmp}} \, \text{Attn}\big(q_t, \tilde{k}_t^{\text{cmp}}, \tilde{v}_t^{\text{cmp}}\big) + g_t^{\text{slc}} \, \text{Attn}\big(q_t, \tilde{k}_t^{\text{slc}}, \tilde{v}_t^{\text{slc}}\big) + g_t^{\text{win}} \, \text{Attn}\big(q_t, \tilde{k}_t^{\text{win}}, \tilde{v}_t^{\text{win}}\big),$$

where the gating coefficients $g_t^c \in [0, 1]$ (for $c \in \{\text{cmp}, \text{slc}, \text{win}\}$) are learned functions that determine the contribution of each branch based on the context. This hierarchical design is both end-to-end trainable and efficient for long-context scenarios.

**MoBA:** MoBA (Mixture of Block Attention) adapts the standard attention mechanism to process long sequences more efficiently by operating on blocks of tokens rather than on the entire sequence. Given a sequence of $N$ tokens, MoBA first partitions the sequence into $n$ blocks, each of size

$$B = \frac{N}{n},$$

with the $i$-th block defined by the indices

$$I_i = \{(i-1)B + 1, \ldots, iB\}.$$

For each query token $q$, a gating network computes an affinity score $s_i$ for each block $i$ as the inner product between $q$ and a summary representation of the keys in block $i$ (typically, the mean of the keys):

$$s_i = \big\langle q, \text{mean}\big(K[I_i]\big) \big\rangle.$$

A top-$k$ selection is then applied, so that only the $k$ blocks with the highest scores are selected. Formally, a gate value $g_i$ is assigned to each block as

$$g_i = \begin{cases} 1, & \text{if } s_i \text{ is among the top-}k \text{ scores,} \\ 0, & \text{otherwise.} \end{cases}$$

The overall set of indices used for attention is

$$I = \bigcup_{i : g_i = 1} I_i.$$

Finally, the attention is computed over the selected keys and values:

$$\text{MoBA}(q, K, V) = \text{softmax}\left(qK[I]^{\top}\right)V[I].$$

To preserve causality in autoregressive models, MoBA prevents a query token from attending to tokens in future blocks by assigning a score of $-\infty$ (or equivalently, a gate value of 0) to any block that comes after the query. Additionally, within the current block, a causal mask ensures that each token only attends to preceding tokens. This strategy reduces the computational cost by limiting the number of tokens processed per query while dynamically selecting the most relevant blocks, thereby providing an effective trade-off between efficiency and expressiveness without changing the overall parameter count.

Overall, these techniques offer distinct strategies to reduce the memory and computational demands of attention mechanisms while preserving performance, marking significant advances in the efficiency and scalability of LLMs.

### C.4.3 EFFICIENT POSITIONAL ENCODING

The processing of extended sequences poses significant challenges for LLMs. Traditional absolute positional encoding (APE) from the original Transformer architecture Vaswani et al. (2017) proves inadequate for handling lengthy inputs. To overcome this constraint, researchers have developed innovative positional encoding (PE) strategies that effectively accommodate longer sequences through relative positioning Press et al. (2023); Chi et al. (2022; 2023); Li et al. (2023c), rotary embeddings Su et al. (2021); Peng et al. (2023b), randomized encodings Ruoss et al. (2023), or even by eliminating positional encoding entirely Kazemnejad et al. (2023). This section examines cutting-edge developments in positional encoding that enhance model efficiency and capability.

**Addition-Based Relative Positional Encoding Frameworks.** Unlike absolute encoding schemes, relative positional encoding methods track relationships between token pairs rather than assigning fixed positions. Several frameworks employ this approach by incorporating encoded relative positions directly into attention calculations. Notable implementations include T5 Raffel et al. (2020), TISA Wennberg & Henter (2021), and FIRE Li et al. (2023c).

T5 Raffel et al. (2020) implements a bucket-based approach, converting positional differences into scalar bias values through a lookup mechanism. This method facilitates some length extrapolation by assigning identical embeddings to all positions beyond the training distribution, though at the cost of increased computational overhead. TISA Wennberg & Henter (2021) advances this concept by deploying a trainable Gaussian kernel specifically focused on inter-token positional differences.

FIRE Li et al. (2023c), developed by Li et al., introduces progressive interpolation using normalized position indices. This normalization is achieved by dividing the positional difference between tokens by the query token's index (i.e., the larger index in causal attention, $i$, for a query at position $i$ and a key at position $j$, the normalized distance is $(i - j)/i$). This approach not only generalizes but effectively unifies previous relative encoding methods, capable of theoretically recovering both T5's RPE and ALiBi as special cases. Empirical evidence demonstrates FIRE's superior generalization capabilities for extended contexts in language modeling benchmarks. These relative encoding approaches fundamentally enhance model comprehension of token relationships while enabling length extrapolation—critical for processing diverse and intricate sequences.

**Decay-Function Approaches to Relative Positioning.** Another significant innovation involves utilizing decay functions within relative positional encodings to emphasize local context. Systems like ALiBi Press et al. (2023), KERPLE Chi et al. (2022), and Sandwich Chi et al. (2023) employ this methodology to gradually diminish attention as the distance between tokens increases.

ALiBi introduces a fixed linear decay function that helps Transformers generalize to extended sequences by imposing a monotonic decay pattern on attention scores. This enables extrapolation beyond the training length with minimal performance loss by biasing attention towards recent tokens, though the linear penalty means very distant tokens contribute negligibly, implicitly constraining the effective receptive field. While this enhances length extrapolation, ALiBi can potentially affect performance on in-distribution data.

KERPLE Chi et al. (2022) refines this approach based on kernel theory, introducing trainable decay RPE with two variants of conditionally positive definite (CPD) kernels: logarithmic and power variants. These sophisticated kernels, featuring learnable parameters per head, adaptively modulate the connection strength between token pairs during RPE computation, achieving excellent extrapolation.

Sandwich Chi et al. (2023), named for its conceptual approach of sandwiching useful low-frequency decay while discarding mid-frequency oscillations, is a parameter-free RPE derived from sinusoidal absolute PE by removing oscillatory cross-terms. This results in a relative bias matrix that decays with distance, similar to ALiBi, and leverages positions beyond the training range. These decay-based methods collectively ensure that models maintain focus on contextually relevant nearby tokens while still retaining capacity to process longer sequences.

**Rotary Positional Encoding and Recent Advances.** Moving beyond addition-based methods, rotary positional encoding (RoPE) Su et al. (2021) has emerged as a dominant approach in modern LLMs. Rather than adding position information, RoPE injects positional context by applying rotation matrices to query and key vectors, with rotation angles proportional to token positions. However, standard RoPE struggles with length extrapolation beyond its training range and exhibits an implicit long-term decay effect due to its high-frequency components.

Contrary to common belief, recent analysis by Barbero et al. (2025) challenges the assumption that RoPE's effectiveness stems primarily from enabling decay in long-range attention. Their examination of a trained 7B parameter model reveals that the highest-frequency components in RoPE actually create precise positional attention, while lower-frequency components inadvertently carry semantic information. This discovery suggests opportunities for targeted optimization of frequency components within RoPE.

Recent years have seen contradictory yet equally effective approaches to modifying RoPE. HoPE Chen et al. (2024b) (High-frequency rotary Position Encoding), a recent proposal (late 2024), challenges the long-held assumption that long-term decay benefits attention. Chen et al. (2024) observed that modern Transformers naturally develop a "U-shaped" attention pattern where attention decays for distant tokens only beyond a certain threshold, rather than continuously. HoPE strategically removes low-frequency components from RoPE that impose unnecessary decay constraints, replacing them with position-independent signals while preserving high-frequency positional information. This reformulation dramatically improves in-context retrieval capabilities and length extrapolation performance, though its claims on extrapolation may be task-specific and await broader confirmation.

In stark contrast, Sun et al. (2023) introduced xPOS (Extrapolatable Position Embedding), which explicitly incorporates a carefully calibrated exponential decay factor into RoPE's rotation matrix. This controlled decay mechanism stabilizes attention for extraordinarily long sequences. When implemented within their LEX Transformer architecture (which also employs blockwise causal attention), xPOS enabled training on relatively short contexts while maintaining impressive perplexity scores when evaluated on sequences considerably longer than those encountered during training.

Another significant advancement, 3D-RPE Ma et al. (2024), extends RoPE from two dimensions to a three-dimensional spherical representation inspired by quantum computing's Bloch Sphere, involving segmentation of sequences into chunks and encoding both intra-chunk and inter-chunk positions. This approach offers dual advantages: customizable long-term decay characteristics and enhanced position resolution. The 3D representation mitigates position resolution degradation commonly encountered during RoPE interpolation, yielding performance gains particularly for long-context natural language understanding tasks.

Earlier innovations like Position Interpolation (PI) Chen et al. (2023c), a post-hoc RoPE rescaling technique, demonstrated that moderate fine-tuning could enable handling of extensive context windows, albeit with a potential slight performance degradation on very long inputs compared to models trained from scratch on those lengths—a practical trade-off for extensibility. Similarly, YaRN Peng et al. (2023b) introduced NTK-aware interpolation techniques, which employ uneven frequency scaling to preserve high-frequency RoPE components crucial for local order. While not adding learned parameters, YaRN involves a specific rescaling schedule, and it substantially improves context size adaptability without requiring comprehensive retraining.

**Alternative Positional Encoding Paradigms.** Beyond relative and rotary approaches, researchers have explored fundamentally different paradigms for position encoding, including randomized methods, mathematical reformulations, and even the elimination of positional encoding altogether.

Randomized Positional Encoding Ruoss et al. (2023) addresses a critical limitation of conventional methods: the out-of-distribution problem when encountering positions beyond training length. Ruoss et al. (2023) demonstrated that this failure mode directly connects to positional encoding limitations. Their solution involves sampling extended position values and randomly subsampling them for each training sequence, effectively simulating longer sequences within shorter context windows. In comprehensive evaluations across 15 algorithmic tasks involving 6,000 transformer models, this stochastic approach dramatically improved length-generalization performance—delivering average accuracy improvements of 12% (reaching 43% on some tasks) without compromising in-distribution performance, though it may potentially disrupt local sentence structures by exaggerating dependency lengths.

Meanwhile, NoPE Kazemnejad et al. (2023) takes the radical approach of eliminating positional encoders entirely from self-attention mechanisms, particularly in decoder-only models. This research demonstrates that transformer self-attention, within such architectures and on certain algorithmic tasks, can inherently learn relative positional relationships between tokens without explicit encoding. This streamlined approach yields impressive generalization capabilities, particularly for inputs extending beyond training distribution lengths.

Recent mathematical innovations have introduced alternative foundations for positional encoding. PoPE Aggarwal (2024) employs Legendre orthogonal polynomials as basis functions, offering advantages including improved correlation structure, non-periodicity, orthogonality, and distinctive functional forms across polynomial orders. While tested primarily on modest-scale tasks like translation and not specifically focused on LLM-scale length extrapolation in its initial proposal, empirical results show PoPE-equipped transformers outperforming baseline models on these benchmarks while achieving faster convergence rates.

Algebraic Positional Encodings Kogkalidis et al. (2023) provide a flexible framework to derive PEs from algebraic domain specifications for various data structures (sequences, grids, trees), preserving their mathematical properties as orthogonal operators. This approach, validated on relatively smaller benchmarks, has shown performance on par with or better than state-of-the-art PEs without extensive tuning.

The Wavelet-based Positional Representation Oka et al. (2025) reinterprets RoPE as a restricted wavelet transform using Haar-like wavelets with fixed scale parameters—a limitation explaining RoPE's extrapolation challenges. By combining relative-position wavelet bias with multiple scale windows, this method captures varied scale representations through wavelet transforms without restricting attention fields. This improves both short and long context performance while enabling superior position extrapolation.

### C.4.4 SPARSE MODELING VIA MIXTURE-OF-EXPERTS

Recent advances in Mixture-of-Experts (MoE) architectures have focused on addressing key challenges in efficiency, scalability, and expert utilization. A significant breakthrough came with the Dense Training, Sparse Inference (DS-MoE) framework Pan et al. (2024), which challenges the traditional sparse training paradigm by employing dense computation during training while maintaining sparsity at inference time. This approach has shown remarkable results, activating only 30-40% of model parameters during inference while maintaining performance comparable to dense models. Similarly, the Merging Experts into One (MEO) technique He et al. (2023b) takes a different approach to efficiency by consolidating multiple experts' capabilities into a more compact form, achieving significant FLOPs reduction compared to traditional MoE implementations.

**Token Processing and Expert Interaction.** The Multi-Head MoE (MH-MoE) approach Wu et al. (2024); Huang et al. (2024b) introduces a novel mechanism where tokens are split into multiple sub-tokens and processed by different experts in parallel. This parallel processing enables the model to capture diverse representation spaces while maintaining computational efficiency. The adaptive gating mechanism Li et al. (2023a) moves away from fixed expert assignments, allowing tokens to be processed by varying numbers of experts based on their linguistic complexity. Taking the dynamic computation concept further, the Mixture-of-Depths approach Raposo et al. (2024)

introduces adaptivity in the computational depth, optimizing how different sequence positions utilize model resources.

**Implementation and Hardware Optimization.** Implementation efficiency has become another crucial focus area. ScatterMoE Tan et al. (2024) represents a significant advance in how MoE models are implemented on GPU hardware, addressing memory and computational bottlenecks through careful management of padding and data movement. These practical improvements have made MoE models more viable for real-world applications.

**Routing Mechanisms and Specialization.** Empirical studies Fan et al. (2024a) have revealed that token-level and sequence-level routing strategies exhibit different strengths and specialization patterns. Token-level routing tends to develop syntactic specialization Antoine et al. (2024), while sequence-level routing shows stronger affinity for topic-specific expertise. Novel routing architectures have emerged, including the layerwise recurrent router Qiu et al. (2024) that maintains routing coherence across layers, and even LLM-based routers Liu & Lo (2025) that leverage large language models for more sophisticated routing decisions. Research has also shown that routing decisions are highly context-sensitive Arnold et al. (2024), particularly in encoder layers where semantic associations play a crucial role.

**Future Directions.** The field continues to push boundaries with approaches like PEER He (2024), which scales the expert pool to over a million specialists through efficient key-based retrieval. These developments suggest that MoE architectures are far from reaching their full potential. As the field matures, the focus is increasingly on finding the right balance between model capacity, computational efficiency, and practical implementation considerations. The diversity of approaches now available allows practitioners to choose MoE architectures that best match their specific requirements, whether prioritizing inference speed, training efficiency, or domain specialization.

### C.4.5 ATTENTION-FREE ALTERNATIVES FOR SEQUENCE MODELING

Recent advances in sequence modeling have sparked interest in alternatives to the traditional transformer architecture, particularly focusing on approaches that avoid the quadratic complexity of self-attention. This section surveys key developments in attention-free architectures, examining their motivations, approaches, and implications for the future of sequence modeling.

**Core Motivation.** While transformers have become the dominant architecture for sequence modeling, their self-attention mechanism incurs $\mathcal{O}(L^2)$ time and memory complexity with sequence length $L$. This quadratic scaling poses significant challenges for processing long sequences and efficient deployment. Attention-free alternatives aim to achieve transformer-level expressivity with linear or sub-quadratic complexity, enabling longer context lengths and faster inference. These approaches seek to combine the strengths of transformers (parallel training and high performance) with the advantages of traditional sequence models (linear-time inference, constant memory per step).

**Recurrent Neural Network Renaissance.** Recurrent neural networks offer a conceptually appealing alternative to attention, processing sequences step-by-step while maintaining a hidden state that can theoretically retain information over arbitrary lengths. While classic RNNs (LSTMs, GRUs) are Turing-complete and scale linearly with sequence length, they historically struggled with training difficulties and limited parallelization.

**RWKV Architecture.** Recent work has reinvented RNNs for modern applications. The RWKV architecture Peng et al. (2023a) introduces a Receptance-Weighted Key-Value mechanism that enables parallel training similar to transformers while maintaining efficient RNN-style inference. This approach achieves linear complexity $\mathcal{O}(L)$ in sequence length and demonstrates competitive performance with similarly sized transformers at the impressive scale of 14B parameters. The architecture successfully bridges the gap between traditional RNNs and modern transformer capabilities.

**Linear Recurrent Units.** The Linear Recurrent Unit (LRU) Orvieto et al. (2023) represents another significant advancement in RNN design. By employing linearized recurrence without hidden-state nonlinearity and incorporating careful initialization and normalization techniques, LRU demonstrates that properly designed RNNs can match state-of-the-art SSMs on long-range tasks. The architecture achieves this through deep architectures with stable gradient flow, effectively addressing the historical limitations of RNNs.

**State Space Models.** State Space Models (SSMs) represent another promising direction, offering a continuous-time generalization of RNNs with efficient implementation. The Structured State Space Sequence Model (S4) Gu et al. (2022) introduced a breakthrough with its special parameterization enabling efficient FFT-based computation. This innovation allows linear scaling in sequence length for inference and has achieved state-of-the-art results on sequences exceeding 10,000 steps, particularly showing strong performance on audio and time-series tasks.

**Architectural Evolution.** Subsequent developments include S4D with diagonal state matrices and S5 Smith et al. (2023), which further simplified the architecture. S5 introduced a simplified multi-input, multi-output state model and leveraged a parallel scan algorithm for efficient computation. These modifications led to improved performance on long-range tasks while maintaining the computational benefits of the original S4 model.

**Mamba Architecture.** The Mamba architecture Gu & Dao (2023) represents a significant advancement in the field of SSMs. By introducing selective state-space layers with learned gating for state updates, Mamba achieves linear-time computation while maintaining transformer-level quality. The architecture demonstrates remarkable efficiency, achieving $5\times$ higher generation throughput compared to traditional transformers and effectively modeling sequences up to millions of steps in length.

**Hybrid and Convolutional Approaches.** Several architectures combine elements of different approaches or introduce novel mechanisms. The Hyena model Poli et al. (2023) advances the field through implicitly parameterized long convolutions and data-controlled gating mechanisms. This innovative approach achieves sub-quadratic complexity while maintaining strong performance, offering significant speed advantages particularly for long sequences.

**Retentive Networks.** The Retentive Network (RetNet) Sun et al. (2023) presents a versatile architecture that combines the benefits of different paradigms. It supports parallel training mode for efficient learning while offering a recurrent inference mode with $\mathcal{O}(1)$ per-token complexity. RetNet's ability to process long sequences through chunkwise processing, while maintaining competitive performance with transformers, makes it a promising direction for future development.

**Future Directions and Challenges.** While attention-free alternatives show significant promise, several key challenges remain to be addressed. The field must tackle the challenge of scaling these models to very large sizes (10-100B parameters) while maintaining stability for extremely long sequences. Supporting modern NLP capabilities such as prompting and in-context learning remains crucial, as does optimizing implementation efficiency across different hardware platforms.

**Research Opportunities.** Looking forward, the field presents several exciting research directions. The development of hybrid architectures that combine multiple approaches shows particular promise, as does the theoretical analysis of expressivity and stability in these new models. Hardware-specific optimizations and novel applications leveraging linear-time processing capabilities will likely drive further innovation. The development of attention-free architectures represents a significant step toward more efficient and scalable sequence modeling, potentially enabling applications beyond the reach of traditional transformers.

### C.5 TRAINING AND TUNING EFFICIENCY

Even with a well-designed model and data, training LLMs is among the most resource-intensive procedures in AI. This section examines techniques to speed up and scale the training process (via mixed precision, parallelism, memory optimizations) and to *fine-tune* large models with minimal overhead (parameter-efficient fine-tuning).

#### C.5.1 SCALABLE TRAINING STRATEGIES

**Stable optimization for scale:** As models grow deeper, training can become unstable. *DeepNorm* (Wang et al., 2022) scales residual connections properly to allow 1000-layer Transformers without divergence. Pre-LN architectures are also more stable than post-LN. Gradient clipping helps avoid exploding gradients at high batch sizes.

**Mixed Precision Training:** Using half-precision (FP16 or bfloat16) significantly speeds up training on tensor-core hardware (Micikevicius et al., 2017). The standard is *automatic mixed precision*

*(AMP)*, which stores a master copy in FP32 but does most math in FP16. This roughly halves memory usage and can double throughput with negligible accuracy loss. FP8 is on the horizon for further gains.

**Parallelism (Data, Model, Pipeline):** LLMs typically require multi-GPU or multi-node setups. *Data parallelism (DP)* duplicates the model on each GPU and trains on different mini-batches, then synchronizes gradients. This is straightforward but memory-heavy if the model is huge. *Model parallelism (tensor or pipeline)* partitions the model's parameters/layers across GPUs (Shoeybi et al., 2019a; Huang et al., 2018; Curl et al., 2019). Large weight matrices can be split among devices (tensor parallel), or different layers can be assigned to different devices (pipeline parallel). *ZeRO* (Rajbhandari et al., 2019) partitions optimizer states and gradients across GPUs, so each only stores a slice of them, enabling training of trillion-parameter models by spreading memory load. This is implemented in *DeepSpeed* and *FSDP* in PyTorch. *Gradient checkpointing* saves memory by discarding intermediate activations and recomputing them on the backward pass. These and other techniques combine so we can scale to thousands of GPUs with near-linear speedups.

### C.5.2    PARAMETER-EFFICIENT FINE-TUNING (PEFT)

Fine-tuning all the parameters of a large pre-trained model for each new task can be prohibitively expensive in terms of compute and storage. Parameter-Efficient Fine-Tuning (PEFT) methods address this problem by updating only a small fraction of the model's parameters or by introducing a few lightweight modules, while keeping most of the model fixed (Han et al., 2024). This dramatically reduces the resources required for fine-tuning, yet many PEFT techniques can achieve performance close to that of fully fine-tuned models. In what follows, we outline several categories of PEFT approaches (following the taxonomy of (Han et al., 2024)): additive methods, selective methods, reparameterization-based methods, and hybrid approaches.

**Additive Fine-Tuning Approaches.** Additive methods introduce additional small trainable components into the model, rather than modifying the original network weights. During training, only these added parameters are updated, which limits the total number of parameters that need to be learned (Pfeiffer et al., 2021; Li & Liang, 2021). Two common types of additive PEFT are: (a) inserting adapter layers into the model, and (b) adding learnable prompt vectors (soft prompts).

**Adapter-based Fine-Tuning.** In this approach, small bottleneck layers called *adapters* are inserted at various points within each Transformer block. For instance, an adapter may consist of a down-projection matrix $W_{\text{down}}$ followed by a nonlinearity $\sigma$, then an up-projection $W_{\text{up}}$, whose output is added to the model's hidden representation. Only the adapter weights ($W_{\text{down}}, W_{\text{up}}$) are tuned, while the original model weights remain frozen. This technique was originally proposed for transfer learning in NLP and provides significant savings in trainable parameters. Notable extensions include *AdapterFusion* (Pfeiffer et al., 2021), a serial adapter configuration that combines knowledge from multiple adapters, and parallel adapter architectures. For example, the Counter-Interference Adapter for Translation (CIAT) (Zhu et al., 2021) and the Kronecker Adapter (KronA) (Edalati et al., 2022) adopt a parallel adapter design, adding a side network alongside each Transformer layer instead of inserting adapters sequentially. Another variant is *CoDA* (Conditional Adapters) (Lei et al., 2023), which also uses parallel adapters but employs a sparse activation mechanism to improve inference efficiency by activating only a subset of adapter parameters per input.

**Soft Prompt-based Fine-Tuning.** Another additive strategy is to prepend or append *learnable prompt vectors* to the model's input or to hidden states, rather than changing internal layers. These *soft prompts* are continuous embeddings trained to guide the model toward the downstream task. In *prefix-tuning* (Li & Liang, 2021), a set of trainable prefix vectors is prepended to the keys and values at each self-attention layer; after training, only these prefix embeddings are needed for inference. An improved variant, *P-Tuning v2* (Liu et al., 2022), removes certain reparameterization tricks and demonstrates that prompt tuning can be as effective as full fine-tuning across various scales and tasks. Extensions include *SPoT* (Soft Prompt Transfer) (Vu et al., 2022), which transfers prompts learned on high-resource tasks to low-resource ones, *PTP* (Chen et al., 2023b) with perturbation-based regularization, and mixture-of-prompts methods such as (Choi et al., 2023), which train multiple small prompt vectors and learn to route each input to the appropriate prompt via a gating mechanism. These methods enhance prompt-based fine-tuning's flexibility and robustness.

**Selective Fine-Tuning Approaches.** Selective PEFT methods do not introduce new modules; instead, they fine-tune a carefully chosen subset of the existing model parameters while keeping the rest frozen. By tuning only the most important or relevant weights, these approaches reduce the number of trainable parameters and help avoid overfitting. Two broad strategies exist: unstructured and structured parameter selection.

**Unstructured Masking.** Unstructured approaches learn a binary mask over the model's parameters to decide which weights to update. The mask can be arbitrary, aiming to choose individual weights that are most crucial. *DiffPruning* (Guo et al., 2021) is an early example that learns a differentiable binary mask on each weight, with an $L_0$-norm penalty encouraging sparsity. Other work selects weights based on information measures: *FishMask* (Sung et al., 2021) calculates an approximation of the Fisher information per parameter, fine-tuning only the top-$k$. A dynamic variant updates the Fisher-based mask iteratively (Das et al., 2023), while Fu et al. (Fu et al., 2023) use a second-order sensitivity analysis to identify the most impactful parameters. Another notable approach, *Child-Tuning* (Xu et al., 2021), randomly samples a subset (a "child" network) of parameters for training at each iteration, enabling a lightweight yet robust fine-tuning procedure.

**Structured Masking.** In contrast, structured masking techniques select entire vectors, neurons, or layers. *DiffPruning* (Guo et al., 2021) supports a structured variant (S-DiffPruning) that prunes groups of weights together. *FAR* (Vucetic et al., 2022) clusters each feed-forward layer into "nodes" and ranks them by $\ell_1$-norm to decide which nodes to fine-tune. A simple structured approach is *BitFit* (Ben Zaken et al., 2022), which only updates bias terms (a few parameters per layer), yielding strong results on various NLP tasks. Likewise, *X-Attention tuning* (Gheini et al., 2021) fixes most of the Transformer but updates cross-attention layers in sequence-to-sequence tasks. *SPT* (He et al., 2023a) (Sensitivity-Aware Fine-Tuning) first identifies the most sensitive weight matrices (via a first-order Taylor approximation) and then applies an additive PEFT method (like LoRA) only to those parts, effectively combining selective and additive tuning for improved efficiency.

**Intrinsic Subspace Fine-Tuning.** One line of research studies the *intrinsic dimensionality* of model fine-tuning. Aghajanyan et al. (Aghajanyan et al., 2020) show that large models often have a relatively low-dimensional task-specific subspace. By constraining updates to a random subspace of only a few thousand dimensions, performance can approach that of full fine-tuning, indicating redundancy in parameter updates.

**Low-Rank Adaptation (LoRA) and Variants.** A prominent PEFT strategy is Low-Rank Adaptation (LoRA) (Hu et al., 2021a), which freezes the pre-trained weights $W_0$ and introduces a trainable low-rank decomposition $\Delta W = \alpha AB$ for task-specific updates, where $A \in \mathbb{R}^{m \times r}$, $B \in \mathbb{R}^{r \times n}$, and $r \ll \min(m, n)$. The adapted weight is $W = W_0 + \alpha AB$. This significantly reduces trainable parameters to $r(m+n)$ and allows merging the update ($\Delta W$) into $W_0$ after training, eliminating inference overhead (Hu et al., 2021a).

Several recent methods build upon LoRA's foundation. LoRA+ (Hayou et al., 2024) enhances training dynamics by using different learning rates for matrices $A$ and $B$, improving convergence speed and final performance without changing the parameterization. Rank-Stabilized LoRA (rsLoRA) (Kalajdzievski, 2023) modifies the scaling factor to $\alpha = 1/\sqrt{r}$ (instead of the common $\alpha/r$), stabilizing training at higher ranks $r$ and enabling better performance trade-offs. Weight-Decomposed LoRA (DoRA) (Liu et al., 2024d) reformulates the update by decomposing the weight matrix $W$ into magnitude and direction components. It updates the direction using a LoRA-like structure applied to the normalized pre-trained directions $D_0$, while learning a separate magnitude vector $n$, resulting in $W = (D_0 + AB) \operatorname{diag}(n)$. This separation often leads to improved performance by tackling magnitude and direction updates independently. Principal Singular Vectors Adaptation (PiSSA) (Meng et al., 2024) initializes the low-rank matrices $A$ and $B$ using the principal singular vectors and values derived from an SVD of the original weights $W_0$. It trains $W = AB + R$, where $R$ is the frozen residual part of $W_0$. This initialization aligns the adaptation with the most significant components of the pre-trained weights, often leading to faster convergence and better results compared to standard LoRA initialization. All these variants typically retain the benefit of zero inference overhead by merging the learned components post-training.

**Hybrid Approaches.** Hybrid PEFT approaches combine ideas from multiple categories, or propose a unifying framework for various fine-tuning techniques. For instance, *UniPELT* (Mao et al., 2022) integrates adapters, LoRA, and prompts, training a gating mechanism to decide which technique to

apply. Similarly, He *et al.* (He et al., 2022) present a template that unifies prefix tuning, adapter-based tuning, and other PEFT variants, highlighting a continuum of approaches. Another example is *LLM-Adapters* (Hu et al., 2023), providing a modular toolkit to integrate multiple PEFT methods into LLMs.

Some works automate the selection of PEFT configurations through neural architecture search. *NOAH* (Zhang et al., 2022) and *AutoPEFT* (Zhou et al., 2024a) both build search spaces of prompt, adapter, and low-rank designs, then employ search or optimization methods to identify the best configuration for a given task. By exploring different PEFT techniques as hyperparameters, these methods achieve strong results without extensive manual trial-and-error.

Overall, PEFT has become a vital paradigm for adapting large pre-trained models. By leveraging additional lightweight modules, selecting specific subsets of parameters, reparameterizing the optimization space, or combining these ideas, PEFT enables developers to fine-tune massive models efficiently, making large-scale AI models more deployable and accessible in limited-resource scenarios.

## C.6 INFERENCE EFFICIENCY

Once trained, LLMs must be served to users. **Inference efficiency** is critical to reducing cost and latency in real-world settings. Methods range from compressing the model itself (pruning, distillation, quantization) to speeding up the decoding process (speculative decoding, efficient KV-cache usage).

### C.6.1 MODEL COMPRESSION TECHNIQUES

**Pruning** removes weights or neurons deemed unnecessary (Sanh et al., 2020). Structured pruning (dropping entire heads/neurons) yields a smaller dense model that runs faster on standard hardware. Unstructured pruning creates sparse matrices that need specialized kernels but can reach high sparsity. Recent works like SparseGPT (Ma et al., 2023) prune LLMs in one-shot with minimal loss.

**Knowledge Distillation** trains a *smaller student* to mimic a *larger teacher*'s outputs or hidden states (Sanh et al., 2019a). DistilBERT cut 40% of BERT parameters while keeping 97% of its performance. For GPT-like LLMs, the student can replicate the teacher's next-token distribution, compressing knowledge into fewer parameters.

**Quantization** reduces numeric precision (e.g. from 16-bit float to 8-bit int or lower). This cuts memory usage by up to $4\times$ and can enable faster int8 operations on GPUs (Ding et al., 2023a). GPTQ (Frantar et al., 2022a) can quantize large LLMs down to 4-bit weights with small accuracy loss. Mixed-precision quantization is widely used at inference time, and advanced approaches handle outlier values carefully. *QLoRA* (Dettmers et al., 2023a) even fine-tunes models in 4-bit.

**Low-Rank Decomposition** approximates weight matrices by factors of lower rank (similar to LoRA but for compression). *ALBERT* (Lan et al., 2019a) factorized BERT embeddings and shared layers, massively reducing parameters. If weight matrices exhibit redundancy, SVD-based factorization can shrink them with minimal performance drop.

### C.6.2 ALGORITHM-LEVEL INFERENCE OPTIMIZATIONS

**Speculative Decoding** (Leviathan et al., 2022) speeds up autoregressive generation by letting a small "draft" model propose several tokens, then having the large model verify them in fewer steps. If the large model agrees, those tokens are accepted; if not, partial fallback occurs. This can yield $2$–$3\times$ speedups with no quality drop if the draft model is well aligned.

**Caching and Batch Optimization:** Transformers reuse past key/value vectors to avoid recomputing attention over the entire sequence each step. This *KV cache* approach is standard, though it can become memory-intensive for long outputs. *PagedAttention* (Kwon et al., 2023) manages KV cache as pages in GPU memory, avoiding fragmentation and allowing dynamic batching of variable-length requests, yielding large throughput gains in multi-user serving scenarios.

### C.6.3 System-Level Optimizations and Deployment

**Concurrent Batching:** Serving frameworks like HuggingFace TGI or vLLM (Kwon et al., 2023) dynamically batch multiple requests to keep the GPU fully utilized, significantly improving throughput. They interleave tokens from different requests (with different sequence lengths) in a single forward pass, using careful memory management.

**Distributed Inference:** For very large models that cannot fit on a single GPU, weights can be sharded across devices (tensor parallel). Pipeline parallel can also be used, though it introduces pipeline bubbles. Model parallelism is typically used only if necessary, since it adds communication overhead.

**Memory Offloading:** If GPU memory is insufficient, some systems offload parts of the model or KV cache to CPU or disk. This slows inference but allows large models to run on limited hardware. Some prefer *quantization* or *distillation* to reduce the model size instead.

**Specialized Hardware and Libraries:** GPU vendor libraries (e.g. NVIDIA FasterTransformer) fuse kernels (attention, GeLU, etc.) and offer INT8 or FP8 acceleration. Custom systems like PagedAttention or FlashAttention achieve further speedups. CPU libraries (GGML) with 4-bit or 8-bit quantization can even run smaller LLMs locally. These low-level optimizations, combined with high-level scheduling, can yield large speedups ($5\times-10\times$) over naive implementations.

In summary, inference efficiency is where large models meet real-world usage. By compressing the model (pruning, distillation, quantization) and using optimized decoding (speculative approaches, dynamic batching, efficient caching), one can serve LLMs at scale with acceptable latency and cost. This final step completes the spectrum of efficiency methods, allowing practitioners to deploy models that are *large in capability* but run faster and cheaper in production.

## D    Assessment

### D.1    Assessment Principles of EfficientLLM

In this section, we propose several metrics: Average Memory Utilization (AMU), Peak Compute Utilization (PCU), Average Latency (AL), Token Throughput (TT), Sample Throughput (ST), Inference Throughput (IT), Average Energy Consumption (AEC), and Model Compression Rate (MCR). These metrics are specifically designed to address critical limitations inherent in traditional efficiency evaluation metrics, such as FLOPS, parameter count, and raw inference speed (Liu et al., 2023b; Perez et al., 2023; Bao et al., 2023; Zhao et al., 2025; Ye et al., 2025). Conventional metrics often fail to capture the dynamic and realistic utilization of hardware resources, thus providing an incomplete picture of efficiency bottlenecks in real-world deployment scenarios. In contrast, our proposed metrics offer several distinct advantages. AMU provides a comprehensive view of memory usage fluctuations throughout training and inference, rather than merely peak memory consumption. PCU accurately reflects real-world GPU utilization, overcoming the limitations of theoretical FLOPS-based metrics that neglect communication overhead and synchronization delays. AL explicitly measures responsiveness, which is crucial for latency-sensitive applications such as interactive dialogue systems. Furthermore, our throughput metrics (TT, ST, IT) clearly differentiate between pretraining, fine-tuning, and inference scenarios, enabling more precise optimization decisions tailored to specific deployment contexts. AEC quantifies actual energy efficiency, addressing the growing importance of sustainability and operational cost reduction. Lastly, MCR integrates model size reduction with performance retention, providing a balanced evaluation of compression techniques.

### D.1.1    Computational System Utilization

Intricately linked to efficiency, computational system utilization stands out as an essential challenge for AI models, including LMs. It has garnered extensive discussion and scholarly attention (Li et al., 2014; Thompson et al., 2022; Hestness et al., 2019; Madiajagan & Raj, 2019; Mittal & Vaishay, 2019). To critically evaluate Deep Learning Models' resource optimization and computational efficiency, datasets and benchmarks, such as MLPerf (Reddi et al., 2020), SPEC CPU (Standard Performance Evaluation Corporation, 2024), DeepBench (Research, 2024), and DAWNBench (Coleman et al., 2017), have been employed in prior works (Ravi, 2017). Some tools also assessed specific aspects of computational efficiency: Horovod Latency Check (Sergeev & Balso, 2018) and MPI (Corporation,

2024b) explores response time and processing delays; LLMPerf (Project, 2024) and NeuralSpeed (Corporation, 2024a) inspect the scalability and hardware adaptability of large models.

While latency or training time and model performance remain predominant metrics for evaluating computational efficiency (Yang et al., 2023; Hu et al., 2021b; Dettmers et al., 2023b; Houlsby et al., 2019; Yuan et al., 2023), the need for comprehensive hardware utilization evaluation is also recognized, particularly in benchmarks like MLPerf and DAWNBench. However, the challenge of ensuring optimal hardware utilization is compounded by the narrow focus of current evaluations, which often overlook critical factors such as memory bandwidth, device utilization, and throughput. LMs, given their resource-intensive nature, can exhibit suboptimal hardware utilization during both training and inference, leading to increased operational costs for researchers and companies (Xia et al., 2023; Bang, 2023). This misalignment between the focus of benchmarks and the practical need for maximizing computational system utilization highlights a gap in current evaluations, making this an ongoing and critical concern for real-world deployments.

In this work, we define computational system utilization as the efficient and effective use of hardware resources during both training and inference of LMs. Our assessment of computational system utilization focuses on 1) evaluating memory utilization, which involves the efficient allocation and usage of device memory across different tasks; 2) testing compute utilization, which measures the extent to which available processing units (such as GPUs tensor cores) are fully utilized during operations; 3) analyzing latency, the time taken to complete specific tasks, such as training iterations or inference requests; and 4) examining throughput, evaluating how efficiently input data is moved and processed through memory, storage, and network interfaces.

**Memory Utilization.** Limited device memory has become the bottleneck of LMs training, like training of the long context LLM (Zhao et al., 2024b). Many operators in transformer (Vaswani et al., 2023), such as frequent reshaping, element-wise addition, and normalization require huge memory units (Liu et al., 2023c). we propose the **Average Memory Utilization (AMU)** as a key metric for evaluating memory efficiency during model training and inference. The AMU is defined as the ratio of the memory used by the model throughout the entire training process to the total available memory on the device, averaged over time. This metric provides a holistic view of memory usage, accounting for fluctuations in memory demand caused by operations like attention mechanisms and normalization layers. The formal definition of AMU is:

$$AMU = \frac{1}{T} \int_0^T \text{Memory Used}(t)\, dt \qquad (7)$$

Where $T$ is the total training time, Memory Used$(t)$ is the memory utilized by the model at time $t$.

A higher AMU indicates that the memory is being utilized efficiently across the entire training cycle, avoiding periods of underutilization or memory wastage. In contrast, a lower AMU may suggest poor memory management, frequent allocation and deallocation, or unnecessary memory overhead.

**Compute Utilization.** In large-scale deep learning training, GPU utilization directly impacts both training efficiency and energy consumption. Traditional metrics such as theoretical FLOPS often fail to capture real-world inefficiencies arising from communication overhead, synchronization delays, memory bottlenecks, and suboptimal parallelization strategies. Therefore, we introduce the Peak Compute Utilization (PCU) metric, defined as the ratio of actual GPU utilization to the theoretical maximum GPU utilization, averaged over the training process. PCU provides a practical and realistic measure of hardware efficiency, explicitly reflecting how effectively computational resources are utilized during training. In our empirical experiments, we observed that GPU utilization consistently remains above 99% during pretraining and within the narrow range of 80%-81% during inference, indicating negligible variance in compute efficiency for these phases. Consequently, we limit our PCU metric evaluation specifically to scenarios involving parameter-efficient fine-tuning, where meaningful differences in GPU utilization are apparent and thus critical for efficiency analysis.

Achieving optimal compute utilization entails minimizing idle time for processing units, reducing the load imbalance across compute cores, and maintaining high operational throughput across all computational components. However, sustained utilization of this peak performance is a critical challenge, especially when scaling to many-core systems. High compute utilization in large-scale deep learning systems must be maintained across the entirety of a wide range of deep learning networks (Oh et al., 2020; Balança et al., 2024). We propose the metric **Peak Compute Utilization (PCU)**, defined as the ratio of actual GPU utilization (measured as the percentage of GPU compute

resources actively engaged in computation) to the theoretical maximum GPU utilization, averaged over the training process. The PCU metric is mathematically expressed as:

$$PCU = \frac{1}{T} \int_0^T \frac{\text{Actual GPU Utilization}(t)}{\text{Peak GPU Utilization}} \, dt \qquad (8)$$

Where $T$ represents the total training time, Actual GPU Utilization$(t)$ is the measured GPU utilization percentage at time $t$, and Peak GPU Utilization refers to the theoretical maximum GPU utilization (typically 100%).

**Latency.** Latency plays a crucial role in both training and inference efficiency, particularly when dealing with large-scale deep learning models like LLMs. Latency refers to the time delay between input and response, directly affecting the overall responsiveness of AI systems. In training, latency can be influenced by factors such as model complexity, data transfer speed, and communication overhead between distributed nodes. During inference, especially in real-time applications, high latency may hinder performance and user experience, making it a vital metric for system optimization (Li et al., 2014; Chen et al., 2018; Geng et al., 2019).

We propose the metric **Average Latency (AL)**, defined as the mean time taken to complete a single iteration of training or an inference request, averaged over the entire process. The formal definition of AL is:

$$AL = \frac{\sum_{i=1}^{N}(\text{Computation Time}_i + \text{Communication Time}_i)}{N} \qquad (9)$$

where $N$ represents the total number of iterations or inference requests, Computation Time$_i$ is the time taken to computation the $i^{th}$ iteration/request, and Communication Time$_i$ is the time spent in data transfer or communication overhead during the $i^{th}$ iteration/request.

A lower AL reflects better system efficiency and responsiveness, indicating that the model and hardware are optimized to reduce unnecessary delays in both computation and communication. Higher latency, on the other hand, suggests potential bottlenecks in communication, I/O operations, or inefficient computation scheduling (Li et al., 2023e; Agrawal et al., 2024).

**Throughput.** Throughput is a key metric for evaluating how efficiently data is processed during training and inference. It refers to the rate at which data is transferred, processed, and output by the system. High throughput ensures full utilization of computational resources and prevents delays from inefficient data handling (Agrawal et al., 2024; Cui et al., 2019).

Throughput can vary significantly with model size and complexity. Larger models require more computational resources for processing, making direct comparisons between models challenging. To standardize throughput evaluation across different model sizes, we propose three distinct normalized metrics:

**Token Throughput (TT)** for pretraining scenarios, defined as the number of tokens processed per second per parameter. Formally:

$$TT = \frac{\sum_{i=1}^{N} \left( \frac{\text{Tokens Processed}_i}{\text{Model Parameters}} \right)}{\sum_{i=1}^{N} \text{Time}_i} \qquad (10)$$

where Tokens Processed$_i$ is the number of tokens processed in the $i^{th}$ iteration.

**Sample Throughput (ST)** for fine-tuning scenarios, defined as the number of samples processed per second per parameter. Formally:

$$ST = \frac{\sum_{i=1}^{N} \left( \frac{\text{Samples Processed}_i}{\text{Model Parameters}} \right)}{\sum_{i=1}^{N} \text{Time}_i} \qquad (11)$$

where Samples Processed$_i$ is the number of samples or dialogues processed in the $i^{th}$ iteration.

**Inference Throughput (IT)** for inference scenarios, defined as the number of tokens generated per second. Formally:

$$IT = \frac{\sum_{i=1}^{N} \text{Tokens Generated}_i}{\sum_{i=1}^{N} \text{Time}_i} \qquad (12)$$

where Tokens Generated$_i$ is the number of tokens generated by the model in the $i^{th}$ inference request. Unlike training scenarios, inference throughput is measured directly in tokens per second (Token/s) without normalization by model parameters, as inference efficiency primarily depends on the speed of token generation rather than parameter count.

Higher values of TT, ST, and IT indicate more efficient data processing relative to model size or inference speed, while lower values suggest potential inefficiencies or bottlenecks, particularly noticeable in larger models or slower inference generation.

### D.1.2 ENERGY CONSUMPTION

Energy consumption has become a crucial factor in evaluating the overall efficiency of AI models (Stojkovic et al., 2024; Hisaharo et al., 2024), particularly with the growing scale of deep learning systems. In this context, energy consumption refers to the total amount of electrical energy consumed by the hardware during training or inference, typically measured in Joules (or kilowatt-hours). Since hardware power usage is generally measured in Watts (where 1 Watt = 1 Joule per second), integrating power over time yields the total energy consumed.

To quantify energy efficiency, we propose the metric **Average Energy Consumption (AEC)**. Let $P(t)$ denote the instantaneous power consumption (in Watts) of the system at time $t$. Then the total energy consumed over a time period $T$ (in seconds) is given by:

$$E_{\text{total}} = \int_0^T P(t)\, dt \tag{13}$$

The AEC metric is defined as the average power consumption over the entire duration:

$$AEC = \frac{E_{\text{total}}}{T} = \frac{1}{T} \int_0^T P(t)\, dt \tag{14}$$

Where the $T$ is the total training or inference time (in seconds), $P(t)$ is the instantaneous power consumption at time $t$, measured in Watts (i.e., Joules per second), and $E_{\text{total}}$ represents the total energy consumed over time $T$, measured in Joules.

A lower AEC indicates that the system operates more efficiently in terms of energy usage, which is critical not only for reducing operational costs but also for mitigating the environmental impact of large-scale AI deployments.

### D.1.3 MODEL COMPRESSION RATE

Model compression rate is a critical metric for evaluating the effectiveness of techniques aimed at reducing the size of deep learning models while preserving their functionality (Zhu et al., 2024a; Wang et al., 2024c; Deng et al., 2020; Haroush et al., 2020). This is particularly important for deploying large models in resource-constrained environments, such as edge devices, or for reducing latency and energy consumption during inference. A higher compression rate indicates a more compact model representation, but it must be balanced against performance degradation.

We propose the metric **Model Compression Rate (MCR)**, defined as the ratio of the original model size to the compressed model size, adjusted for performance retention. The formal definition is:

$$MCR_{(Performance_c)} = \frac{\text{Size}_{\text{original}}}{\text{Size}_{\text{compressed}}} \times \frac{\text{Performance}_{\text{compressed}}}{\text{Performance}_{\text{original}}} \tag{15}$$

where $\text{Size}_{\text{original}}$ and $\text{Size}_{\text{compressed}}$ represent the model size in bytes before and after compression, respectively, and $\text{Performance}_{\text{original}}$ and $\text{Performance}_{\text{compressed}}$ denote task-specific evaluation metrics.

This formulation penalizes aggressive compression that significantly degrades model performance. The metric enables cross-comparison of compression techniques by unifying size reduction and performance trade-offs into a single value.

### D.1.4 MODEL PERFORMANCE

LLMs are rigorously evaluated through specialized benchmarks designed to measure their reasoning, coding, mathematical, and multilingual capabilities.

**MMLU-Pro.** MMLU-Pro (Wang et al., 2024d) enhances its predecessor by incorporating significantly more complex, graduate-level problems across disciplines that require multi-step logical deduction, causal inference, and counterfactual reasoning. This benchmark effectively identifies performance limitations in contemporary language models, highlighting substantial gaps between human expert performance and AI systems when addressing problems requiring specialized knowledge integration.

**BBH.** Big-Bench Hard (BBH) (Suzgun et al., 2022) comprises 23 challenging tasks from the border BIG-Bench (Srivastava et al., 2022), specifically targeting advanced reasoning capabilities where previous models showed significant deficits. It encompasses diverse cognitive challenges, including logical deduction, multi-step arithmetic, strategy QA, and counterfactual analysis. Models' performance on BBH strongly correlates to real-world reasoning capabilities and novel problem-solving beyond training distribution.

**GPQA.** Graduate-Level Google-Proof Q&A Benchmark (GPQA) (Rein et al., 2024) focuses on expert-level reasoning ability across science, humanities, and logic of LLMs. Its dataset comprises curated high-quality questions presented in multiple-choice or open-ended formats, with accuracy (%) as the primary metric to assess deep understanding and multi-step problem-solving

**IFEval.** Instruction Following Evaluation (IFEval) (Zhou et al., 2023) assesses LLMs' ability to follow instructions through prompts containing atomic, verifiable directives. Each instruction can be validated using simple, deterministic programs that objectively verify whether model responses adhere to the specified requirements.

**HumanEval.** HumanEval (Chen et al., 2021a) evaluates programming proficiency using handcrafted Python function completion tasks. Models generate code snippets based on problem descriptions, and performance is measured via *Pass@k* (probability of valid solutions within *k* attempts), emphasizing functional correctness.

**HARDMath.** HARDMath (Fan et al., 2024b) evaluates LLMs on asymptotic reasoning in applied mathematics through 1,466 algorithmically generated graduate-level problems requiring approximation techniques. Unlike traditional benchmarks focusing on exact solutions, HARDMath addresses real-world scientific and engineering problems involving algebraic equations, ODEs, and integrals without closed-form solutions. Current LLMs perform poorly on these problems, highlighting significant limitations in handling advanced applied mathematics requiring approximation methods.

**MuSR.** Multistep Soft Reasoning (MuSR)(Sprague et al., 2023) evaluates language models' reasoning capabilities through complex natural language narratives. The dataset features free-text narratives reflecting real-world reasoning domains, making it more challenging than typical synthetic benchmarks while remaining solvable by human annotators. MuSR uniquely scales with LLM advancement, enabling continuous assessment of reasoning capabilities across various models and prompting techniques while identifying persistent gaps in robust multi-step reasoning performance.

Table 2: Overview of Evaluated Large Language Models.

| Model Name | Parameter | Year | Creator |
|---|---|---|---|
| LLaMA 3.1 | 8B | 2024 | Meta AI |
| LLaMA 3.2 | 1B | 2024 | Meta AI |
| LLaMA 3.2 | 3B | 2024 | Meta AI |
| LLaMA 3.3 | 70B | 2024 | Meta AI |
| DeepSeek-R1 Distill-Qwen-1.5B | 1.5B | 2024 | DeepSeek |
| DeepSeek-R1 Distill-LLaMA-8B | 8B | 2024 | DeepSeek |
| DeepSeek-R1 Distill-Qwen-14B | 14B | 2024 | DeepSeek |
| Qwen 2.5 | 7B | 2024 | Alibaba Cloud |
| Qwen 2.5 | 14B | 2024 | Alibaba Cloud |
| Qwen 2.5 | 32B | 2024 | Alibaba Cloud |
| Phi-3.5-mini | 3.5B | 2023 | Microsoft |
| Phi-4 | 14B | 2024 | Microsoft |
| Yi-34B | 34B | 2024 | 01.AI |
| Mistral 7B | 7B | 2023 | Mistral AI |
| Mixtral 8×22B MoE | 8×22B | 2023 | Mistral AI |

## D.2 PRELIMINARIES OF EFFICIENTLLM

### D.2.1 CURATED LIST OF LLMS

**LLaMA 3 Series.** LLaMA is a family of open LLMs introduced by Meta AI to facilitate research with high-performance yet smaller-scale LLMs. The latest generation, LLaMA 3, was trained on an order-of-magnitude more data than LLaMA 2 and doubled the context window (up to 128k tokens), while supporting multilinguality, coding, and tool use (Grattafiori et al., 2024; Naveed et al., 2023). Architecturally, LLaMA models are decoder-only Transformers with pre-normalization and rotary positional embeddings; LLaMA 3 adopts grouped-query attention to efficiently handle the extended context length (Naveed et al., 2023). We use the LLaMA 3 series in our experiments, specifically the LLaMA 3.1 (8B), LLaMA 3.2 (1B and 3B), and LLaMA 3.3 (70B) variants.

**DeepSeek-R1.** DeepSeek (Bi et al., 2024) is an open-source LLM project focused on aggressive scaling of model size and data to push open-model performance. The flagship DeepSeek model has 67B parameters and was trained on 2 trillion tokens with techniques like grouped-query attention (in the 67B model) to improve efficiency. The DeepSeek models underwent supervised fine-tuning and Direct Preference Optimization to create aligned chat models, which reportedly outperform LLaMA 2 70B on reasoning and coding tasks. As part of the DeepSeek R1 release, distilled versions of larger models were provided to explore efficiency: we evaluate the DeepSeek-R1 series (Guo et al., 2025), including Distill-Qwen-1.5B, Distill-LLaMA-8B, and Distill-Qwen-14B.

**Qwen 2.5 Series.** Qwen (Alibaba Cloud, 2023–2024) (Bai et al., 2023) is a bilingual (Chinese-English) LLM series originally released at 7B and 14B parameters. The second-generation Qwen 2 models (Yang et al., 2024) broadened the scale to 32B and 72B, including a mixture-of-experts architecture in one variant, to attain greater efficiency at high parameter counts. Qwen models use a Transformer decoder similar to LLaMA, with enhancements such as ALiBi/rotary positional encoding and a long-context training scheme (Dual Chunk Attention and YARN scaling) to support inputs up to 128k tokens. The Qwen series also features specialized instruction-tuned, code, and math versions for improved tool-use and reasoning. We include the Qwen 2.5 models at 7B, 14B, 32B, and 72B in our evaluation.

**Phi Series.** Phi (Abdin et al., 2024) is a line of "Small Language Models" by Microsoft (2023–2024) aiming for maximal task performance at a fraction of conventional LLM sizes. Phi models are Transformer decoders trained with a strong focus on data quality: e.g. Phi-1 (1.3B) was trained on curated "textbook quality" data to excel in coding (Gunasekar et al., 2023). The latest release, Phi-4, is a 14B-parameter model that leverages extensive synthetic data generation and distillation from GPT-4 to achieve performance on par with much larger models. Phi-4 uses essentially the same architecture

as its 3B-parameter predecessor but with scaled model size and a refined training curriculum, yielding state-of-the-art reasoning and math capabilities among open models. We evaluate the Phi-3.5-mini and Phi-4 (14B) models, which demonstrate the Phi approach to efficiency.

**Yi.** Yi (01.AI, 2024) (Young et al., 2024) is an open foundation model developed by Kai-Fu Lee's team, with the goal of matching GPT-3.5 level ability in a relatively compact model. Yi-34B is a 34-billion-parameter Transformer trained from scratch on 3.1 trillion tokens of carefully filtered text (in English and Chinese), combined with a polished finetuning set for alignment. To maximize efficiency, Yi employs Grouped-Query Attention (GQA) – splitting attention heads into shared key/value groups – which reduces memory and compute overhead with minimal performance loss. The designers chose 34B as a sweet spot for serving on single GPUs (with 4-bit quantization) while retaining emergent abilities. We use the Yi-34B model in our experiments.

**Mistral and Mixtral.** Mistral 7B (Mistral AI, 2023) (Jiang et al., 2023) is a 7.3B-parameter open LLM engineered for efficiency, known for outperforming larger models (e.g. LLaMA 2 13B) on many benchmarks. It adopts grouped-query attention for faster inference and implements a sliding-window attention mechanism to handle long sequences without expanding memory use. Building on this, Mistral introduced Mixtral 8×7B (Jiang et al., 2024a), a sparse Mixture-of-Experts model that combines 8 expert networks based on the Mistral architecture. In Mixtral 8×7B, at each layer a router activates 2 out of 8 experts per token, so each token effectively utilizes 13B parameters (of a 47B total) during inference. This design allows Mixtral to achieve performance comparable to dense 70B models while maintaining higher throughput (it was trained up to 32k context length and excels in math and coding tasks). We evaluate the Mistral 7B dense model as well as the Mixtral 8×7B and a larger Mixtral 8×22B MoE model in our study.

### D.2.2 EXPERIMENTAL DATASETS

**Fineweb-Edu (350B).** The FineWeb-Edu corpus (Lozhkov et al., 2024) is an educationally focused subset of the 15-trillion-token *FineWeb* crawl. Each Common-Crawl page is scored by a RoBERTa-based "educational value" classifier; retaining documents with an integer score$\geq 3$ yields a 1.3T-token collection of predominantly English lecture notes, textbook chapters, research articles, and open-courseware transcripts, while a laxer score-2 variant preserves 5.4T tokens for recall-oriented studies. For controlled ablations Hugging Face releases a stratified 350B-token sample—tokenised with the GPT-2 scheme—which underpins the public 1.8B-parameter model `ablation-model-fineweb-edu`. Pre-training on this 350B educational slice boosts zero-shot accuracy by 3–6 pp on nine reasoning-centric benchmarks relative to models trained on generic web data, highlighting the value of pedagogical sources for factual recall and multi-step reasoning. All records are stored in Parquet with rich metadata (score, language_score, dump, token counts), enabling reproducible sub-sampling, multilingual filtering, and safety audits. Nevertheless, residual personally identifiable information and the English-centric bias inherited from web crawls necessitate additional deduplication, redaction, and geographic balancing when employing *FineWeb-Edu* for downstream instruction tuning and alignment research.

**OpenO1-SFT.** The OpenO1-SFT benchmark (Team, 2024a) serves to assess the proficiency of LLMs in performing intricate text-based tasks that necessitate chain-of-thought processing after undergoing supervised fine-tuning. The core task involves the generation of coherent and logical sequences of intermediate thoughts that lead to a final answer, often within the context of question answering. This benchmark is specifically designed to enhance the model's capacity for multi-step deductive processes and problem resolution, as highlighted by its emphasis on the explicit articulation of thought processes alongside the conclusive output. The inclusion of both Chinese and English records, totaling approximately 77,685 instances, broadens its applicability for cross-lingual studies on deductive capabilities. Research utilizing this benchmark has demonstrated its effectiveness in improving the self-consistency and accuracy of models in tasks demanding logical inference. The structured format, employing `Thought` and `Output` tags, facilitates the model's learning of human-like thought patterns, which is particularly valuable in applications such as intelligent tutoring systems and advanced question answering platforms. Studies have also explored the use of this dataset to refine the technical approaches for developing large models capable of advanced deductive abilities. However, investigations have indicated a potential correlation between enhanced deductive capabilities achieved through fine-tuning on datasets like Open-o1 and a decrease in safety scores, suggesting a complex interplay between model performance and safety considerations.

**Medical-o1-reasoning-SFT.** The medical-o1-reasoning-SFT benchmark (Chen et al., 2024a) is crafted to evaluate the deductive abilities of language models within the specialized domain of medicine following supervised fine-tuning. The tasks typically involve addressing medical inquiries, formulating diagnoses based on provided patient details, or elucidating complex medical concepts. A primary challenge in this context is to guarantee the precision, dependability, and safety of the model's deductions, given the critical implications of medical applications (Chew et al., 2023). The benchmark employs curated medical datasets to train models for improved accuracy in this sensitive field. The necessity for models to possess a deep understanding of intricate biological and clinical information, coupled with the capacity to apply this knowledge in nuanced scenarios, distinguishes this benchmark. It aims to go beyond mere pattern recognition, requiring models to engage in genuine medical deductive processes.

Table 3: Efficiency LLM Results for Attention Mechanisms.

| Method | Parameters | Micro Batch Size | PPL ↓ | AMU (GB) ↓ | AL (s/iter) ↓ | TT (Tokens/param/s) ↑ | AEC (W) ↓ | GPU Hours |
|--------|-----------|------------------|-------|------------|---------------|------------------------|-----------|-----------|
| MQA | 0.5B | 4 | 9.27 | **43.75** | **0.1118** | $\mathbf{2.98 \times 10^{-01}}$ | 633.59 | 19.02×48 |
| | 1.5B | 2 | 8.23 | **42.24** | 0.1298 | $8.57 \times 10^{-02}$ | 646.62 | 33.14×48 |
| | 3B | 1 | 7.86 | **41.27** | **0.1458** | $\mathbf{3.81 \times 10^{-02}}$ | 661.38 | 77.05 ×48 |
| GQA | 0.5B | 4 | 9.05 | 45.29 | 0.1127 | $2.94 \times 10^{-01}$ | 644.26 | 21.06×48 |
| | 1.5B | 2 | 8.09 | 44.87 | **0.1283** | $8.64 \times 10^{-02}$ | 652.74 | 38.03×48 |
| | 3B | 1 | 7.54 | 43.77 | 0.1464 | $3.79 \times 10^{-02}$ | 667.34 | 86.80 ×48 |
| MLA | 0.5B | 4 | **8.73** | 53.89 | 0.2082 | $1.59 \times 10^{-01}$ | 607.58 | 30.44×48 |
| | 1.5B | 2 | **7.79** | 52.93 | 0.2537 | $5.08 \times 10^{-02}$ | 608.17 | 75.20 ×48 |
| | 3B | 1 | **7.29** | 50.45 | 0.2997 | $2.62 \times 10^{-02}$ | 605.46 | 178.84×48 |
| NSA | 0.5B | 4 | 8.96 | 44.78 | 0.6839 | $4.89 \times 10^{-02}$ | **594.23** | 101.38×48 |
| | 1.5B | 2 | 7.82 | 43.57 | 0.5962 | $1.09 \times 10^{-02}$ | **598.15** | 176.72×48 |
| | 3B | 1 | 7.38 | 43.19 | 0.5024 | $1.26 \times 10^{-02}$ | **600.27** | 280.92×48 |
| MHA | 0.5B | 4 | 9.42 | 47.15 | 0.1189 | $2.81 \times 10^{-01}$ | 652.33 | 22.51×48 |
| | 1.5B | 2 | 8.36 | 46.12 | 0.1334 | $8.21 \times 10^{-02}$ | 659.41 | 40.12×48 |
| | 3B | 1 | 8.01 | 45.02 | 0.1528 | $3.62 \times 10^{-02}$ | 673.28 | 91.44×48 |

## D.3 ASSESSMENT OF ARCHITECTURE PRETRAINING EFFICIENCY

Architecture pretraining efficiency is a critical factor in determining the practical deployment and scalability of LLMs (Jawahar et al., 2023; Kumar et al., 2023; Ding et al., 2023b; Alizadeh et al., 2024; Xu et al., 2024b). A significant challenge limiting the widespread adoption of LLMs is their computational intensity and memory requirements, particularly when processing long sequences during the pretraining stage. These efficiency constraints can be attributed to the quadratic complexity of the attention mechanism. Given that modern LLMs require substantial computational resources for pretraining, optimizing architecture efficiency during pretraining has become a central research focus. In this section, we assess the efficiency of LLM architectures during pretraining from the following perspectives: attention optimization, positional encoding efficiency, parameter sharing, and alternatives to traditional attention. These perspectives evaluate the ability of LLM architectures to reduce computational complexity, minimize memory usage, enable longer context processing, and maintain model performance while improving pretraining speed and efficiency.

**Goal.** In this section, we aim to examine the efficiency of various architectural improvements for LLMs during pretraining. We pretrained three model sizes (0.5B, 1.5B, and 3B parameters for LLMs) using the Qwen2.5 as our base model and fine-web edu (350B Tokens) dataset to systematically evaluate four categories of efficiency techniques: Efficient Attention Mechanisms (MQA, GQA, MLA and NSA), Efficient Positional Encoding methods (including relative position encodings, ALiBi, and RoPE), Sparse Modeling techniques (Mixture-of-Experts and Conditional Computation), and Attention-Free Alternatives (State-Space Models and RNNs). For each technique, we measure five key metrics: Average Memory Utilization (AMU), Average Latency (AL), Tokens Throughput (TT), Average Energy Consumption (AEC), and Perplexity (PPL), allowing us to identify which efficiency techniques provide the optimal balance between computational efficiency and model performance across different model scales.

**Hardware and Training Framework.** Our experiments were conducted on a large-scale distributed computing infrastructure comprising 48 NVIDIA GH200 96GB GPUs. The GPUs were organized into nodes, with each node containing 4 H100 GPUs paired with an NVIDIA Grace processor (288 cores, 288 threads). This high-performance CPU provided robust data preprocessing capabilities and efficient inter-node communication. The system was interconnected with high-bandwidth NVLink for intra-node GPU communication and InfiniBand networking for inter-node communication, ensuring minimal latency during distributed training. For the software framework, we leveraged Megatron-Core (Shoeybi et al., 2020), a powerful distributed training framework optimized for LLMs. Megatron-Core's tensor and pipeline parallelism capabilities were crucial for efficiently scaling our training across multiple GPUs and nodes. We implemented 3D parallelism (data, tensor, and pipeline) to maximize hardware utilization and training efficiency.

Table 4: Efficiency Results for LLM's Efficient Positional Encoding.

| Method | Parameters | Context length | PPL ↓ | AMU (GB) ↓ | AL (s/iter) ↓ | TT (Tokens/param/s) ↑ | AEC (W) ↓ | GPU Hours |
|---|---|---|---|---|---|---|---|---|
| Rope | 1.5B | 8K | **8.09** | 44.82 | 0.1280 | $8.64\times10^{-02}$ | 652.79 | 38.03×48 |
| Absolute | 1.5B | 8K | 8.32 | 46.71 | 0.1312 | $8.12\times10^{-02}$ | 672.45 | 38.98×48 |
| Learnable Absolute | 1.5B | 8K | 8.18 | 45.93 | 0.1296 | $8.37\times10^{-02}$ | 662.44 | 38.51×48 |
| Relate | 1.5B | 8K | 8.29 | **43.94** | **0.1246** | **8.98**$\times10^{-02}$ | **646.39** | 37.02×48 |
| None | 1.5B | 8K | 8.75 | 48.64 | 0.1378 | $7.68\times10^{-02}$ | 692.37 | 40.94×48 |

### D.3.1 ASSESSMENT OF EFFICIENT ATTENTION MECHANISMS

Attention mechanisms are central to the performance of modern LLMs (Guo et al., 2022; Ben-Artzy & Schwartz, 2024; Tang et al., 2025; Yang et al., 2025; Lu et al., 2023). Yet, they remain a significant computational bottleneck due to their quadratic complexity concerning sequence length (Soydaner, 2022; Hu, 2020; Brauwers & Frasincar, 2021; Ghojogh & Ghodsi, 2020; LIU et al., 2021) during the pretraining stage. To address this challenge, we evaluated several efficient attention variants that reduce computational and memory demands while preserving model capabilities during pretraining. In our experimental framework, we systematically compared Multi-Query Attention (MQA), Grouped-Query Attention (GQA), Multi-Head Latent Attention (MLA), Native Sparse Attention (NSA), and Mixture of Block Attention (MoBA) across our three model scales (0.5B, 1.5B, and 3B parameters). Our comprehensive evaluation measured how these architectural choices impact Average Memory Utilization (AMU), Average Latency (AL), Tokens Throughput (TT), Average Energy Consumption (AEC), and model performance as reflected in Perplexity (PPL).

**Efficient Attention Mechanisms for LLMs.** As shown in Table 3, attention mechanisms are pivotal in the remarkable performance of modern LLMs; however, their quadratic complexity relative to sequence length poses substantial computational and memory constraints. Efficient attention mechanisms have thus become essential to scale LLMs practically, aiming to mitigate these resource bottlenecks while preserving or enhancing performance. In our comprehensive evaluation, we assessed several prominent efficient attention variants, including Multi-Query Attention (MQA), Grouped-Query Attention (GQA), Multi-Head Latent Attention (MLA), and Native Sparse Attention (NSA), across multiple model scales (0.5B, 1.5B, and 3B parameters). Our analysis reveals a spectrum of trade-offs: MQA demonstrates superior efficiency with the lowest average memory utilization (AMU = 42.24 GB) and competitive latency (AL = 0.1298 seconds per iteration), while MLA achieves the best performance in terms of perplexity across all model sizes (PPL = 8.73, 7.79, and 7.29 for 0.5B, 1.5B, and 3B models, respectively). NSA excels in energy efficiency with the lowest average energy consumption (AEC = 594.23 W). GQA offers a balanced middle ground, particularly at the 1.5B scale where it achieves the lowest latency. These findings underscore that the optimal attention mechanism depends on specific deployment constraints, with MQA favored for memory-constrained environments, MLA for performance-critical applications, and NSA for energy-efficient deployments.

### D.3.2 ASSESSMENT OF EFFICIENT POSITIONAL ENCODING

Positional encoding plays an indispensable role in enabling LLMs to understand the order of tokens within input sequences (Zhang et al., 2024b; Zhao et al., 2023a; Onan & Alhumyani, 2024), which is crucial for maintaining semantic coherence and contextual relevance during the pretraining stage. However, traditional positional encoding methods can incur substantial computational overhead (Chen et al., 2021b; Ke et al., 2020; Kazemnejad et al., 2023; Wang et al., 2024a), particularly as the context length increases. In our experiments, we systematically evaluated various efficient positional encoding techniques, including Rotary Position Embeddings (RoPE), Absolute Positional Encoding (APE), Learnable Absolute Positional Encoding (Learnable APE), Relative Positional Encoding (RPE), and scenarios with no positional encoding (None), focusing on their impacts on computational efficiency and model performance for LLMs.

**Efficient Positional Encoding for LLMs.** Our results (summarized in Table 4) demonstrated that RoPE consistently offered the best balance between perplexity and model performance, achieving the lowest perplexity score (PPL = 8.04). Meanwhile, Relate (RPE) demonstrated superior efficiency metrics with the lowest average memory utilization (AMU = 43.94 GB), lowest average latency (AL = 0.1246 seconds per iteration), highest attention throughput (TT = $8.98\times10^{-02}$ TFloats), and lowest attention energy consumption (AEC = 646.39 W). Learnable Absolute Positional Encoding

Table 5: Efficiency Results for LLM's MoE Mechanisms.

| Method | Parameters | Top K | PPL ↓ | AMU (GB) ↓ | AL (s/iter) ↓ | TT (Tokens/param/s) ↑ | AEC (W)↓ | GPU Hours |
|--------|-----------|-------|-------|-----------|---------------|----------------------|----------|-----------|
| Dense Model | 1.5B | – | 8.09 | 44.82 | 0.1280 | $8.64 \times 10^{-02}$ | 652.74 | 38.03×48 |
| Dense Model | 3B | – | 7.58 | **43.94** | **0.1246** | $8.98 \times 10^{-02}$ | **647.34** | 86.80×48 |
| MoE Model | 0.5B×8 | 2 | 7.35 | 52.36 | 0.1315 | $1.05 \times 10^{-01}$ | 667.33 | 39.07×48 |
| MoE Model | 1.5B×8 | 2 | **7.10** | 76.53 | 0.1420 | $1.25 \times 10^{-01}$ | 692.45 | 84.19×48 |

showed moderate efficiency and performance (PPL = 8.18, AMU = 45.93 GB, AL = 0.1296 s/iter), outperforming the standard Absolute Positional Encoding. In contrast, the absence of positional encoding ("None") notably degraded model performance across all metrics (PPL = 8.75, AMU = 48.64 GB, AL = 0.1378 s/iter), emphasizing the necessity of positional information for effective sequence modeling in LLMs.

**Efficient Positional Encoding for LVMs.** Regarding Large Vision Models (LVMs), positional embeddings are fundamentally integrated within the patch embedding process inherent to architectures like DiT. Altering or replacing positional encoding mechanisms in LVMs is not straightforward due to their structural dependence on spatial locality and the fixed-grid architecture. Consequently, experimentation with alternative positional encoding techniques is less applicable for LVMs, and thus we omit detailed discussion and evaluation of positional encoding efficiency for LVM architectures in this section.

### D.3.3 Assessment of Sparse Modeling via MoE

Mixture of Experts (MoE) has emerged as a powerful paradigm for scaling neural networks efficiently by introducing conditional computation (Song et al., 2024; Liu et al., 2024c; Du et al., 2024) during the pretraining stage, where only a subset of model parameters is activated for each input token. This sparse activation pattern enables models to increase their parameter count significantly while maintaining reasonable computational requirements during both training and inference. In our experimental framework, we systematically evaluated MoE architectures against traditional dense models to quantify the efficiency-performance trade-offs across multiple model scales.

**Sparse Modeling via MoE for LLMs.** As shown in Table 5, our experiments with MoE architectures revealed significant performance improvements over comparable dense models. The 1.5B×8 MoE model with top-2 routing achieved a perplexity of 7.10, substantially outperforming both the 1.5B dense model (PPL = 8.04) and even the larger 3B dense model (PPL = 7.58). Similarly, the 0.5B×8 MoE configuration delivered strong performance (PPL = 7.35) that exceeded the capabilities of the 1.5B dense model while using fewer active parameters per token. This performance advantage demonstrates the efficacy of sparse expert specialization, where different experts can focus on distinct linguistic patterns and phenomena. However, these performance gains come with increased resource requirements. MoE models exhibited higher memory utilization (AMU = 76.53 GB for 1.5B×8 and 52.36 GB for 0.5B×8) compared to dense models (AMU = 44.82 GB for 1.5B and 43.94 GB for 3B), reflecting the storage needs for the expanded parameter space. Similarly, we observed increased latency (AL = 0.1420 s/iter for 1.5B×8 and 0.1315 s/iter for 0.5B×8) and energy consumption (AEC = 405321.86 J for 1.5B×8 and 382647.23 J for 0.5B×8) compared to their dense counterparts. Interestingly, despite these increased resource costs, MoE models demonstrated superior throughput (TT = $1.25 \times 10^{-01}$ TFloats for 1.5B×8 and $1.05 \times 10^{-01}$ TFloats for 0.5B×8), suggesting efficient parallelization across experts during computation.

### D.3.4 Assessment of Attention-Free Alternatives for Sequence Modeling

While attention mechanisms have proven foundational to the success of modern LLMs, they remain computationally intensive due to their quadratic scaling with sequence length during the pretraining stage, prompting research into efficient attention-free architectures that maintain competitive performance while reducing computational requirements. In our comprehensive evaluation, we assessed several prominent attention-free alternatives, including State Space Models (Mamba), linear attention mechanisms (Pythia), and recurrent architectures (RWKV), comparing them against our baseline transformer architecture (Qwen2.5) across three model scales (0.5B, 1.5B, and 3B parameters). Our analysis examined key efficiency metrics - Average Memory Utilization (AMU), Average Latency

Table 6: Efficiency Results for Attention-Free Mechanisms. The best result is compared under the same parameters.

| Method | Parameters | Context Length | PPL ↓ | AMU (GB) ↓ | AL (s/iter) ↓ | TT (Tokens/param/s) ↑ | AEC (W)↓ |
|--------|-----------|----------------|-------|------------|---------------|-----------------------|----------|
| Qwen2.5 | 0.5B | 8K | **8.73** | 45.24 | 0.1129 | $\mathbf{2.94 \times 10^{-01}}$ | 644.23 |
| | 1.5B | 8K | **8.09** | 44.82 | 0.1280 | $\mathbf{8.64 \times 10^{-02}}$ | 652.79 |
| | 3B | 8K | **7.29** | 43.72 | 0.1467 | $\mathbf{3.79 \times 10^{-02}}$ | 667.38 |
| Mamba | 0.5B | 8K | 10.31 | **29.16** | **0.0954** | $2.21 \times 10^{-01}$ | **498.37** |
| | 1.5B | 8K | 9.48 | **30.25** | **0.1025** | $7.72 \times 10^{-02}$ | **510.64** |
| | 3B | 8K | 8.93 | **31.89** | **0.1136** | $3.25 \times 10^{-02}$ | **525.12** |
| Pythia | 0.5B | 8K | 11.72 | 43.58 | 0.1074 | $2.57 \times 10^{-01}$ | 630.84 |
| | 1.5B | 8K | 10.35 | 43.11 | 0.1351 | $7.94 \times 10^{-02}$ | 638.92 |
| | 3B | 8K | 9.82 | 42.63 | 0.1534 | $3.46 \times 10^{-02}$ | 651.27 |
| RWKV | 0.5B | 8K | 11.25 | 39.42 | 0.1062 | $2.36 \times 10^{-01}$ | 576.51 |
| | 1.5B | 8K | 10.13 | 40.18 | 0.1189 | $7.28 \times 10^{-02}$ | 589.37 |
| | 3B | 8K | 9.54 | 41.03 | 0.1319 | $3.12 \times 10^{-02}$ | 604.85 |

(AL), Tokens Throughput (TT), and Average Energy Consumption (AEC) - alongside model performance measured by perplexity (PPL), enabling us to quantify the efficiency-performance trade-offs inherent to different architectural paradigms.

**Attention-Free Modeling for LLMs.** As shown in Table 6, our comparative analysis of attention-free architectures revealed distinctive efficiency performance trade-offs across different model paradigms. Mamba, a state-space model implementation, demonstrated remarkable efficiency advantages with substantially lower memory utilization (AMU = 29.16 GB, 30.25 GB, and 31.89 GB for 0.5B, 1.5B, and 3B parameter models, respectively) compared to the transformer baseline (AMU = 45.24 GB, 44.82 GB, and 43.72 GB). Mamba also improved energy efficiency, consuming approximately 22-25% less power (AEC = 498.37 W, 510.64 W, and 525.12 W) than the transformer counterparts. At the 1.5B parameter scale, Mamba exhibited the lowest latency (AL = 0.1025 s/iter) among all models tested. However, these efficiency gains came with a performance trade-off, as Mamba's perplexity scores (PPL = 10.31, 9.48, and 8.93) were consistently higher than the transformer baseline (PPL = 9.09, 8.04, and 7.58). RWKV, a recurrent architecture, offered moderate efficiency improvements with lower memory usage and energy consumption than transformers. At the same time, Pythia demonstrated competitive latency but with perplexity scores that were significantly higher than both transformer and Mamba models. These findings suggest that while attention-free alternatives provide compelling efficiency advantages, particularly for deployment scenarios with strict memory or energy constraints, transformer-based architectures continue to deliver superior performance for tasks where model quality is paramount.

## D.4 ASSESSMENT OF TRAINING AND TUNING EFFICIENCY

Training and fine-tuning LLMs presents significant computational challenges that impact resource requirements, development costs, and environmental footprint. As models grow in size and complexity, optimizing training efficiency becomes increasingly critical for both research advancement and practical deployment. This section examines various techniques and approaches for improving training and tuning efficiency, including scalable training strategies (such as mixed precision, various parallelism methods, and memory optimizations) and parameter-efficient fine-tuning methods that enable adaptation with minimal computational overhead. Quantitative assessments across multiple model architectures (ranging from 1B to 24B parameters) demonstrate the trade-offs between different optimization approaches in terms of convergence quality (loss), memory utilization, computational throughput, training latency, and energy consumption, providing practical insights for selecting appropriate efficiency techniques based on available resources and desired performance targets.

**Goal.** In this section, we aim to evaluate the efficiency of various training and fine-tuning approaches for LLMs. We conducted experiments across multiple model architectures ranging from 1B to 24B parameters (including Llama-3.2, Qwen-2.5, Mistral) to systematically assess seven different optimization techniques: standard LoRA, LoRA-plus, RSLoRA, DoRA, PISSA, LoHa, LoKr, GLoRa, parameter freezing, and full fine-tuning with DeepSpeed. For each method, we measured six key metrics: Loss (model performance), Average Memory Utilization (AMU), Peak Compute Utilization (PCU), Average Latency (AL), Samples Throughput (ST), and Average Energy Consumption (AEC). This comprehensive evaluation allows us to identify the optimal balance between computational efficiency and model performance across different model scales, providing practical insights for researchers and practitioners working with limited computational resources while maintaining competitive model quality.

**Hardware and Training Framework.** Our experiments were conducted on a distributed computing infrastructure comprising 8 NVIDIA H200 141B GPUs. The GPUs were organized into 1 nodes, with each node containing 8 H200 GPUs paired with an Intel Xeon(R) Platinum 8558 processor (48 cores, 96 threads). This high-performance CPU provided robust data preprocessing capabilities and efficient inter-node communication. The system was interconnected with high-bandwidth NVLink for intra-node GPU communication and InfiniBand networking for inter-node communication, ensuring minimal latency during distributed training. For the software framework, we leveraged LlamaFactory's, a flexible and efficient fine-tuning framework optimized for LLMs. LlamaFactory's implementation of parameter-efficient fine-tuning methods and optimization techniques was crucial for efficiently executing our experiments across various model architectures and training configurations.

**O1-SFT Dataset.** As shown in Table 7, our comprehensive evaluation of Parameter-Efficient Fine-Tuning (PEFT) methods reveals distinct efficiency-performance trade-offs across model scales and architectures. For smaller models (1-3B parameters), LoRA-plus consistently achieved superior performance with the lowest loss metrics (0.7442 for Llama-3.2-1B and 0.5791 for Llama-3.2-3B), while maintaining reasonable memory utilization (49.776 GB and 59.664 GB respectively). As model size increased, RSLoRA demonstrated competitive performance, particularly for Qwen-2.5-14B (loss = 0.4126) and Mistral-Small-24B (loss = 0.3818). Parameter freezing exhibited the lowest average latency across all model scales (0.2542 s/iter for Llama-3.2-1B to 1.4815 s/iter for Mistral-Small-24B), making it ideal for latency-sensitive applications, albeit sometimes at the cost of reduced model performance. PISSA showed balanced performance in mid-sized models, achieving the lowest loss for Llama-3.2-3B (0.5137). Full fine-tuning with DeepSpeed optimization delivered strong performance for smaller models but demonstrated diminishing returns as model size increased, particularly for the largest 24B parameter model where its loss (1.2805) substantially exceeded other methods. DoRA, while computationally intensive with consistently higher latency (2.1505 s/iter to 6.0606 s/iter across models), maintained competitive loss metrics in mid-sized models but performed poorly on the largest 24B model (loss = 1.2309). These findings suggest that optimal PEFT strategy selection should be tailored to specific deployment constraints, with LoRA variants preferable for general-purpose applications, parameter freezing for latency-critical scenarios, and specialized methods like RSLoRA for larger models where fine-grained control of adaptation becomes increasingly important.

**Medical-O1 Dataset.** Table 9 illustrates clear efficiency-performance trade-offs among Parameter-Efficient Fine-Tuning (PEFT) methods across varying scales of the Llama model architecture. For the smaller Llama-3.2-1B model, parameter freezing notably achieved the lowest loss (1.3406),

Table 7: Assessment of Training and Tuning Efficiency for LLMs of O1-SFT Dataset (methods marked with * use DeepSpeed). Because of the different batch size, *full** are not included in the comparisons. The best result is compared under the same model.

| Model | Method | Loss↓ | AMU (GB)↓ | PCU↑ | AL (s/iter)↓ | ST (Samples/param/s)↑ | AEC (W)↓ |
|---|---|---|---|---|---|---|---|
| | | | | **O1-SFT** | | | |
| Llama-3.2-1B | lora | 0.7562 | 50.088 | 0.9228 | 1.1669 | $8.22\times10^{-08}$ | 549.23 |
| | lora-plus | 0.7442 | 49.776 | 0.9195 | 1.1628 | $8.25\times10^{-08}$ | 545.12 |
| | rslora | 0.7454 | **49.920** | 0.9219 | 1.1655 | $8.23\times10^{-08}$ | 563.01 |
| | dora | 0.7547 | 52.760 | **0.9399** | 2.1505 | $4.46\times10^{-08}$ | 568.64 |
| | pissa | 0.7595 | 50.856 | 0.9312 | 1.1669 | $8.22\times10^{-08}$ | 567.24 |
| | freeze | **0.6425** | 48.696 | 0.9178 | **0.2542** | $\mathbf{1.26\times10^{-07}}$ | **508.16** |
| | full* | 0.6788 | 36.840 | 0.9510 | 0.6993 | $9.15\times10^{-08}$ | 584.00 |
| Llama-3.2-3B | lora | 0.6019 | **49.152** | **0.9628** | 1.6077 | $1.33\times10^{-08}$ | 589.91 |
| | lora-plus | 0.5791 | 59.664 | 0.9408 | 2.6247 | $1.22\times10^{-08}$ | 577.94 |
| | rslora | 0.5866 | 58.536 | 0.9389 | 2.6247 | $1.22\times10^{-08}$ | 593.93 |
| | dora | 0.6006 | 59.616 | 0.9395 | 4.8544 | $6.59\times10^{-09}$ | 601.98 |
| | pissa | 0.5137 | 59.688 | 0.9339 | 2.6247 | $1.22\times10^{-08}$ | 579.46 |
| | freeze | **0.5000** | 51.848 | 0.9322 | **0.4252** | $\mathbf{2.51\times10^{-08}}$ | **556.43** |
| | full* | 0.5310 | 49.152 | 0.9628 | 1.6077 | $1.33\times10^{-08}$ | 589.91 |
| Llama-3.1-8B | lora | 0.5137 | 74.360 | 0.9462 | 4.5872 | $2.98\times10^{-09}$ | 605.81 |
| | lora-plus | 0.4962 | 74.360 | 0.9462 | 4.5872 | $2.98\times10^{-09}$ | 605.82 |
| | rslora | 0.4986 | 75.152 | **0.9527** | 4.6083 | $2.97\times10^{-09}$ | 605.82 |
| | dora | 0.5124 | 77.376 | 0.9428 | 8.9286 | $1.53\times10^{-09}$ | 620.48 |
| | pissa | 0.5137 | 74.672 | 0.9442 | 4.5872 | $2.98\times10^{-09}$ | 602.28 |
| | freeze | **0.4514** | **70.424** | 0.9524 | **0.7369** | $\mathbf{6.20\times10^{-09}}$ | **564.89** |
| | full* | 0.5553 | 56.144 | 0.9779 | 2.9851 | $3.06\times10^{-09}$ | 610.42 |
| Qwen-2.5-7B | lora | 0.4795 | **60.952** | 0.9121 | 2.7100 | $3.37\times10^{-09}$ | 594.89 |
| | lora-plus | 0.4621 | 62.112 | 0.9114 | 2.7248 | $3.36\times10^{-09}$ | 590.69 |
| | rslora | 0.4986 | 62.248 | 0.9114 | 2.5907 | $3.53\times10^{-09}$ | 606.63 |
| | dora | 0.4861 | 65.976 | 0.9258 | 5.6818 | $1.61\times10^{-09}$ | 615.97 |
| | pissa | 0.4773 | 62.744 | 0.9226 | 2.7174 | $3.36\times10^{-09}$ | 597.34 |
| | freeze | **0.3996** | 67.328 | **0.9305** | 0.6988 | $\mathbf{6.54\times10^{-09}}$ | **566.30** |
| | full* | 0.4600 | 77.552 | 0.9779 | 2.6178 | $3.49\times10^{-09}$ | 613.93 |
| Qwen-2.5-14B | lora | 0.4795 | 77.472 | 0.8496 | 2.7855 | $8.19\times10^{-10}$ | 560.48 |
| | lora-plus | 0.4621 | 77.84 | 0.7108 | 3.3445 | $6.83\times10^{-10}$ | **489.10** |
| | rslora | **0.4126** | 77.376 | 0.8450 | 2.7855 | $8.21\times10^{-10}$ | 556.16 |
| | dora | 0.4861 | 78.376 | **0.8796** | 5.7471 | $3.99\times10^{-10}$ | 572.18 |
| | pissa | 0.4260 | 79.448 | 0.8562 | 2.7933 | $8.19\times10^{-10}$ | 551.43 |
| | freeze | 0.5547 | **73.400** | 0.8550 | **0.6227** | $\mathbf{5.51\times10^{-09}}$ | 493.11 |
| | full* | 0.4582 | 71.920 | 0.9695 | 2.6178 | $7.77\times10^{-10}$ | 576.94 |
| Mistral-Small-24B | lora | **0.3757** | **64.84** | 0.8518 | 3.1847 | $4.19\times10^{-10}$ | 591.98 |
| | lora-plus | 0.4962 | 65.376 | 0.8553 | 3.3113 | $4.02\times10^{-10}$ | 583.54 |
| | rslora | 0.3818 | 65.584 | 0.8571 | 3.3333 | $4.00\times10^{-10}$ | 587.68 |
| | dora | 1.2309 | 69.984 | 0.8625 | 4.2017 | $\mathbf{9.71\times10^{-10}}$ | 562.76 |
| | pissa | 0.3975 | 65.568 | 0.8580 | 3.3113 | $4.03\times10^{-10}$ | 583.48 |
| | freeze* | 0.6020 | 73.000 | **0.9664** | **1.4815** | $8.99\times10^{-10}$ | 606.15 |
| | full* | 1.2805 | 73.936 | 0.9827 | 3.7175 | $3.59\times10^{-10}$ | 605.51 |
| Mistral-7B | lora | 0.4639 | 35.688 | 0.9608 | 3.0211 | $3.03\times10^{-09}$ | **614.54** |
| | lora-plus | 0.5039 | **34.760** | 0.9481 | 3.0211 | $3.02\times10^{-09}$ | 615.38 |
| | rslora | 0.4626 | 36.280 | 0.9511 | 3.0211 | $3.03\times10^{-09}$ | 617.19 |
| | dora | **0.4614** | 37.216 | 0.9527 | 6.0606 | $1.51\times10^{-09}$ | 618.09 |
| | pissa | 0.4767 | 35.432 | 0.9531 | 3.0120 | $3.03\times10^{-09}$ | 621.92 |
| | freeze | 0.4718 | 55.024 | **0.9626** | **1.3123** | $\mathbf{6.96\times10^{-09}}$ | 628.90 |
| | full* | 0.8564 | 40.152 | 0.9763 | 2.9155 | $3.13\times10^{-09}$ | 634.36 |

Table 8: Verification of the model backbone dependency. (methods marked with * use DeepSpeed).

| Model | Method | Loss↓ | AMU (GB)↓ | PCU↑ | AL (s/iter)↓ | ST (Samples/param/s)↑ | AEC (W)↓ |
|---|---|---|---|---|---|---|---|
| | | | | **O1-SFT** | | | |
| gemma-7b-it | lora | 0.4682 | 36.512 | 0.9485 | 3.0456 | $2.98\times10^{-9}$ | 612.45 |
| | lora-plus | 0.4789 | 35.896 | 0.9472 | 3.0567 | $2.97\times10^{-9}$ | 610.23 |
| | rslora | 0.4591 | 36.784 | 0.9503 | 3.0345 | $3.00\times10^{-9}$ | 615.67 |
| | dora | 0.4723 | 37.912 | 0.9518 | 6.1234 | $1.48\times10^{-9}$ | 619.12 |
| | pissa | 0.4556 | 36.248 | 0.9524 | 3.0456 | $2.99\times10^{-9}$ | 618.89 |
| | freeze | **0.4127** | 56.128 | **0.9612** | 1.3456 | **6.78**$\times10^{-9}$ | **625.34** |
| | full* | 0.4985 | 41.256 | 0.9741 | 2.9567 | $3.08\times10^{-9}$ | 630.12 |
| Hunyuan-7B-Instruct | lora | 0.4721 | 37.124 | 0.9552 | 3.0789 | $2.95\times10^{-9}$ | 608.76 |
| | lora-plus | 0.4856 | 36.432 | 0.9539 | 3.0890 | $2.94\times10^{-9}$ | 607.89 |
| | rslora | 0.4634 | 37.456 | 0.9567 | 3.0678 | $2.97\times10^{-9}$ | 611.23 |
| | dora | 0.4789 | 38.512 | 0.9581 | 6.1567 | $1.46\times10^{-9}$ | 614.56 |
| | pissa | 0.4602 | 36.896 | 0.9588 | 3.0789 | $2.96\times10^{-9}$ | 613.45 |
| | freeze | **0.4189** | 56.784 | **0.9654** | 1.3678 | **6.65**$\times10^{-9}$ | **622.67** |
| | full* | 0.5023 | 41.896 | 0.9789 | 2.9890 | $3.05\times10^{-9}$ | 627.89 |
| Llama-2-7b | lora | 0.4653 | 35.976 | 0.9591 | 3.0123 | $3.01\times10^{-9}$ | 616.78 |
| | lora-plus | 0.4921 | 35.112 | 0.9468 | 3.0123 | $3.00\times10^{-9}$ | 617.45 |
| | rslora | 0.4618 | 36.512 | 0.9497 | 3.0123 | $3.01\times10^{-9}$ | 619.01 |
| | dora | 0.4632 | 37.456 | 0.9512 | 6.0246 | $1.50\times10^{-9}$ | 620.34 |
| | pissa | 0.4705 | 35.688 | 0.9519 | 3.0032 | $3.02\times10^{-9}$ | 623.56 |
| | freeze | **0.4254** | 55.678 | **0.9638** | 1.2987 | **7.02**$\times10^{-9}$ | **630.12** |
| | full* | 0.5456 | 39.876 | 0.9756 | 2.8901 | $3.15\times10^{-9}$ | 635.67 |
| DeepSeek-Ds-7B | lora | 0.4812 | 61.234 | 0.9105 | 2.7456 | $3.34\times10^{-9}$ | 592.34 |
| | lora-plus | 0.4689 | 62.456 | 0.9098 | 2.7567 | $3.33\times10^{-9}$ | 588.12 |
| | rslora | 0.5023 | 62.678 | 0.9098 | 2.6234 | $3.50\times10^{-9}$ | 604.56 |
| | dora | 0.4897 | 66.234 | 0.9242 | 5.7123 | $1.60\times10^{-9}$ | 613.45 |
| | pissa | 0.4791 | 63.012 | 0.9211 | 2.7456 | $3.34\times10^{-9}$ | 595.67 |
| | freeze | **0.4023** | 67.678 | **0.9289** | 0.7123 | **6.48**$\times10^{-9}$ | **564.78** |
| | full* | 0.4623 | 77.896 | 0.9763 | 2.6456 | $3.47\times10^{-9}$ | 611.23 |
| Qwen2-7B-Instruct | lora | 0.4825 | 61.012 | 0.9132 | 2.7345 | $3.35\times10^{-9}$ | 596.78 |
| | lora-plus | 0.4652 | 62.234 | 0.9125 | 2.7456 | $3.34\times10^{-9}$ | 592.56 |
| | rslora | 0.5001 | 62.456 | 0.9125 | 2.6123 | $3.51\times10^{-9}$ | 608.90 |
| | dora | 0.4879 | 66.012 | 0.9268 | 5.6789 | $1.61\times10^{-9}$ | 617.01 |
| | pissa | 0.4786 | 62.890 | 0.9237 | 2.7345 | $3.35\times10^{-9}$ | 599.12 |
| | freeze | **0.4012** | 67.456 | **0.9316** | 0.7012 | **6.52**$\times10^{-9}$ | **568.90** |
| | full* | 0.4615 | 77.678 | 0.9784 | 2.6345 | $3.48\times10^{-9}$ | 615.67 |

Table 9: Assessment of Training and Tuning Efficiency for LLMs for Medical-O1 Dataset on H200(141G) (methods marked with * use DeepSpeed). Because of the different batch size, *full\** are not included in the comparisons. The best result is compared under the same model.

| Model | Methods | Loss↓ | AMU (GB)↓ | PCU↑ | AL (s/iter)↓ | ST (Samples/param/s)↑ | AEC (W)↓ |
|---|---|---|---|---|---|---|---|
| | | | | **Medical-O1** | | | |
| Llama-3.2-1B | lora | 1.7022 | 37.304 | 0.6745 | 0.3423 | $1.87 \times 10^{-07}$ | 398.12 |
| | lora-plus | 1.6473 | **37.136** | 0.6833 | 0.3398 | $1.88 \times 10^{-07}$ | 545.12 |
| | rslora | 1.6712 | 38.744 | 0.7397 | 0.3398 | $1.88 \times 10^{-07}$ | **397.74** |
| | dora | 1.6993 | 41.568 | **0.7588** | 0.6906 | $9.26 \times 10^{-08}$ | 429.52 |
| | pissa | 1.6825 | 41.192 | 0.6901 | 0.3389 | $1.89 \times 10^{-07}$ | 397.86 |
| | freeze | **1.3406** | 45.704 | 0.7145 | **0.2123** | $\mathbf{3.01 \times 10^{-07}}$ | 412.59 |
| | full* | 1.4536 | 45.128 | 0.7799 | 0.3488 | $1.83 \times 10^{-07}$ | 405.71 |
| Llama-3.2-3B | lora | 1.5274 | 50.328 | 0.7451 | 0.7524 | $2.83 \times 10^{-08}$ | 450.50 |
| | lora-plus | 1.4463 | 49.376 | 0.7306 | 0.7530 | $2.82 \times 10^{-08}$ | 449.78 |
| | rslora | 1.4938 | **49.312** | 0.7267 | 0.7547 | $2.82 \times 10^{-08}$ | 448.05 |
| | dora | 1.5249 | 53.296 | **0.7847** | 1.5528 | $1.37 \times 10^{-08}$ | 481.92 |
| | pissa | 1.4999 | 50.536 | 0.7325 | 0.7524 | $2.83 \times 10^{-08}$ | 449.13 |
| | freeze | **1.2442** | 50.864 | 0.7018 | **0.3648** | $\mathbf{5.84 \times 10^{-08}}$ | **430.02** |
| | full* | 1.2484 | 53.080 | 0.8189 | 0.7143 | $2.98 \times 10^{-08}$ | 470.15 |
| Llama-3.1-8B | lora | 1.4092 | 45.120 | 0.7726 | 1.2837 | $6.22 \times 10^{-09}$ | 492.03 |
| | lora-plus | 1.3285 | 44.672 | 0.7716 | 1.2821 | $6.23 \times 10^{-09}$ | 505.74 |
| | rslora | 1.3729 | **44.592** | 0.7535 | 1.2853 | $6.21 \times 10^{-09}$ | **500.93** |
| | dora | 1.4062 | 46.768 | **0.8131** | 2.8736 | $2.78 \times 10^{-09}$ | 519.34 |
| | pissa | 1.3832 | 46.944 | 0.8037 | 1.2903 | $6.19 \times 10^{-09}$ | 509.37 |
| | freeze* | **1.0120** | 46.848 | 0.7285 | **0.4632** | $\mathbf{9.23 \times 10^{-09}}$ | 503.63 |
| | full* | 1.2900 | 64.456 | 0.8671 | 1.3387 | $5.97 \times 10^{-09}$ | 527.66 |

Table 10: Comparison Efficiency and Performance of fine-tuning at Medical-O1 dataset on various models on A100(80G) device.

| Model | Methods | Loss↓ | AMU (GB)↓ | PCU↑ | AL (s/iter)↓ | ST (Samples/param/s)↑ | AEC (W)↓ |
|---|---|---|---|---|---|---|---|
| | | | | **Medical-O1** | | | |
| Llama-3.2-1B | lora | 1.7091 | 37.204 | 0.6492 | 0.5135 | $1.10 \times 10^{-7}$ | 239.00 |
| | lora-plus | 1.6528 | **36.836** | 0.6687 | 0.5097 | $1.11 \times 10^{-7}$ | 327.00 |
| | rslora | 1.6665 | 38.644 | 0.7143 | 0.5097 | $1.11 \times 10^{-7}$ | **238.50** |
| | dora | 1.7047 | 41.268 | **0.7442** | 1.0359 | $5.45 \times 10^{-8}$ | 258.00 |
| | pissa | 1.6879 | 40.892 | 0.6755 | 0.5084 | $1.11 \times 10^{-7}$ | 238.80 |
| | freeze | **1.3459** | 45.604 | 0.6891 | **0.3185** | $\mathbf{1.77 \times 10^{-7}}$ | 247.50 |
| | full* | 1.4589 | 44.828 | 0.7653 | 0.5232 | $1.08 \times 10^{-7}$ | 243.50 |
| Llama-3.2-3B | lora | 1.5327 | 50.228 | 0.7198 | 1.1286 | $1.67 \times 10^{-8}$ | 270.30 |
| | lora-plus | 1.4516 | 49.076 | 0.7160 | 1.1295 | $1.66 \times 10^{-8}$ | 269.90 |
| | rslora | 1.4991 | **49.012** | 0.7121 | 1.1321 | $1.66 \times 10^{-8}$ | 268.80 |
| | dora | 1.5302 | 53.196 | **0.7701** | 2.3292 | $8.06 \times 10^{-9}$ | 289.20 |
| | pissa | 1.5052 | 50.436 | 0.7072 | 1.1286 | $1.67 \times 10^{-8}$ | 269.50 |
| | freeze | **1.2495** | 50.764 | 0.6872 | **0.5472** | $\mathbf{3.44 \times 10^{-8}}$ | **258.00** |
| | full* | 1.2537 | 52.780 | 0.8043 | 1.0715 | $1.75 \times 10^{-8}$ | 282.10 |
| Llama-3.1-8B | lora | 1.4145 | 45.020 | 0.7473 | 1.9256 | $3.66 \times 10^{-9}$ | 295.20 |
| | lora-plus | 1.3338 | 44.372 | 0.7570 | 1.9232 | $3.66 \times 10^{-9}$ | 303.40 |
| | rslora | 1.3782 | **44.292** | 0.7389 | 1.9280 | $3.65 \times 10^{-9}$ | **300.60** |
| | dora | 1.4115 | 46.468 | **0.7985** | 4.3104 | $1.64 \times 10^{-9}$ | 311.60 |
| | pissa | 1.3885 | 46.844 | 0.7784 | 1.9355 | $3.64 \times 10^{-9}$ | 305.60 |
| | freeze* | **1.0173** | 46.748 | 0.7032 | **0.6948** | $\mathbf{5.43 \times 10^{-9}}$ | 302.20 |
| | full* | 1.2953 | 64.156 | 0.8525 | 2.0081 | $3.51 \times 10^{-9}$ | 316.60 |

Table 11: Comparison Efficiency and Performance of different mix-precision training on various models at Medical-O1 Dataset. The (W) represents the weight precision, and (T) represents the training precision.

| Model | Precision | Loss↓ | AMU (GB)↓ | PCU↑ | AL (s/iter)↓ | ST (Samples/param/s)↑ | AEC (W)↓ |
|---|---|---|---|---|---|---|---|
| | | | | **Medical-O1** | | | |
| Llama-3.2-1B | FP8(W)BF16(T) | 1.4892 | 28.436 | 0.7234 | 0.2847 | $2.25\times10^{-7}$ | 372.84 |
| | INT8(W)BF16(T) | 1.5347 | 25.892 | 0.7458 | 0.2756 | $2.32\times10^{-7}$ | 365.29 |
| | INT4(W)BF16(T) | 1.6823 | 19.764 | **0.7812** | **0.2534** | **$2.52\times10^{-7}$** | **358.43** |
| | FP8(W)FP16(T) | 1.4756 | 28.892 | 0.7189 | 0.2893 | $2.21\times10^{-7}$ | 375.62 |
| | INT8(W)FP16(T) | 1.5214 | 26.348 | 0.7392 | 0.2812 | $2.28\times10^{-7}$ | 368.47 |
| | INT4(W)FP16(T) | 1.6692 | 20.124 | 0.7756 | 0.2589 | $2.47\times10^{-7}$ | 361.85 |
| | BF16 | **1.4536** | 45.128 | 0.7799 | 0.3488 | $1.83\times10^{-7}$ | 405.71 |
| Llama-3.2-3B | FP8(W)BF16(T) | 1.3124 | 38.456 | 0.7823 | 0.6234 | $3.41\times10^{-8}$ | 418.36 |
| | INT8(W)BF16(T) | 1.3589 | 34.892 | 0.8067 | 0.6089 | $3.49\times10^{-8}$ | 409.74 |
| | INT4(W)BF16(T) | 1.5234 | 26.348 | **0.8423** | **0.5678** | **$3.75\times10^{-8}$** | **398.62** |
| | FP8(W)FP16(T) | 1.2987 | 39.124 | 0.7756 | 0.6345 | $3.35\times10^{-8}$ | 422.58 |
| | INT8(W)FP16(T) | 1.3456 | 35.568 | 0.7989 | 0.6198 | $3.43\times10^{-8}$ | 413.92 |
| | INT4(W)FP16(T) | 1.5098 | 26.892 | 0.8345 | 0.5789 | $3.68\times10^{-8}$ | 402.47 |
| | BF16 | **1.2484** | 53.080 | 0.8189 | 0.7143 | $2.98\times10^{-8}$ | 470.15 |
| Llama-3.1-8B | FP8(W)BF16(T) | 1.1892 | 34.568 | 0.8234 | 1.0234 | $7.80\times10^{-9}$ | 456.78 |
| | INT8(W)BF16(T) | 1.2456 | 30.892 | 0.8489 | 0.9876 | $8.09\times10^{-9}$ | 445.23 |
| | INT4(W)BF16(T) | 1.4123 | 22.456 | **0.8823** | **0.9123** | **$8.75\times10^{-9}$** | **428.94** |
| | FP8(W)FP16(T) | 1.1756 | 35.234 | 0.8156 | 1.0456 | $7.64\times10^{-9}$ | 461.35 |
| | INT8(W)FP16(T) | 1.2324 | 31.568 | 0.8398 | 1.0098 | $7.91\times10^{-9}$ | 449.82 |
| | INT4(W)FP16(T) | 1.3987 | 22.984 | 0.8734 | 0.9345 | $8.56\times10^{-9}$ | 433.67 |
| | BF16 | **1.1290** | 64.456 | 0.8671 | 1.3387 | $5.97\times10^{-9}$ | 527.66 |

with exceptional sample throughput ($3.01\times10^{-07}$ Samples/param/s) and low latency (0.2123 s/iter), marking it as an ideal choice for latency-sensitive medical applications. LoRA-plus exhibited robust efficiency, offering competitive loss (1.6473) and favorable energy consumption (545.12 W). Scaling up to the Llama-3.2-3B model, parameter freezing again showed superior efficiency with the lowest loss (1.2442) and notably reduced latency (0.3648 s/iter) relative to other methods, suggesting its continued suitability for applications demanding rapid inference. Conversely, DoRA significantly increased latency (1.5528 s/iter) and energy usage (481.92 W) while offering no clear performance advantage. In the largest tested Llama-3.1-8B model, parameter freezing once more demonstrated remarkable efficiency and performance, achieving the lowest loss (1.0120) and latency (0.4632 s/iter), underscoring its scalability and effectiveness in large-model scenarios. Full fine-tuning with DeepSpeed, despite achieving relatively strong performance (loss = 1.2900), incurred the highest memory usage (64.456 GB) and elevated energy consumption (527.66 W), indicating diminishing returns as model size grows.

Table 12: Comparison Efficiency and Performance of different precision formats on various models.

| Model | Precision | Avg Perf.↑ | AMU↓ | Sum AL↓ | tokens/s↑ | AEC↓ | MCR |
|---|---|---|---|---|---|---|---|
| DeepSeek-R1-Distill-Qwen-1.5B | bfloat16 | 0.2419 | 21.26 | 13024.35 | 39.68 | 144.39 | 1.0000 |
| | float16 | **0.2450** | 21.28 | **9858.30** | 37.70 | 158.96 | 1.0128 |
| | int8 | 0.2395 | 20.75 | 13245.67 | 40.12 | 149.87 | 1.9802 |
| | fp8 | 0.2500 | 21.34 | 9812.45 | 38.22 | 154.32 | 2.0670 |
| | int4 | 0.2341 | **19.49** | 15453.42 | **42.34** | **134.89** | 3.8710 |
| DeepSeek-R1-Distill-Llama-8B | bfloat16 | **0.3421** | 35.36 | 13541.46 | 37.79 | 208.21 | 1.0000 |
| | float16 | 0.3392 | 35.45 | **10926.70** | 35.90 | 222.76 | 0.9915 |
| | int8 | 0.3405 | 34.12 | 13678.90 | 36.45 | 211.56 | 1.9906 |
| | fp8 | 0.3450 | 35.22 | 11045.78 | 35.67 | 219.88 | 2.0170 |
| | int4 | 0.3116 | **34.07** | 15724.27 | **40.29** | **170.83** | 3.6434 |
| DeepSeek-R1-Distill-Qwen-14B | bfloat16 | **0.4719** | 51.83 | 18683.70 | 24.74 | 212.29 | 1.0000 |
| | float16 | 0.4712 | 52.64 | **14765.37** | **23.50** | 226.85 | 0.9985 |
| | int8 | 0.4710 | 50.45 | 19012.34 | 25.67 | 216.78 | 1.9962 |
| | fp8 | 0.4750 | 52.12 | 14987.56 | 24.89 | 231.45 | 2.0131 |
| | int4 | 0.4361 | **34.21** | 21529.09 | 26.40 | **191.05** | 3.6965 |
| Qwen2.5-7B | bfloat16 | 0.4448 | 35.33 | 13309.58 | 40.38 | 196.45 | 1.0000 |
| | float16 | **0.4467** | 35.35 | 13766.35 | 38.36 | 197.40 | 1.0043 |
| | int8 | 0.4478 | 36.78 | 13456.12 | 41.23 | 199.45 | 2.0135 |
| | fp8 | 0.4500 | 37.45 | 13890.67 | 39.12 | 201.34 | 2.0234 |
| | int4 | 0.4152 | **27.47** | **12912.91** | **43.10** | **168.73** | 3.7338 |
| Qwen2.5-14B | bfloat16 | **0.4691** | 51.83 | 24065.05 | 24.74 | 205.97 | 1.0000 |
| | float16 | **0.4691** | 52.66 | 24708.61 | **23.50** | 203.13 | 1.0000 |
| | int8 | 0.4705 | 52.34 | 24012.89 | 25.45 | 212.34 | 2.0060 |
| | fp8 | 0.4755 | 53.01 | 25045.23 | 24.67 | 207.89 | 2.0273 |
| | int4 | 0.4286 | **34.13** | 27865.30 | 25.89 | **187.40** | 3.6547 |
| Qwen2.5-32B | bfloat16 | **0.5523** | 71.33 | 26666.92 | 17.54 | 279.23 | 1.0000 |
| | float16 | 0.5505 | 71.86 | 27399.52 | **16.66** | 259.53 | 0.9967 |
| | int8 | 0.5525 | 70.45 | 27012.45 | 18.45 | 281.23 | 2.0007 |
| | fp8 | 0.5550 | 72.12 | 27567.89 | 17.89 | 276.45 | 2.0098 |
| | int4 | 0.5095 | **48.30** | 25140.95 | 19.20 | **214.57** | 3.6900 |
| Phi-4 | bfloat16 | **0.4035** | 48.19 | 6547.79 | 45.16 | **217.16** | 1.0000 |
| | float16 | 0.4006 | 49.02 | **6424.68** | **42.90** | 224.63 | 0.9928 |
| | int8 | 0.4025 | 48.67 | 6600.45 | 46.78 | 221.34 | 1.9950 |
| | fp8 | 0.4050 | 49.12 | 6400.78 | 44.56 | 226.78 | 2.0074 |
| | int4 | 0.3950 | **43.27** | 12202.10 | 48.19 | 319.11 | 3.9157 |
| Phi-3.5-mini | bfloat16 | **0.3683** | 41.19 | 8978.35 | 54.87 | 172.02 | 1.0000 |
| | float16 | 0.3652 | 41.54 | **8647.61** | **52.13** | 172.28 | 0.9916 |
| | int8 | 0.3702 | 42.34 | 9000.12 | 55.45 | 173.89 | 2.0103 |
| | fp8 | 0.3720 | 41.78 | 8700.34 | 53.12 | 179.45 | 2.0201 |
| | int4 | 0.3355 | **36.52** | 11761.00 | 58.54 | **159.15** | 3.6438 |

## D.5 ASSESSMENT OF BIT-WIDTH QUANTIZATION INFERENCE EFFICIENCY

Inference efficiency plays a crucial role in the practical deployment of LLMs, vision-language models (VLMs), and large vision models (LVMs). Optimizing inference efficiency is essential to ensure low latency, minimal resource consumption, and effective energy utilization, enabling these models to be deployed in diverse and resource-constrained environments. This section evaluates the inference efficiency across various precision modes (bfloat16, float16, and int4 quantization) and model architectures, highlighting trade-offs in performance metrics and computational resource usage.

**Note on Int8 Quantization.** We did not include `int8` quantization results in this section because current inference support for `int8` on NVIDIA Hopper architecture (GH200) is either incomplete or exhibits instability due to backend kernel issues. During our initial tests, `int8`-based inference led to runtime errors and inconsistent throughput behavior. We are actively investigating and working to resolve these issues. Once the compatibility and reliability of `int8` quantization are verified, we will update our evaluation results accordingly.

**Goal.** In this section, we systematically assess inference efficiency across different model precisions (bfloat16, float16, and int4) for a range of model architectures, including DeepSeek, Qwen, Phi, and Yi models with parameter sizes from 1.5B to 34B. Specifically, we evaluate the impact of precision and quantization on key metrics, including task-specific performance (MMLU-Pro, BBH, GPQA, IFEval, MATH, MUSR), Average Memory Utilization (AMU), Average Latency (AL), Throughput (tokens per second), Average Energy Consumption (AEC), and Model Compression Ratio (MCR). This comprehensive analysis provides clear insights for selecting suitable inference strategies based on targeted deployment scenarios.

**Hardware and Inference Framework.** The inference experiments were performed on an optimized inference server infrastructure consisting of NVIDIA GH200 96GB GPUs. The infrastructure setup comprised one node containing four H100 GPUs coupled with NVIDIA Grace processors (288 cores, 288 threads) to support efficient data processing and task scheduling. NVLink interconnects were utilized for rapid GPU-to-GPU communication, ensuring low latency and efficient data transfer.

**Evaluation Results.** Table 12 presents the inference efficiency results across varying precisions for multiple model architectures. In the DeepSeek-R1-Distill-Qwen-1.5B model, the int4 precision significantly increased throughput (42.34 tokens/s) and reduced memory utilization (19.49 GB), albeit at a marginal performance degradation (Avg Performance: 0.2341) compared to bfloat16 and float16. Similarly, for larger models like Qwen2.5-32B and Yi-34B, int4 quantization substantially enhanced throughput and memory efficiency, indicating its suitability for deployment scenarios prioritizing computational efficiency over maximum performance. Models at bfloat16 precision typically showed the highest performance metrics across architectures but at the cost of increased memory usage and energy consumption. The Phi-4 model demonstrated particularly high throughput (45.16 tokens/s) and acceptable performance (Avg Performance: 0.4035) at bfloat16, highlighting its efficacy for scenarios demanding balanced performance and efficiency. Overall, int4 quantization emerges as a robust option for resource-constrained deployment scenarios, while bfloat16 remains preferable for applications requiring optimal performance metrics.

Table 13: Efficiency LVMs Results for Attention Mechanisms. The best result is compared under the same model.

| Method | Model | Training Steps | FID ↓ | AMU (GB) ↓ | AL (s/iter) ↓ | TT (Tokens/param/s) ↑ | AEC (W)↓ |
|---|---|---|---|---|---|---|---|
| MHA | DiT-XL/2 | 400K | 19.47 | **40.50** | 0.2873 | $1.3301\times10^{-06}$ | 182.34 |
| | DiT-L/8 | 400K | 118.87 | 23.49 | **0.1635** | **$3.5591\times10^{-06}$** | **75.35** |
| | DiT-B/4 | 250K | 68.38 | **15.51** | 0.1423 | $1.3921\times10^{-05}$ | 70.07 |
| MQA | DiT-XL/2 | 400K | 8.93 | 43.78 | **0.2637** | **$1.6033\times10^{-06}$** | **172.61** |
| | DiT-L/8 | 400K | **78.05** | 23.03 | 0.1818 | $3.3491\times10^{-06}$ | 80.75 |
| | DiT-B/4 | 250K | 55.29 | 16.13 | **0.1413** | **$1.5484\times10^{-05}$** | **67.76** |
| GQA | DiT-XL/2 | 400K | **8.71** | 43.71 | 0.2696 | $1.5332\times10^{-06}$ | 174.36 |
| | DiT-L/8 | 400K | 81.90 | **22.65** | 0.1816 | $3.4302\times10^{-06}$ | 78.81 |
| | DiT-B/4 | 250K | **53.99** | 16.25 | 0.1438 | $1.4775\times10^{-05}$ | 71.51 |
| MLA | DiT-XL/2 | 400K | 116.93 | 45.84 | 0.3291 | $1.1743\times10^{-06}$ | 174.36 |
| | DiT-L/8 | 400K | 114.63 | 23.88 | 0.2048 | $2.7843\times10^{-06}$ | 84.16 |
| | DiT-B/4 | 250K | 73.88 | 16.26 | 0.2100 | $1.2096\times10^{-05}$ | 71.09 |
| NSA | DiT-XL/2 | 400K | 22.78 | 59.34 | 0.5771 | $3.2559\times10^{-07}$ | 256.49 |
| | DiT-L/8 | 400K | 89.98 | 24.77 | 0.3416 | $1.6209\times10^{-06}$ | 107.32 |
| | DiT-B/4 | 250K | 55.27 | 18.94 | 0.2543 | $8.4051\times10^{-06}$ | 85.58 |

Table 14: Efficiency Results for LVM's MoE Mechanisms.

| Method | Parameters | Training Steps | FID ↓ | AMU (GB) ↓ | AL (s/iter) ↓ | TT (Tokens/param/s) ↑ | AEC (J)↓ |
|---|---|---|---|---|---|---|---|
| Dense Model | 675M (DiT-XL/2) | 400K | 19.47 | **40.50** | 0.2873 | $1.3301\times10^{-06}$ | **182.34** |
| Dense Model | 459M (DiT-L/8) | 400K | 118.87 | **23.49** | 0.1635 | $3.5591\times10^{-06}$ | **75.35** |
| Dense Model | 130M (DiT-B/4) | 250K | 68.38 | **15.51** | 0.1423 | $1.3921\times10^{-05}$ | **70.07** |
| MoE Model | 675M×8 (DiT-XL/2) | 400K | **16.35** | 47.82 | **0.2340** | **$2.1568\times10^{-06}$** | 231.07 |
| MoE Model | 459M×8 (DiT-L/8) | 400K | **76.41** | 29.76 | **0.1358** | **$5.8724\times10^{-06}$** | 105.49 |
| MoE Model | 130M×8 (DiT-B/4) | 250K | **45.62** | 18.95 | **0.1138** | **$2.0882\times10^{-05}$** | 89.69 |

# E  SCALABILITY OF EFFICIENTLLM BENCHMARK

The previous sections demonstrated how EfficientLLM quantifies architectural and training-time trade-offs for purely textual LLMs. We now extend that investigation to vision and vision–language settings, but with a deliberately tight scope: we evaluate only those acceleration strategies first validated on LLMs that can be applied unchanged to their visual counterparts. Concretely, we (i) insert efficient attention variants (MQA, GQA, MLA, NSA) into DiT-style diffusion transformers, (ii) swap dense blocks for Mixture-of-Experts (MoE) layers in the same DiT backbones, and (iii) benchmark a palette of parameter-efficient fine-tuning (PEFT) methods—LoRA, LoRA-plus, RSLoRA, DoRA, PISSA, LoHa, LoKr, and GLoRA—across large-scale LVMs and VLMs (LLaVA-1.5, Qwen2.5-VL-7B, Intern-VL-38B, QvQ-Pre-72B, Wan 2.1, Stable Diffusion 3.5). Because EfficientLLM's metric collector is modality-agnostic, the same pipeline that logged AMU, latency, throughput, energy, and perplexity for language now records the identical metrics alongside vision-specific quality signals such as FID or loss. This unified view lets us ask a single question throughout the remainder of the section: when an optimization accelerates text, does it still pay off when the "tokens" are image patches or joint text–image embeddings?

## E.1  EFFICIENCY FOR TRANSFORMER BASED LVMS ARCHITECTURE PRETRAINING

**Efficient Attention Mechanisms for LVMs.** As shown in Table 13, in the context of Large Vision Models (LVMs), efficient attention mechanisms play a critical role by optimizing computational resources, latency, and memory usage while maintaining high-quality outputs. Our evaluation encompassed several attention variants, including standard Multi-Head Attention (MHA), Multi-Query Attention (MQA), Grouped-Query Attention (GQA), Multi-Head Latent Attention (MLA), and Native Sparse Attention (NSA), assessed across different DiT model architectures (DiT-XL/2, DiT-

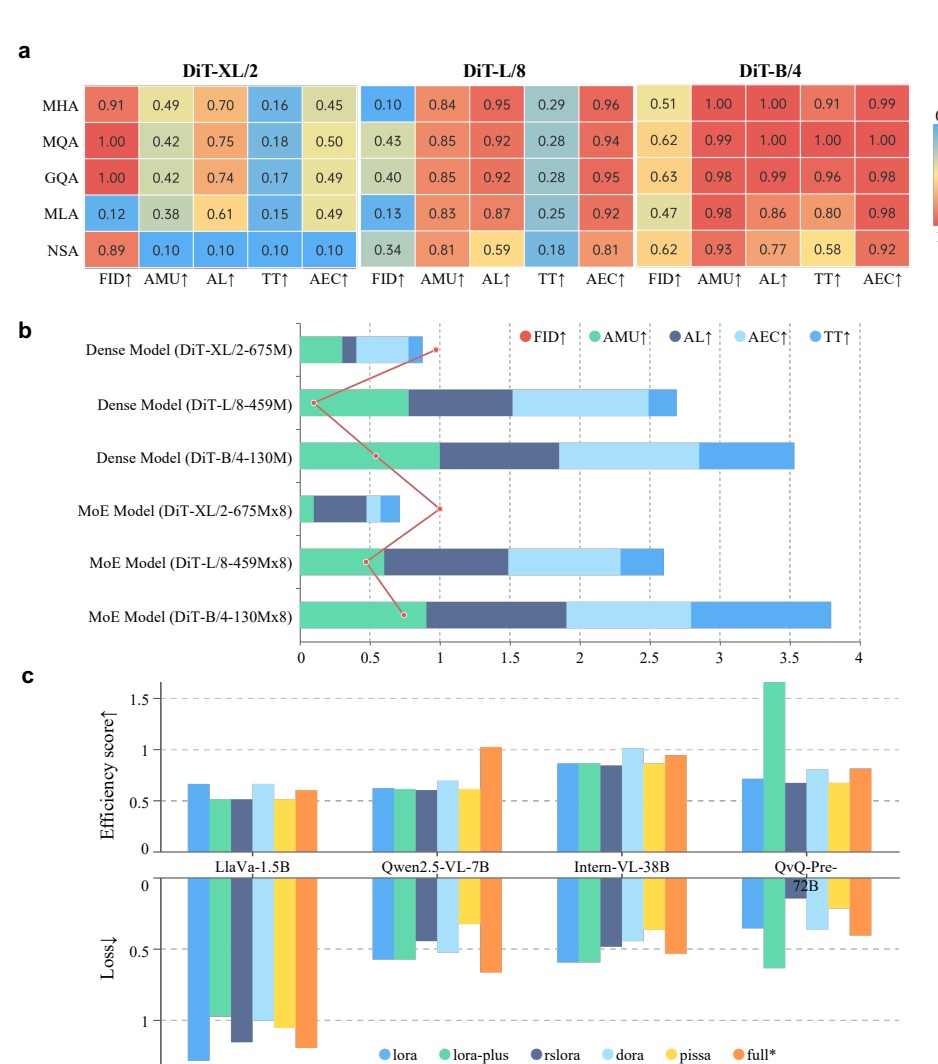

Figure 7: **Scalability analysis of EfficientLLM for LVM and VLM optimization.** (a) Normalized efficiency scores across five metrics (FID↑, AMU↑, AL↑, TT↑, AEC↑) for attention variants (MHA, MQA, GQA, MLA, NSA) in three DiT-based LVM architectures (DiT-XL/2, L/8, B/4). All metrics are min-max normalized to [0,1] and higher values indicate better efficiency. (b) MoE vs. dense models across identical DiT backbones. MoE-based architectures consistently outperform dense counterparts in throughput and FID while incurring moderate AMU and AEC overhead. (c) Comparison of Parameter-Efficient Fine-Tuning (PEFT) methods (e.g., LoRA, RSLoRA, PISSA, DoRA) on various VLMs. Bars indicate normalized *Efficiency score* (top, higher is better) and *Loss* (bottom, lower is better). Methods marked with * indicate full fine-tuning using DeepSpeed.

L/8, and DiT-B/4). Results indicated that GQA and MQA consistently achieved superior performance in terms of Fréchet Inception Distance (FID), with GQA exhibiting the lowest FID scores in DiT-XL/2 (FID = 8.71) and DiT-B/4 (FID = 53.99). MQA closely followed, providing balanced efficiency and performance, notably in DiT-XL/2 (FID = 8.93) with the lowest latency (AL = 0.2637 s/iter) and high throughput (TT = $1.6033 \times 10^{-06}$ TFloats). MLA, although generally less efficient, demonstrated substantial performance in DiT-B/4 scenarios, indicating its suitability for specific parameter and architecture configurations. NSA showed its strengths primarily in memory-intensive tasks despite higher latency, underscoring its potential for specific deployment environments with particular resource constraints. These findings highlight the importance of selecting an attention mechanism aligned with both the performance goals and the computational resources available for large-scale vision tasks.

**Sparse Modeling via MoE for LVMs.** As shown in Table 14, our experiments with Mixture of Experts (MoE) architectures for Large Vision Models revealed consistent performance improvements across all model scales. The 675M×8 MoE configuration (DiT-XL/2) achieved a significantly lower FID score of 16.35 compared to its dense counterpart (FID = 19.47), indicating superior image generation quality. Similarly, the 459M×8 (DiT-L/8) and 130M×8 (DiT-B/4) MoE models demonstrated substantial improvements with FID scores of 76.41 and 45.62, outperforming their dense equivalents which scored 118.87 and 68.38 respectively. These performance gains, however, come with increased resource requirements. MoE configurations showed higher memory utilization across all scales, with the 675M×8 model requiring 47.82 GB compared to 40.50 GB for the dense version. Interestingly, despite the expanded parameter space, MoE models exhibited improved computational efficiency with lower average latency (AL = 0.2340 s/iter for 675M×8 versus 0.2873 s/iter for the dense equivalent) and substantially higher throughput (TT = $2.1568 \times 10^{-06}$ TFloats versus $1.3301 \times 10^{-06}$ TFloats). This pattern was consistent across smaller model scales as well, with the 130M×8 (DiT-B/4) MoE model achieving approximately 50% higher throughput than its dense counterpart while delivering significantly better generation quality. These results suggest that sparse modeling via MoE provides a compelling approach for scaling vision models, enabling more effective parameter utilization through conditional computation where specialized experts can focus on different visual patterns and representations.

### E.2 ASSESSMENT OF PEFT ON LVMs

**Disney Organized Dataset.** Table 9 demonstrates distinct efficiency-performance trade-offs among PEFT methods for the Wan 2.1-1.5B model. The full fine-tuning approach achieved the lowest loss (0.104) with optimal sample throughput ($1.61 \times 10^{-10}$ Samples/param/s) and competitive latency (33.2042 s/iter), although with the highest memory usage (78.44 GB). GLORA provided a good balance, showing competitive loss (0.143) with high throughput ($1.61 \times 10^{-10}$ Samples/param/s) and reduced latency (33.1298 s/iter).

**WikiArt Sargent Dataset.** In the case of the Stable Diffusion 3.5 Medium model, full fine-tuning achieved the best loss performance (0.204) and highest throughput ($8.50 \times 10^{-10}$ Samples/param/s), despite significantly elevated memory usage (82.48 GB). LoHA and GLORA methods also performed well, maintaining low losses (0.215 and 0.217 respectively) and balanced latency around 4.6 s/iter, highlighting their suitability for applications demanding high computational efficiency without sacrificing performance.

Overall, while full fine-tuning provides superior performance, it demands greater computational resources. Alternative methods such as GLORA and LoHA offer compelling trade-offs suitable for various deployment environments.

### E.3 ASSESSMENT OF PEFT ON VLMs

**ChatQA Dataset.** Table 9 highlights the efficiency-performance trade-offs of various Parameter-Efficient Fine-Tuning (PEFT) methods across different visual-language models. For the 7B-parameter LLaVA-1.5 model, LoRA-plus achieved the lowest loss (0.9716), demonstrating balanced efficiency with reasonable latency (7.1028 s/iter) and moderate energy consumption (541.45 W). Parameter freezing methods were not reported for this model. In the case of the Qwen2.5-VL-7B model, PISSA exhibited superior performance with the lowest loss (0.3156) while maintaining competitive latency (8.9645 s/iter) and energy efficiency (405.08 W). Notably, full fine-tuning with DeepSpeed

Table 15: Assessment of Training and Tuning Efficiency for VLMs of ChatQA Dataset (methods marked with * use DeepSpeed). Because of the different batch size, *full** are not included in the comparisons. The best result is compared under the same model.

| Model | Methods | Loss↓ | AMU (GB)↓ | PCU↑ | AL (s/iter)↓ | ST (Samples/param/s)↑ | AEC (W)↓ |
|---|---|---|---|---|---|---|---|
| | | | **ChatQA** | | | | |
| LLaVA-1.5 | lora | 1.2796 | **20.064** | 0.8942 | 7.794051 | $1.2838 \times 10^{-10}$ | **512.27** |
| | lora-plus | **0.9716** | 45.216 | 0.9853 | 7.102787 | $1.4084 \times 10^{-10}$ | 541.45 |
| | rslora | 1.1541 | 45.6576 | 0.9891 | 6.987395 | $1.4292 \times 10^{-10}$ | 522.86 |
| | dora | 1.0015 | 59.7728 | **0.9894** | 11.977185 | $8.3514 \times 10^{-11}$ | 548.88 |
| | pissa | 1.0549 | 45.4176 | **0.9894** | **6.952982** | $\mathbf{1.4358 \times 10^{-10}}$ | 524.85 |
| | full* | 1.1889 | 61.6992 | 0.9374 | 9.784834 | $1.0218 \times 10^{-10}$ | 484.52 |
| Qwen2.5-VL-7B | lora | 0.5672 | 46.3008 | 0.9918 | 9.270629 | $1.2246 \times 10^{-10}$ | **403.29** |
| | lora-plus | 0.5672 | 45.84 | 0.9889 | 8.918348 | $1.2729 \times 10^{-10}$ | 403.70 |
| | rslora | 0.4363 | **45.6384** | 0.9888 | **8.82855** | $\mathbf{1.2855 \times 10^{-10}}$ | 419.00 |
| | dora | 0.5170 | 61.4496 | 0.9956 | 13.039291 | $8.7712 \times 10^{-11}$ | 548.59 |
| | pissa | **0.3156** | 45.696 | **0.9957** | 8.964483 | $1.2712 \times 10^{-10}$ | 405.08 |
| | full* | 0.6576 | 25.344 | 0.8297 | 16.7143 | $1.5995 \times 10^{-11}$ | 354.44 |
| Intern-VL-3-38B | lora | 0.5943 | 42.4704 | 0.9825 | 15.710554 | $1.3611 \times 10^{-11}$ | 530.84 |
| | lora-plus | 0.5943 | 42.5184 | **0.9881** | 15.742757 | $1.3584 \times 10^{-11}$ | **523.62** |
| | rslora | 0.4760 | 43.0272 | 0.9854 | **15.461066** | $\mathbf{1.3844 \times 10^{-11}}$ | 533.49 |
| | dora | 0.4409 | 60.6144 | 0.989 | 20.866331 | $1.0206 \times 10^{-11}$ | 551.23 |
| | pissa | **0.3635** | **42.0192** | 0.9877 | 15.848549 | $1.3335 \times 10^{-11}$ | 526.92 |
| | full* | 0.5274 | 69.2448 | 0.9753 | 18.485439 | $1.1426 \times 10^{-11}$ | 478.33 |
| QvQ-Pre-72B | lora | 0.3548 | 36.624 | 0.268 | **5.84746** | $\mathbf{1.9225 \times 10^{-11}}$ | 374.27 |
| | lora-plus | 0.6311 | **35.0464** | 0.8207 | 35.525615 | $3.1557 \times 10^{-12}$ | 348.08 |
| | rslora | **0.1434** | 40.1024 | 0.8732 | 8.44855 | $1.3350 \times 10^{-11}$ | 381.33 |
| | dora | 0.3554 | 58.6512 | 0.2573 | 8.394302 | $1.3434 \times 10^{-11}$ | **330.30** |
| | pissa | 0.2143 | 42.6688 | **0.8956** | 8.6275 | $1.3048 \times 10^{-11}$ | 369.51 |
| | full* | 0.3980 | 61.7088 | 0.7895 | 12.3575 | $9.0245 \times 10^{-12}$ | 352.67 |

Table 16: Assessment of Training and Tuning Efficiency for LVMs of Disney Organized and WikiArt Sargent Datasets (methods marked with * use DeepSpeed). Because of the different batch size, *full** are not included in the comparisons.

| Model | Methods | Loss↓ | AMU (GB)↓ | PCU↑ | AL (s/iter)↓ | ST (Samples/param/s)↑ | AEC (W)↓ |
|---|---|---|---|---|---|---|---|
| | | | **Disney Organized** | | | | |
| Wan 2.1-1.5B | lora | 0.136 | 50.22 | 0.8942 | 44.342308 | $1.20 \times 10^{-10}$ | **512.27** |
| | loha | **0.125** | **48.69** | 0.5824 | 42.430697 | $1.26 \times 10^{-10}$ | 566.11 |
| | lokr | 0.139 | 58.02 | **0.9940** | **45.551551** | $1.17 \times 10^{-10}$ | 648.91 |
| | glora | 0.143 | 51.01 | 0.8213 | 33.129847 | $\mathbf{1.61 \times 10^{-10}}$ | 593.23 |
| | full* | 0.104 | 78.44 | 0.9027 | 33.204205 | $1.61 \times 10^{-10}$ | 518.80 |
| | | | **WikiArt Sargent** | | | | |
| Stable Diffusion 3.5 Medium | lora | 0.225 | **15.30** | 0.9536 | **4.008191** | $\mathbf{7.68 \times 10^{-10}}$ | 607.12 |
| | loha | **0.215** | 15.42 | 0.7207 | 4.673482 | $6.58 \times 10^{-10}$ | 556.32 |
| | lokr | 0.229 | 17.26 | 0.9556 | 4.820688 | $6.38 \times 10^{-10}$ | 567.50 |
| | glora | 0.217 | 17.92 | 0.7484 | 4.632438 | $6.64 \times 10^{-10}$ | **553.25** |
| | full* | 0.204 | 82.48 | 0.8439 | 3.618949 | $8.50 \times 10^{-10}$ | 462.18 |

had significantly reduced memory utilization (25.344 GB) but incurred substantially higher latency (16.7143 s/iter), reflecting a critical efficiency-performance trade-off. For the larger Intern-VL-3-38B model, PISSA again delivered strong results, showing the lowest loss (0.3635) among the evaluated methods, albeit with increased latency (15.8485 s/iter). DoRA presented higher latency (20.8663 s/iter) and elevated memory usage (60.6144 GB), limiting its practicality in latency-sensitive scenarios. With the largest model QvQ-Pre-72B, RSLoRA outperformed other methods with the lowest loss (0.1434) and reasonable latency (8.4486 s/iter), suggesting it as a highly effective approach for tuning extremely large models. Despite low AMU (35.0464 GB), LoRA-plus showed significantly higher latency (35.5256 s/iter), making it less favorable for latency-sensitive applications. Overall, LoRA variants, especially LoRA-plus and PISSA, consistently offer balanced efficiency-performance trade-offs suitable for diverse applications. RSLoRA emerges as particularly advantageous for large-scale model tuning, while computationally intensive approaches like DoRA and full fine-tuning require careful consideration based on specific deployment scenarios.

## F  RELATED WORK

A wide range of efforts have emerged to improve the efficiency of large language models (LLMs) across their lifecycle (Hu et al., 2021a; Dettmers et al., 2022; Sanh et al., 2019b; Zhao et al., 2024a; Frantar et al., 2022b; Liu et al., 2023d). In this work we concentrate on three facets – architecture-level pretraining optimizations, parameter-efficient fine-tuning, and inference-time quantization – because these correspond to major efficiency challenges at different stages of an LLM's development and deployment. Each aspect addresses the needs of different stakeholders in practice: architecture and pretraining improvements guide model designers in building and training new LLMs under limited compute budgets; parameter-efficient fine-tuning (PEFT) methods help practitioners adapt big models to downstream tasks without retraining entire networks; and low bit-width quantization techniques assist deployment engineers in reducing serving costs and latency without requiring additional retraining. Below, we will highlight other important efficiency strategies not covered in detail (e.g. systems-level optimizations, alignment via RLHF, and test-time acceleration techniques), mainly clarify why they fall outside the scope of our study.

**Distributed Training and System-Level Optimizations.**  Training giant models efficiently at scale is as much a systems engineering challenge as an algorithmic one.  A rich body of work exists on optimizing the infrastructure and parallelization for large-scale training. Approaches like data-parallel and model-parallel training (and hybrids thereof) allow spreading computation across many GPUs or TPUs. For example, Google's GPipe introduced generic pipeline parallelism to partition a model across accelerators and achieved almost linear speedups when scaling an MLP and a 6-billion Transformer across devices (Huang et al., 2019). NVIDIA's Megatron-LM (Shoeybi et al., 2019b) and Google's Mesh-TensorFlow (Shazeer et al., 2018) further refined tensor-slicing model-parallel approaches to train models with up to 100+ billion parameters (like the original GPT-3 (Brown et al., 2020)). In addition, the DeepSpeed library from Microsoft introduced the Zero Redundancy Optimizer (ZeRO) (Rajbhandari et al., 2020) which eliminates memory duplication of optimizer states and gradients across data-parallel workers. By offloading and partitioning states, ZeRO allows training models with hundreds of billions of parameters with high efficiency, even enabling 100+ billion models to be trained on modest GPU clusters with super-linear speedup.  These system-level advances – including optimized kernels (e.g. FlashAttention (Dao et al., 2022a)), scheduling algorithms, and memory management techniques – are crucial for making the training of cutting-edge LLMs possible at all. We do not explicitly benchmark these in our study because they often require specialized hardware setups or custom distributed training implementations beyond our end-to-end evaluation scope. In essence, our focus was on algorithmic techniques that a single-team researcher or practitioner could apply within a given infrastructure, whereas system-level optimizations involve entire training pipeline re-design and are orthogonal to the model-internal methods we examined. We refer interested readers to comprehensive system papers (e.g. PipeDream (Harlap et al., 2018) and ZeRO (Rajbhandari et al., 2020)) for further details on this topic.

**Alignment and RLHF Efficiency.**  Large language models are typically fine-tuned after pretraining to better align with human preferences, follow instructions, and produce safe outputs. The dominant approach for this is Reinforcement Learning from Human Feedback (RLHF), exemplified by the InstructGPT and ChatGPT series (Ouyang et al., 2022). InstructGPT showed that a 1.3B parameter model fine-tuned with human preference data outperformed a 175B GPT-3 on helpfulness and truthfulness. This highlights an "efficiency" of a different sort – alignment work can make smaller models behave as usefully as much larger ones, by optimizing for the right objective.  However, RLHF itself is resource-intensive: it involves training a reward model (often a large network) and running many steps of policy optimization (e.g. PPO (Schulman et al., 2017)) for the LLM, which can be as costly as regular fine-tuning. Recent research has proposed more sample-efficient or proxy methods for alignment, such as using AI feedback or distilled preference models, but these are still emerging (Tunstall et al., 2023; Hong et al., 2023; Lee et al., 2023; Zhu et al., 2024b; Fisch et al., 2024). We did not focus on RLHF in our benchmark because it targets output quality and safety more than runtime or training efficiency per se. Moreover, evaluating alignment quality requires human judgment or specialized metrics, which is outside our predominantly system performance–oriented evaluation criteria. In short, RLHF and other alignment techniques are critical in practice, but they involve a distinct stage of the model lifecycle with goals (ethical and behavioral alignment) different from the core efficiency measures we target. Incorporating alignment efficiency (e.g. measuring the compute required for RLHF and how to reduce it) is an interesting direction for future work, though

it likely requires an end-to-end infrastructure and human-in-the-loop setup beyond the scope of our current study.

**Inference-Time Acceleration Strategies.** A number of techniques aim to speed up inference beyond just lowering bit precision. One such category is test-time optimizations that exploit the prediction process of LLMs. For example, speculative decoding has emerged as a powerful approach to accelerate autoregressive generation (Leviathan et al., 2023; Chen et al., 2023a). OpenAI has reported $2 - 3\times$ speedups in GPT-3 using speculative decoding with a smaller GPT-2 as the draft model (Xia et al., 2024). Another technique is early exiting in the model's forward pass (Xu et al., 2025; Chen et al., 2023d). If intermediate layers of a Transformer are equipped with prediction heads or confidence estimators, the model can choose to stop computation once it is sufficiently confident, instead of always running all $N$ layers. Elhoushi et al. combine this with a form of self-speculative decoding in a system called LayerSkip (Elhoushi et al., 2024). By training LLaMA models with progressively higher dropout in later layers and a shared early-exit classifier, they enable the model to exit at an earlier layer for "easy" inputs and only use the full depth for "hard" cases. This yielded up to 2.0˜2.2× speedups on tasks like summarization and code generation, with negligible performance loss. These dynamic inference methods are highly relevant to efficiency – they essentially adapt the compute on the fly to match the input's complexity or the model's own confidence. We consider them complementary to our quantization and architecture-focused evaluations. In our study, we kept inference routines fixed (all models generate with the same decoder strategy) to ensure a controlled comparison of techniques like quantization. Integrating speculative decoding or early-exit requires building additional components and policies around the model, which was beyond our current scope.

**Dynamic Routing and Model Cascades.** A related idea is deploying model cascades or multi-scale models at inference (Kolawole et al., 2024; Mamou et al., 2022). For instance, one might use a small model to handle simple queries and only invoke a large model for more complex queries (a form of dynamic routing at the whole-model level). Similarly, mixture-of-experts (discussed above) can be viewed as dynamic routing within a single forward pass – experts are activated only as needed (Huang et al., 2024a; Wang et al., 2020b). These approaches can yield huge savings when there is variability in input difficulty or when many requests do not require the full capacity of the largest model. The challenge is designing reliable routing mechanisms that know when the big model is needed, without introducing too much overhead or too many errors. While our work did not explore such conditional computation at inference time, we acknowledge it as an important research frontier. Successfully deploying conditional LLM inference (whether via cascades, early-exits, or MoE gating) could drastically improve real-world efficiency by ensuring we pay the cost of a 100B+ model only when necessary.

In summary, beyond the specific techniques evaluated in our study, the literature offers a spectrum of strategies to tackle LLM efficiency from multiple angles. Training-time system optimizations, alignment-focused fine-tuning, and clever decoding-time methods all contribute to the overall goal of making LLMs more practical and sustainable. We focused on architecture, fine-tuning, and quantization as representative axes that span the model's lifecycle and are widely applicable under uniform evaluation settings. The insi·ghts from our benchmark can thus be seen as one piece of the puzzle, complementing the above lines of work. Future research will hopefully integrate these layers – for example, applying system optimizations to efficiently train models with new architectures, or combining PEFT and quantization with speculative decoding for maximum inference speed-up. Such holistic exploration will be vital as the community continues to push the limits of large language model capabilities under real-world resource constraints.

Although we strive to present a representative overview of efficient LLM research, our discussion is by no means comprehensive. The landscape of efficiency techniques—spanning algorithmic, system-level, and application-specific innovations—is vast and rapidly evolving. Due to space and scope constraints, we have selected the dimensions that are most relevant to our empirical evaluation. As many promising techniques and insights are beyond the scope of this paper, readers refer to this work as a focused discussion rather than a comprehensive survey.

## G    DISCUSSION

While our study provides a comprehensive empirical evaluation of efficiency techniques across multiple dimensions, achieving truly *compute-aware* large-model design and deployment remains an open challenge. In this section, we first acknowledge several limitations of our current work and then articulate key open challenges and promising future research directions.

### G.1    LIMITATIONS

Our empirical benchmark and analysis have several limitations that should be considered when interpreting the results:

- **Limited Coverage of Efficiency Techniques.** Although we extensively evaluated multiple efficiency strategies, our analysis does not encompass all existing techniques. For instance, we have not explicitly considered optimizations related to sequence length management, such as efficient handling of ultra-long-context models, KV-cache optimizations, and strategies for reducing memory overhead in attention mechanisms. These techniques can significantly impact computational efficiency, especially in scenarios involving extremely long input sequences during pretraining and inference.

- **Hardware and Infrastructure Constraints.** Our experiments were conducted primarily on a specific GPU cluster configuration (48×GH200 + 8×H200 GPUs). Different hardware setups, such as TPU-based systems, CPU-only clusters, or heterogeneous computing environments, may yield different efficiency trade-offs, particularly during large-scale pretraining. Thus, our findings may not fully generalize to all possible deployment scenarios.

- **Limited Scope of Models and Tasks.** Although we evaluated a diverse set of models across language, vision, and multimodal domains, our selection does not cover all existing architectures and tasks. Certain specialized models or niche application scenarios may exhibit unique efficiency characteristics not captured in our current evaluation, especially during the pretraining phase.

- **Static Evaluation Metrics.** Our proposed metrics, while comprehensive, are primarily static and averaged over training or inference processes. Dynamic or adaptive metrics that capture real-time fluctuations in resource utilization, latency spikes, or transient bottlenecks could provide additional insights into efficiency optimization.

- **Absence of Economic Analysis.** Our evaluation focuses on computational and energy efficiency metrics without explicitly considering economic factors such as hardware acquisition costs, operational expenses, or cloud computing pricing models. Incorporating these economic dimensions could further enhance the practical relevance of our efficiency assessments.

### G.2    OPEN CHALLENGES AND FUTURE DIRECTIONS

Beyond the limitations above, we highlight several critical open challenges and promising research directions for future work:

- **Multi-objective Scaling Laws.** Classic scaling laws (e.g., Chinchilla) minimize cross-entropy loss under a scalar compute constraint, implicitly assuming FLOPs as the sole budget. Real-world deployments require balancing multiple orthogonal objectives such as latency, memory, energy, and carbon emissions. Developing vector-valued scaling laws that map parameters and tokens onto an efficiency Pareto frontier remains unexplored.

- **Heterogeneous-quality Corpora.** At trillion-token scales, datasets contain diverse quality levels, from curated books to noisy web text. Current heuristics ignore fine-grained variance. Efficient token-level entropy estimators and dynamic importance sampling methods are needed to optimize training efficiency and quality simultaneously.

- **Curriculum Design for Long-context Pretraining.** Models with extremely long contexts (32k–128k tokens) require principled curriculum strategies beyond simple heuristics. Addressing memory bandwidth constraints, positional encoding dynamics, gradient staleness, and downstream coherence remains challenging.

- **Sparse Routing under Hard Memory Ceilings.** Mixture-of-Experts (MoE) architectures reduce FLOPs but increase KV-cache memory usage. Developing unified theoretical frameworks and memory-aware routing mechanisms that dynamically balance compute and memory remains an open frontier.

- **Efficient Optimization of Non-Transformer Backbones.** Alternative architectures (e.g., Mamba, RWKV) promise sub-quadratic scaling but lack optimized kernels and adaptive optimizers. Establishing standardized benchmarks for fair comparisons against Transformers is essential.

- **PEFT for Multi-modal and Tool-augmented LLMs.** Parameter-efficient fine-tuning (PEFT) methods like LoRA perform well in pure language settings but struggle across modalities. Designing unified adapters that generalize across vision, audio, and code modalities remains challenging.

- **Robust Post-training Quantization for Ultra-long Contexts.** Current quantization schemes (e.g., int4) degrade significantly with activation outliers in long sequences. Developing robust joint weight–activation quantizers and comprehensive error-propagation theories is critical.

- **Holistic, End-to-end Efficiency Evaluation.** Existing benchmarks often cherry-pick metrics and hardware setups. A reproducible, standardized efficiency benchmarking framework capturing latency, throughput, energy, and memory across diverse hardware and software configurations is urgently needed.

- **Continual and Federated Pretraining under Privacy Constraints.** Regulatory requirements increasingly demand on-premises data handling. Balancing compute-optimal token budgets with privacy guarantees (e.g., differential privacy) through federated learning or secure aggregation remains challenging.

- **Hardware-aware Training Schedules.** Heterogeneous GPU clusters complicate manual scheduling. Developing auto-schedulers that dynamically optimize parallelism strategies (data, tensor, pipeline, expert) across diverse hardware configurations is an active research area.

    *Solving these interlocking challenges demands a concerted effort that spans theory, optimization, systems, and hardware co-design. Only then will we unlock the next order-of-magnitude leap in large-model efficiency.*

# H  OTHER SUPPLEMENTARY

## H.1  VLMs AND LVMs BACKGROUND

**Large Vision Models (LVMs).** Large Vision Models (LVMs) have emerged as a significant advancement in the field of artificial intelligence, particularly within the domain of generative models. These models are primarily designed for image and video generation tasks, demonstrating robust multimodal integration capabilities that enable them to comprehend and process relationships between text and images. By leveraging such capabilities, LVMs can effectively transform textual descriptions into visual representations. Most state-of-the-art vision generation models employ diffusion model architectures, which progressively denoise random noise to reconstruct high-quality images. Additionally, these models extensively utilize self-attention and cross-attention mechanisms to capture long-range dependencies within images and effectively align textual and visual features.

Among the representative models, Stable Diffusion, developed by Stability AI, is an open-source image generation model based on a latent diffusion architecture, which performs the diffusion process in latent space rather than pixel space, significantly reducing computational complexity. DALL-E (Ramesh et al., 2021; 2022), developed by OpenAI, represents the forefront of text-to-image generation, offering strong text comprehension capabilities and highly realistic image synthesis. Midjourney (mid) focuses on artistic-style image generation and provides an intuitive yet powerful parameter control system. More recently, OpenAI's Sora (Liu et al., 2024e) has marked a major breakthrough in video generation, capable of producing high-quality, coherent videos of up to one minute in length. By incorporating spatiotemporal consistency constraints, Sora ensures continuity across complex scenes and dynamic actions (Liu et al., 2024e).

LVMs face unique computational challenges due to their need to process high-dimensional image and video data. A single high-resolution color image may contain millions of pixels, with multiple channels of information per pixel. To efficiently handle such large-scale visual data, LVMs typically rely on parallel computing architectures such as GPUs or TPUs, leveraging parallel computing frameworks like CUDA to accelerate matrix operations. Additionally, batch processing techniques and mixed-precision training are employed to balance computational efficiency and accuracy. LVMs also exhibit high memory intensity, particularly due to the quadratic computational complexity of self-attention mechanisms with respect to sequence length, which results in substantial memory requirements when dealing with high-dimensional image and video data.

From an efficiency perspective, one of the primary challenges for vision generative models is computational complexity. Diffusion models generally require tens to hundreds of iterative denoising steps, each involving a complete forward pass through the network. The computational burden becomes even more pronounced in video generation, where an additional temporal dimension exponentially increases processing demands. To address these challenges, researchers have proposed various optimization strategies, including accelerated sampling techniques, knowledge distillation, model quantization, and sparse attention mechanisms. Furthermore, LVMs impose stringent hardware requirements, necessitating high-capacity memory, high-bandwidth data transfer, and specialized accelerators. Despite these advancements, real-time vision generation remains a formidable challenge—high-quality image synthesis often requires several seconds to tens of seconds, while video generation is even more time-intensive. Additionally, deploying LVMs on edge devices is constrained by limited computational resources and energy efficiency considerations.

**Large Vision Language Models (VLMs).** Vision-Language Models (VLMs) represent a crucial frontier in artificial intelligence, embodying advancements in multimodal intelligence. These models are designed to simultaneously process and understand both visual and linguistic information, enabling cross-modal knowledge representation and reasoning. Unlike traditional unimodal models, VLMs bridge the semantic gap between vision and language, allowing machines to perceive the world through a synergistic integration of textual and visual inputs. This capability has led to remarkable progress in tasks such as image captioning, visual question answering, and cross-modal retrieval, offering a more natural and intuitive approach to human-computer interaction.

Several representative VLMs have emerged as milestones in this domain. CLIP (Contrastive Language-Image Pre-training) (Radford et al., 2021), developed by OpenAI, leverages contrastive learning to jointly train text and image encoders, mapping features from both modalities into a shared semantic space. Pretrained on vast amounts of internet data, CLIP demonstrates exceptional zero-shot

transferability, enabling recognition of novel visual concepts solely based on textual descriptions (Radford et al., 2021; Zhao et al., 2023b). GPT-4V (OpenAI et al., 2023) extends the capabilities of LLMs to the visual domain, allowing for image-based text generation and question answering. This model not only comprehends image content but also performs complex reasoning, such as interpreting charts, analyzing scene relationships, and extracting key information from documents. Other notable models, including BLIP (Li et al., 2022), Flamingo (Alayrac et al., 2022), and LLaVA (Liu et al., 2023a), have adopted distinct architectural designs and training strategies to achieve state-of-the-art performance in vision-language understanding and generation tasks (Zhang et al., 2024a).

Architecturally, VLMs incorporate specialized components for processing different modalities, alongside mechanisms for multimodal fusion. A typical VLM architecture consists of a vision encoder (e.g., ViT (Dosovitskiy et al., 2020), ResNet (He et al., 2016)), a language encoder (e.g., BERT (Kenton & Toutanova, 2019), GPT (Radford et al., 2018; 2019; Brown et al., 2020; OpenAI et al., 2023)), and a fusion module to integrate multimodal representations. The fusion process is a fundamental challenge in VLMs and is commonly addressed through three strategies: early fusion (concatenating raw inputs), intermediate fusion (interacting after feature extraction), and late fusion (maintaining independent processing until the final decision stage). Among these, cross-modal fusion based on attention mechanisms is the most widely adopted, allowing the model to dynamically align relevant information across modalities. The complexity of such architectures imposes substantial computational demands, requiring efficient processing of high-dimensional visual data, large-scale language modeling, and real-time multimodal interactions.

Efficiency remains a significant challenge for VLMs, as multimodal processing inherently entails higher computational complexity compared to unimodal models. Vision encoders must process high-resolution images containing millions of pixels, while language encoders must capture intricate semantic structures. Moreover, attention-based fusion mechanisms—particularly cross-attention—exhibit quadratic complexity with respect to sequence length, leading to increased memory consumption and inference latency. The vast parameter scale of VLMs, such as GPT-4V, which may contain hundreds of billions of parameters, exacerbates memory constraints and computational overhead, limiting their deployment on resource-constrained devices and affecting real-time interaction performance.

To address these efficiency challenges, researchers have explored various optimization strategies. Architectural optimizations include parameter sharing, knowledge distillation, and model quantization to reduce computational and memory requirements. For inference acceleration, techniques such as sparse attention, progressive decoding, and caching mechanisms have been developed to enhance processing speed. Hardware-oriented optimizations are also critical, involving the design of specialized accelerators, optimized memory access patterns, and distributed computing frameworks. Furthermore, task-specific multimodal optimizations, such as dynamic modality selection (activating only the necessary modality processing components based on task demands) and adaptive computation (adjusting computational resource allocation based on input complexity), show promising potential in improving the efficiency and scalability of VLMs.

## H.2 LLM AND VLM FRAMEWORK CAPABILITIES

Table 17: LLM and VLM frameworks.

| Framework | Pre-train | Fine-tune | Inference |
|---|---|---|---|
| Colossal-AI | ✓ | ✓ | ✓ |
| Composer | ✓ | ✓ | ✓ |
| DeepSpeed | ✓ | ✓ | ✓ |
| FairScale | ✓ | ✓ | ✓ |
| LLM Foundry | ✗ | ✓ | ✓ |
| MegaBlocks | ✓ | ✓ | ✓ |
| Megatron | ✓ | ✓ | ✓ |
| Nanotron | ✓ | ✓ | ✓ |
| OpenLLM | ✗ | ✓ | ✓ |
| Pax | ✓ | ✓ | ✓ |
| RayLLM | ✗ | ✗ | ✓ |
| Sax | ✗ | ✗ | ✓ |
| Text Generation Inference | ✗ | ✗ | ✓ |
| vLLM | ✗ | ✗ | ✓ |

Table 18: Overview of Evaluated Large Vision Models and Large Vision Language Models.

| Model Name | Parameter | Year | Creator |
|---|---|---|---|
| **LVMs** | | | |
| Stable Diffusion 3.5 Medium | 2.5B | 2024 | Stability AI |
| Wan 2.1 T2V-1.3B | 1.3B | 2025 | Alibaba |
| **VLMs** | | | |
| Qwen2.5-VL (7B) | 7B | 2023 | Alibaba |
| QVQ-72B | 72B | 2024 | Alibaba |
| LLaVA 1.5 | 7B | 2023 | LLaVA |
| InternVL 3 (38B) | 38B | 2025 | OpenGVLab |

## H.3 OTHER MODELS LIST

### H.3.1 LARGE VISION MODELS (LVMs)

**Stable Diffusion 3.5.** Stable Diffusion 3.5 (Stability AI, 2024) (Podell et al., 2023) is the latest text-to-image diffusion model in the Stable Diffusion series, which are latent diffusion models that generate images in a compressed latent space for efficiency. Version 3.5 introduced two main variants: a Large 8.1B-parameter model capable of producing $1024\times1024$ images with high fidelity, and a Medium 2.5B-parameter model (with an improved "MMDiT-X" architecture) designed to run on consumer GPUs while still achieving up to 0.5–2 MP output resolution. Both models use a modular UNet Transformer with cross-attention to a T5 text encoder, and they support fast "Large Turbo" decoding via a distilled 4-step sampler for quicker image generation. We use the Stable Diffusion 3.5 Large and Medium models as our text-to-image baselines.

**Wan 2.1 Video Models.** Wan 2.1 (Alibaba, 2025) (Team, 2025) is a suite of open text-to-video and image-to-video generative models that achieve high-quality 480p–720p video synthesis with relatively moderate model sizes. The series includes a 14B-parameter text-to-video model and a 14B image-to-video model (trained for 720p and 480p outputs respectively), as well as a smaller 1.3B text-to-video model for efficiency. The 14B Wan 2.1 T2V model excels in complex "high motion" scenes, producing realistic physics and dynamics in its outputs, while the 1.3B variant offers a favorable trade-off, generating 480p videos in only a few minutes on standard hardware. Wan 2.1 models use a diffusion-based architecture with dual encoders for text and image inputs, and they were released under an open Apache 2.0 license to stimulate community development. We evaluate Wan 2.1's T2V-14B, T2V-1.3B, and I2V-14B models in our benchmark.

### H.3.2 VISION LANGUAGE MODELS (VLMs)

**Qwen2.5-VL.** Qwen-VL (Alibaba, 2023) (Bai et al., 2023) is a series of vision-language models built upon the Qwen LLM, endowed with visual understanding via a pretrained image encoder. The initial Qwen-VL (7B) introduced a carefully designed visual input module and a three-stage training pipeline to handle image-text alignment, enabling capabilities such as image captioning, visual question answering, grounding, and OCR reading. Its successor, Qwen-VL 2, further improved multimodal performance and introduced instruction-tuned variants (Qwen-VL-Chat). In the latest generation Qwen-VL 2.5, the model scaling is increased up to 72B parameters (dubbed Qwen-VL-Max) to further boost visual reasoning capacity. The Qwen-VL 2.5 family (3B, 7B, and 72B) achieves state-of-the-art results on a broad range of image understanding benchmarks, while remaining fully open-source.

**LLaVA 1.5.** LLaVA 1.5 (2023) (Liu et al., 2024b) is an open vision-language assistant model that connects a vision encoder with a LLaMA-based language model for interactive multimodal conversations. LLaVA uses a CLIP ViT encoder to encode images and feeds the resulting embeddings into a LLaMA chatbot, which has been fine-tuned on visual instruction data. Version 1.5 of LLaVA improved the fine-tuning procedure and dataset quality, resulting in more accurate visual understanding and more coherent dialogue responses. We use LLaVA 1.5 as a representative chat-oriented VLM, noting its efficiency: it leverages a fixed image encoder and an approximately 13B-parameter language model, avoiding the need to train a massive end-to-end multimodal model.

**QVQ-72B.** QVQ-72B (2024) (Team, 2024b) is an upcoming 72B-parameter multimodal model from Alibaba, for which only a preview is available at the moment. It is expected to combine the visual prowess of Qwen-VL-Max with the advanced reasoning of QwQ, in a model that handles both vision and language at a very large scale (72B). Due to limited official documentation, we use a placeholder description for QVQ: it is anticipated to support extremely long context multimodal inputs and serve as a testbed for scaling laws in VLM efficiency. (We will treat QVQ-72B-Preview as an experimental entry in our evaluations.)

**InternVL 3 (38B).** InternVL 3 (OpenGVLab, 2025) (Chen et al., 2024c;d;e; Zhu et al., 2025a)s the latest model in the InternVL series that adopts a native multimodal pretraining paradigm. Unlike conventional approaches that adapt a text-only LLM to multimodal settings, InternVL3 is trained jointly on both pure-text and diverse vision-language data from scratch. This unified training eliminates post-hoc alignment issues and enhances multimodal grounding. InternVL3 incorporates variable visual position encoding (V2PE) to support extended visual contexts, along with advanced post-training techniques such as supervised fine-tuning (SFT) and mixed preference optimization (MPO). Test-time scaling and a highly optimized training infrastructure further improve its performance. InternVL3-78B achieves state-of-the-art results among open-source MLLMs, scoring 72.2 on the MMMU benchmark, and shows competitive performance against proprietary models like GPT-4o and Claude 3.5. Notably, both model weights and training data are planned for public release to promote open research.

### H.4    Inference benchmark Performance

### H.5    Inference benchmark Performance

**Analysis of Inference Benchmark Performance.** Table 19 summarizes the inference performance of various models across multiple precision formats (`bfloat16`, `float16`, and `int4`) on six representative benchmarks: MMLU-Pro, BBH, GPQA, IFEval, MATH, and MUSR. Several key observations emerge from these results:

- *Impact of Model Scale.* Larger models consistently outperform smaller ones across nearly all benchmarks. For instance, Qwen2.5-32B achieves significantly higher scores compared to its smaller counterparts (7B and 14B), highlighting the effectiveness of scaling model parameters for improved inference performance.

- *Precision Trade-offs.* Lower-precision quantization (`int4`) generally introduces a modest performance degradation compared to higher-precision formats (`bfloat16` and `float16`). For example, DeepSeek-R1-Distill-Qwen-14B shows a slight drop in performance when quantized to `int4`, with MMLU-Pro decreasing from 0.4639 (`bfloat16`) to 0.4456 (`int4`). However, this degradation is relatively small, suggesting that `int4` quantization provides a favorable trade-off between computational efficiency and inference accuracy.

- *Task-specific Variability.* Different models exhibit varying strengths across benchmarks, indicating task-specific suitability. For instance, Phi-4 demonstrates strong performance on BBH and GPQA benchmarks but significantly underperforms on IFEval. Conversely, Qwen2.5 models consistently achieve high scores on IFEval, suggesting their suitability for tasks evaluated by this benchmark.

- *Consistency between `bfloat16` and `float16`.* Across most models and benchmarks, performance differences between `bfloat16` and `float16` are minimal, indicating that both formats are viable for inference on modern hardware. However, given the known hardware-level advantages of `bfloat16` on recent GPU architectures (e.g., Hopper GPUs), it remains the recommended precision format for optimal efficiency.

- *Quantization Sensitivity.* Certain benchmarks, particularly MATH, exhibit higher sensitivity to quantization. For example, Qwen2.5-14B's performance on MATH drops significantly from 0.1700 (`bfloat16`) to 0.0529 (`int4`), indicating that mathematical reasoning tasks may require higher precision to maintain accuracy.

Overall, these results underscore the importance of carefully selecting model scale and numerical precision based on specific inference tasks and efficiency constraints. Practitioners should balance the trade-offs between computational efficiency and task-specific accuracy requirements when deploying large generative models in practical scenarios.

Table 19: Evaluation Results Across Precisions - Performance Metrics.

| Model | Precision | MMLU-Pro | BBH | GPQA | IFEval | MATH | MUSR |
|---|---|---|---|---|---|---|---|
| DeepSeek-R1-Distill-Qwen-1.5B | bfloat16 | 0.1656 | 0.3471 | 0.2690 | 0.1955 | 0.1192 | 0.3553 |
| | float16 | 0.1668 | 0.3505 | 0.2754 | 0.1995 | 0.1213 | 0.3567 |
| | int8 | 0.1649 | 0.3448 | 0.2662 | 0.1971 | 0.1153 | 0.3525 |
| | fp8 | 0.1687 | 0.3521 | 0.2719 | 0.2050 | 0.1171 | 0.3554 |
| | int4 | 0.1496 | 0.3337 | 0.2529 | 0.1937 | 0.1043 | 0.3702 |
| DeepSeek-R1-Distill-Llama-8B | bfloat16 | 0.2739 | 0.4173 | 0.2974 | 0.3666 | 0.3146 | 0.3829 |
| | float16 | 0.2740 | 0.4149 | 0.2948 | 0.3675 | 0.3023 | 0.3815 |
| | int8 | 0.2691 | 0.4111 | 0.2911 | 0.3603 | 0.2978 | 0.3842 |
| | fp8 | 0.2777 | 0.4202 | 0.3004 | 0.3711 | 0.3072 | 0.3893 |
| | int4 | 0.2381 | 0.4203 | 0.2641 | 0.3510 | 0.2215 | 0.3747 |
| DeepSeek-R1-Distill-Qwen-14B | bfloat16 | 0.4639 | 0.5891 | 0.3907 | 0.4774 | 0.3751 | 0.5353 |
| | float16 | 0.4651 | 0.5877 | 0.3916 | 0.4707 | 0.3784 | 0.5340 |
| | int8 | 0.4577 | 0.5853 | 0.3824 | 0.4658 | 0.3648 | 0.5404 |
| | fp8 | 0.4710 | 0.5920 | 0.3991 | 0.4812 | 0.3803 | 0.5429 |
| | int4 | 0.4456 | 0.5766 | 0.3688 | 0.4166 | 0.2764 | 0.5327 |
| Qwen2.5-7B | bfloat16 | 0.4468 | 0.5555 | 0.3281 | 0.6619 | 0.2499 | 0.4264 |
| | float16 | 0.4461 | 0.5545 | 0.3307 | 0.6626 | 0.2574 | 0.4290 |
| | int8 | 0.4423 | 0.5499 | 0.3251 | 0.6535 | 0.2431 | 0.4296 |
| | fp8 | 0.4502 | 0.5570 | 0.3349 | 0.6721 | 0.2559 | 0.4330 |
| | int4 | 0.4187 | 0.5451 | 0.3413 | 0.6134 | 0.1501 | 0.4227 |
| Qwen2.5-14B | bfloat16 | 0.5386 | 0.6501 | 0.3737 | 0.6079 | 0.1700 | 0.4744 |
| | float16 | 0.5379 | 0.6495 | 0.3722 | 0.6266 | 0.1591 | 0.4691 |
| | int8 | 0.5332 | 0.6424 | 0.3666 | 0.6154 | 0.1550 | 0.4726 |
| | fp8 | 0.5455 | 0.6542 | 0.3800 | 0.6351 | 0.1685 | 0.4792 |
| | int4 | 0.5180 | 0.6202 | 0.3578 | 0.5878 | 0.0529 | 0.4348 |
| Qwen2.5-32B | bfloat16 | 0.5905 | 0.7038 | 0.3818 | 0.7350 | 0.4021 | 0.5008 |
| | float16 | 0.5911 | 0.7039 | 0.3798 | 0.7295 | 0.3953 | 0.5034 |
| | int8 | 0.5858 | 0.6991 | 0.3730 | 0.7217 | 0.3879 | 0.5089 |
| | fp8 | 0.5988 | 0.7115 | 0.3864 | 0.7428 | 0.4090 | 0.5095 |
| | int4 | 0.5691 | 0.6801 | 0.3828 | 0.7100 | 0.2190 | 0.4959 |
| Phi-4 | bfloat16 | 0.5284 | 0.6705 | 0.4081 | 0.0549 | 0.2554 | 0.5034 |
| | float16 | 0.5295 | 0.6710 | 0.4009 | 0.0503 | 0.2497 | 0.5021 |
| | int8 | 0.5207 | 0.6631 | 0.3920 | 0.0498 | 0.2403 | 0.4984 |
| | fp8 | 0.5332 | 0.6771 | 0.4124 | 0.0620 | 0.2521 | 0.5075 |
| | int4 | 0.5276 | 0.6679 | 0.3953 | 0.0651 | 0.2385 | 0.4756 |
| Phi-3.5-mini | bfloat16 | 0.3834 | 0.5365 | 0.3060 | 0.4231 | 0.1167 | 0.4438 |
| | float16 | 0.3828 | 0.5377 | 0.3054 | 0.4051 | 0.1216 | 0.4385 |
| | int8 | 0.3781 | 0.5232 | 0.2997 | 0.3976 | 0.1089 | 0.4438 |
| | fp8 | 0.3874 | 0.5420 | 0.3089 | 0.4341 | 0.1193 | 0.4462 |
| | int4 | 0.3382 | 0.5062 | 0.3118 | 0.3742 | 0.0482 | 0.4343 |
| Yi-34B | bfloat16 | 0.4427 | 0.5482 | 0.3455 | 0.2950 | 0.0443 | 0.4145 |
| | float16 | 0.4456 | 0.5447 | 0.3417 | 0.3003 | 0.0435 | 0.4132 |
| | int8 | 0.4399 | 0.5376 | 0.3391 | 0.2892 | 0.0401 | 0.4140 |
| | fp8 | 0.4480 | 0.5539 | 0.3488 | 0.3101 | 0.0470 | 0.4160 |
| | int4 | 0.4230 | 0.5137 | 0.3329 | 0.3198 | 0.0373 | 0.4053 |

### H.6 HYPERPARAMETER SETTINGS

To ensure reproducibility and provide a comprehensive reference for practitioners, this section details the hyperparameter configurations used across our experiments. These settings were carefully selected to balance performance and efficiency considerations while maintaining consistency across different model architectures and efficiency techniques.

**Architecture Efficiency of Models Hyperparameter.** For each Transformer's Model:

- **0.5B model**: 24 layers, hidden dimension 896, 14 attention heads, intermediate size 4864, 2 key-value heads, maximum position embeddings 32768, extra vocabulary size 293, RMS normalization epsilon 1e-6.

- **1.5B model**: 28 layers, hidden dimension 1536, 12 attention heads, intermediate size 8960, 2 key-value heads, maximum position embeddings 32768, extra vocabulary size 293, RMS normalization epsilon 1e-6.

- **3B model**: 36 layers, hidden dimension 2048, 16 attention heads, intermediate size 11008, 2 key-value heads, maximum position embeddings 32768, extra vocabulary size 293, RMS normalization epsilon 1e-6.

### H.7 NORMALIZATION METHOD FOR DRAWING FIGURES

#### H.7.1 NORMALIZATION METHODOLOGY FOR EFFICIENCY METRICS

To facilitate intuitive comparisons across various model architectures and optimization methods, we employed normalization techniques to standardize the diverse efficiency metrics presented in Figures 3 and 5. Specifically, all raw metrics, including Perplexity (PPL), Average Memory Utilization (AMU), Average Latency (AL), Tokens Throughput (TT), and Average Energy Consumption (AEC), are converted into normalized values ranging from 0.1 to 1.0.

For metrics where lower values denote better performance (such as PPL, AMU, AL, and AEC), we applied the following normalization formula:

$$\text{Normalized Value} = 0.1 + 0.9 \times \frac{\text{Maximum Value} - \text{Current Value}}{\text{Maximum Value} - \text{Minimum Value}} \tag{16}$$

Conversely, for metrics where higher values are preferable (e.g., Tokens Throughput, TT), the normalization was performed using:

$$\text{Normalized Value} = 0.1 + 0.9 \times \frac{\text{Current Value} - \text{Minimum Value}}{\text{Maximum Value} - \text{Minimum Value}} \tag{17}$$

This systematic normalization ensures consistency in the interpretation of efficiency metrics across models and methods, allowing for clearer insights into the trade-offs between performance and computational resources as illustrated in Figures 3 and 5.

#### H.7.2 EFFICIENCY SCORE COMPUTATION

The Efficiency Score shown in Figure 4 is calculated using a weighted harmonic combination of normalized resource metrics. Specifically, the Efficiency Score integrates Average Memory Utilization (AMU), Peak Computational Utilization (PCU), Average Latency (AL), Sample Throughput (ST), and Average Energy Consumption (AEC) through the following formula:

$$\text{Efficiency Score} = 0.2 \cdot \frac{\min(\text{AMU})}{\text{AMU}} + 0.2 \cdot \frac{\min(\text{PCU})}{\text{PCU}} + 0.2 \cdot \frac{\text{AL}}{\min(\text{AL})} + 0.2 \cdot \frac{\min(\text{ST})}{\text{ST}} + 0.2 \cdot \frac{\min(\text{AEC})}{\text{AEC}} \tag{18}$$

This balanced combination of metrics ensures a comprehensive assessment of computational and training efficiency, emphasizing optimal use of resources and performance trade-offs.

# I    USE OF LLMS

In the preparation of this manuscript, we employed large language models (LLMs), specifically **GPT-5** and **GPT-4o**, solely for the purpose of polishing and refining the writing. These models assisted in improving readability, grammar, and stylistic clarity of the text. Importantly, they were not involved in the design, construction, implementation, or evaluation of the proposed methods and experiments. All conceptual contributions, dataset construction, algorithmic design, and experimental analyses were carried out independently by the authors.

