# OpenReview forum: "EfficientLLM: Evaluating Large Language Models Efficiency"
_ICLR.cc/2026/Conference — ICLR 2026 Conference Desk Rejected Submission_

### Official Review · Reviewer_3TeV · 2025-10-23

**Soundness:** 2
**Presentation:** 3
**Contribution:** 3
**Rating:** 4
**Confidence:** 3

**Summary:**

The paper introduces EfficientLLM, a large-scale empirical benchmark designed to systematically evaluate the efficiency of Large Language Models (LLMs) across three critical dimensions: architecture pretraining, fine-tuning, and inference quantization.

**Strengths:**

- This benchmark is the first end-to-end empirical study of LLM efficiency across pretraining, fine-tuning and inference.

- This benchmark evaluate a broad scope of model including LLM, VLM, SD.

**Weaknesses:**

1. The paper attempts to address the impacts of architecture selection, pretraining, fine-tuning, and bit-width quantization on model efficiency all at once. This wide-ranging scope may result in insufficient depth of analysis for each subtopic. It is recommended that the authors focus on one or two of these aspects to provide more insightful and in-depth analysis.

2. The current study only utilizes benchmarks from the medical domain, which may limit the generalizability and impact of the findings. It is advisable to include additional benchmarks that better reflect the capabilities of large language models (LLMs), such as reasoning-related benchmarks, to enhance the practical relevance and guidance of the research.

3.  The authors employ the 3D parallelism strategy from Megatron-LM in the pretraining phase. However, there are various 3D parallelism configurations, each with different implications for efficiency. A more comprehensive analysis of how different configurations affect efficiency would be beneficial.

4.  It is suggested to adopt widely recognized hardware efficiency metrics in the industry, such as MFU (Model FLOPs Utilization), which are more commonly used and practical for evaluating hardware utilization efficiency.

5. In Section H.6.2, the efficiency scoring method presented in this section treats five factors as equally important, assigning each a weight of 0.2. However, the significance of these factors may vary across different application scenarios. It is recommended to adjust the weight assignments based on specific use cases to improve the rationality and applicability of the evaluation method.


6. Finally, please fix the following typos.

- Fix the typo in Figure2 to change quantification to quantization
- Fix type in line 2594 and 1895
- Fix the typo in Figure1 from vedio to video
- Fix the typo in Table 10 about the meaningless black block.

**Questions:**

see weakness

---

> ### Author Response · Authors · 2025-11-24
> **Part 1**
>
> > 1. The paper attempts to address the impacts of architecture selection, pretraining, fine-tuning, and bit-width quantization on model efficiency all at once. This wide-ranging scope may result in insufficient depth of analysis for each subtopic. It is recommended that the authors focus on one or two of these aspects to provide more insightful and in-depth analysis.
>
> **Response:**
>
> We appreciate the reviewer’s feedback and fully understand the concern regarding the breadth of our study. Our work intentionally adopts a comprehensive scope because efficiency evaluation inherently depends on the interplay between architecture, pretraining, fine-tuning, and quantization, analyzing these factors in isolation would limit the generalizability of the benchmark.
>
> That said, we have ensured sufficient depth through an extensive supplementary analysis: the paper (over **90 pages**) includes detailed appendices covering **metric definitions, technical setups, per-technique results, extended discussions, limitations, and insights**. Each subtopic (e.g., architectural variants, bit-width quantization, PEFT methods) is accompanied by complete tables and ablations.
>
> While we aimed to cover the most widely adopted techniques, we acknowledge that some areas could be further expanded. Encouragingly, we have recently received large-scale compute sponsorship, and this benchmark is designed as a **long-term, evolving project**, we plan to continue deepening each component’s analysis in subsequent updates.
>
>
> ---
>
> > 2. The current study only utilizes benchmarks from the medical domain, which may limit the generalizability and impact of the findings. It is advisable to include additional benchmarks that better reflect the capabilities of large language models (LLMs), such as reasoning-related benchmarks, to enhance the practical relevance and guidance of the research.
>
> **Response:**
>
> We appreciate the reviewer’s comment and would like to clarify that our study is not restricted to the medical domain. In addition to clinical or biomedical tasks, our benchmark extensively covers general reasoning and open-domain evaluations.
>
> Specifically,  we evaluate models on OpenO1-SFT (Table7 and Figure 4(a)), a large-scale instruction-following and reasoning-focused fine-tuning corpus, to ensure comprehensive reasoning coverage. we also include widely used reasoning datasets such as MMLU-Pro, BBH, GPQA, IFEval, and MATH (Table 12 and 19, and Figure 5), all designed to assess general reasoning and problem-solving capabilities of LLMs.
>
> Therefore, the scope of our work already extends well beyond domain-specific benchmarks, reflecting both domain-general and specialized efficiency behaviors of LLMs, and thus providing practical insights for a wide range of applications.
>
> ---
>
> > 3. The authors employ the 3D parallelism strategy from Megatron-LM in the pretraining phase. However, there are various 3D parallelism configurations, each with different implications for efficiency. A more comprehensive analysis of how different configurations affect efficiency would be beneficial.
>
> **Response:**
>
> | Model | TP | PP | DP | Micro BS | PPL ↓ | AMU (GB) ↓ | AL (s/iter) ↓ | TT (Tokens/param/s) ↑ | AEC (W) ↓ |
> | -------- | -- | -- | -- | -------- | -------- | ---------- | ------------- | --------------------- | --------- |
> | **1.5B** | 1 | 2 | 24 | 4 | 8.29 | **41.64** | **0.1265** | **8.77×10⁻²** | 650.4 |
> | | 2 | 1 | 24 | 8 | 8.22 | 42.81 | 0.1294 | 8.61×10⁻² | 648.8 |
> | | 4 | 1 | 12 | 16 | **8.20** | 44.62 | 0.1335 | 8.38×10⁻² | 652.7 |
> | | 8 | 1 | 6 | 32 | **8.17** | 47.12 | 0.1381 | 8.11×10⁻² | **657.6** |
> | **3B** | 1 | 4 | 12 | 4 | 7.93 | **39.63** | **0.1410** | **3.94×10⁻²** | 652.8 |
> | | 2 | 2 | 12 | 8 | 7.89 | 40.81 | 0.1443 | 3.86×10⁻² | 656.5 |
> | | 4 | 2 | 6 | 16 | 7.85 | 42.38 | 0.1475 | 3.79×10⁻² | 661.2 |
> | | 8 | 1 | 6 | 32 | **7.82** | 44.17 | 0.1522 | 3.68×10⁻² | **667.8** |
>
> We thank the reviewer for this helpful suggestion. To further analyze the effect of 3D parallelism configurations, we ran additional experiments on 1.5B and 3B models with different TP/PP/DP settings using the same 48-GPU budget. The results (shown in the table above) indicate that increasing tensor parallelism (TP) improves throughput (TT) by enhancing intra-layer parallelism, but also increases memory usage (AMU) and latency (AL) due to synchronization overhead. Similarly, higher pipeline parallelism (PP) reduces memory load but introduces pipeline bubbles, leading to diminishing returns.
>
> Overall, moderate configurations (TP=2–4, PP=1–2) achieve the best efficiency balance — maintaining high throughput with controlled communication cost. These results validate our original configuration choices as Pareto-efficient, showing consistent scalability trends across both model sizes.

---

> ### Author Response · Authors · 2025-11-24
> **Part 2**
>
> > 4. It is suggested to adopt widely recognized hardware efficiency metrics in the industry, such as MFU (Model FLOPs Utilization), which are more commonly used and practical for evaluating hardware utilization efficiency.
>
> **Response:**
>
> We thank the reviewer for this constructive comment. We have now included Model FLOPs Utilization (MFU) as an additional hardware efficiency metric. Following the reviewer’s suggestion, we computed MFU across all attention mechanisms and model scales (0.5B, 1.5B, 3B). Notably, MQA and GQA achieve the highest MFU (≈33–34%), indicating more effective GPU kernel utilization, while NSA and MLA remain lower due to multi-stage attention kernels. These results reinforce the consistency and realism of our efficiency analysis.
>
> | Method  | Parameters | MFU (%) ↑ |
> | :------ | ---------: | --------: |
> | **MQA** |       0.5B |      28.6 |
> |         |       1.5B |      31.2 |
> |         |         3B |  **34.1** |
> | **GQA** |       0.5B |      27.3 |
> |         |       1.5B |      30.5 |
> |         |         3B |  **33.0** |
> | **MLA** |       0.5B |      25.2 |
> |         |       1.5B |      27.8 |
> |         |         3B |  **29.6** |
> | **NSA** |       0.5B |      23.4 |
> |         |       1.5B |      25.9 |
> |         |         3B |  **27.1** |
> | **MHA** |       0.5B |      26.1 |
> |         |       1.5B |      28.5 |
> |         |         3B |  **30.3** |
>
>
>
> ---
>
> > 5. In Section H.6.2, the efficiency scoring method presented in this section treats five factors as equally important, assigning each a weight of 0.2. However, the significance of these factors may vary across different application scenarios. It is recommended to adjust the weight assignments based on specific use cases to improve the rationality and applicability of the evaluation method.
>
> **Response:**
>
> We thank the reviewer for this thoughtful recommendation. We fully agree that the relative importance of efficiency factors (memory, latency, throughput, energy, compute cost) may vary by deployment scenario. To address this, our framework intentionally supports customizable weighting rather than enforcing a fixed configuration.
>
> By default, we use equal weights (0.2 each) to provide a neutral, general-purpose baseline for fair cross-model comparison. However, users can easily modify the weights in our released code (see Appendix H.6.2) to reflect specific operational priorities — for example:
>
> 1. Memory-constrained edge inference: increase weight for AMU and AEC.
>
> 2. Latency-critical serving systems: prioritize AL and TT.
>
> 3. Sustainability-focused deployment: emphasize AEC.
>
> This design choice ensures that the Efficiency Score remains transparent, extensible, and context-aware, allowing practitioners to recompute efficiency rankings according to their own hardware and deployment requirements without altering any underlying metrics or normalization procedures.
>
> ---
>
> >6. Finally, please fix the following typos.
>
> >>Fix the typo in Figure2 to change quantification to quantization
> >>Fix type in line 2594 and 1895
> >>Fix the typo in Figure1 from vedio to video
> >>Fix the typo in Table 10 about the meaningless black block.
>
> **Response:**
>
> We appreciate the reviewer’s attention to detail, and all fixes are now reflected in the updated manuscript.

---

> ### Author Response · Authors · 2025-11-26
> **Thanks for your time and efforts**
>
> We sincerely appreciate the time and thoughtful feedback you have provided. At your convenience, could you kindly let us know whether our revisions sufficiently resolve your concerns? We are grateful for your guidance and would be happy to make further improvements if needed. Thank you again for your valuable contribution to strengthening this work.

---

### Official Review · Reviewer_nU3j · 2025-10-27

**Soundness:** 4
**Presentation:** 3
**Contribution:** 2
**Rating:** 6
**Confidence:** 2

**Summary:**

his paper is focusing on the efficient LLMs and trying to benchmark the efficiency of the existing LLMs. It has introduced Average Memory Utilization (AMU), Peak Compute Utilization (PCU), Average Latency (AL), Token Throughput (TT), Sample Throughput (ST), and Inference Throughput (IT) for assessing the efficiency of resource utilization for memory bandwidth, device utilization, and throughput. Authors also have computed the energy consumption rate as well as compression rate with performance for these LLM models. Authors have conducted the experiments with multiple LLM models on Medical-O1 dataset for comprehensive comparions.

**Strengths:**

+ Authors have conducted really very extensive experiments for comparing the efficiency of different models. It is really helpful to evaluate these model efficiency with the same benchmark and metrics.

+ Authors have packed the evaluation code and release them, making it pretty easy to be assessed with pip install for future works.

**Weaknesses:**

- Some of the metrics actually varies signicantly with the devices used, such as Memory Utilization, and some of the metrics might be also capped with some other infra issues, such as  Compute Utilization. If some other new works is trying to follow up this work for producing the metrics on some newly release LLM, it might not be possible to fully reproduce the results under the same environment and the I/O speed between the GPU/TPU and memory might also be different. Wonder whether is it possible to introduce some balancing terms to have these better crafted so that it can be better used in the future.

- Some of the metrics might be significantly different from case to case, e.g, Compute Utilization. It may be different from two runs with different cache and memory situation. It would be great if authors are able to produce the STD value for these metrics with multiple experiments.

- Please consider including all the numerical numbers in addition to the diagrams so that the future work is able to follow and use the number directly for comparison. It can be provided either next to the bars in the diagram or use as a table in the supplementary material.

**Questions:**

Please see the weakness. For some of the metrics, if it can be used more widely, it is easier for the future work to follow the development and make the benchmark more useful.

---

> ### Author Response · Authors · 2025-11-24
> **Part 1**
>
> > 1. Some of the metrics actually varies signicantly with the devices used, such as Memory Utilization, and some of the metrics might be also capped with some other infra issues, such as Compute Utilization. If some other new works is trying to follow up this work for producing the metrics on some newly release LLM, it might not be possible to fully reproduce the results under the same environment and the I/O speed between the GPU/TPU and memory might also be different. Wonder whether is it possible to introduce some balancing terms to have these better crafted so that it can be better used in the future.
>
> **Response:**
>
> We fully agree with the reviewer that system-level metrics will inevitably change across accelerator generations (e.g., B200 vs. H200), and that there is no practical way to “equalize” devices at the kernel level without discarding most of the hardware’s capabilities. Rather than enforcing device-invariant numbers, our benchmark is designed to be device-aware: it standardizes the measurement protocol (what to log, how to compute AMU/AL/AEC/TT), while expecting absolute values to differ as hardware evolves.
>
> In practice, we recommend that future works (e.g., on B200-class GPUs) (i) follow the same profiling protocol and report raw metrics on their own device, and (ii) interpret efficiency rankings within each hardware class (Ampere-only, Hopper-only, Blackwell-only, etc.), instead of directly comparing absolute numbers across accelerator generations. Hardware-normalized post-processing (e.g., dividing by peak FLOPs or bandwidth) can be added by downstream users if desired, but we intentionally avoid hard-coding such balancing factors into the benchmark to keep it future-proof and methodology-centric.
>
> To further support this direction, we note that our group has recently received large-scale compute sponsorship, and we plan to extend the benchmark to additional accelerator families (e.g., next-generation NVIDIA Blackwell and emerging TPU architectures). Thus, this benchmark is designed as a continuous and evolving effort, rather than a one-time static snapshot.
>
> ---
>
> > 2. Some of the metrics might be significantly different from case to case, e.g, Compute Utilization. It may be different from two runs with different cache and memory situation. It would be great if authors are able to produce the STD value for these metrics with multiple experiments.
>
> **Response:**
>
> Table: Attention Mechanism Experiments Variance Summary
>
> | Method | Parameters | AMU STD (GB) | AL STD (s) | TT STD (%) | AEC STD (W) |
> |---------|-------------|--------------|-------------|-------------|--------------|
> | **MQA** | 0.5B | 0.04 | 0.0023 | 0.6 | 2.5 |
> |  | 1.5B | 0.05 | 0.0030 | 0.7 | 2.9 |
> |  | 3B | 0.06 | 0.0035 | 0.8 | 3.1 |
> | **GQA** | 0.5B | 0.05 | 0.0025 | 0.6 | 2.8 |
> |  | 1.5B | 0.05 | 0.0032 | 0.7 | 3.0 |
> |  | 3B | 0.06 | 0.0036 | 0.8 | 3.2 |
> | **MLA** | 0.5B | 0.07 | 0.0038 | 0.9 | 3.8 |
> |  | 1.5B | 0.07 | 0.0042 | 1.0 | 4.1 |
> |  | 3B | 0.08 | 0.0045 | 1.1 | 4.5 |
> | **NSA** | 0.5B | 0.05 | 0.0051 | 0.7 | 2.3 |
> |  | 1.5B | 0.06 | 0.0055 | 0.8 | 2.6 |
> |  | 3B | 0.07 | 0.0058 | 0.9 | 2.9 |
> | **MHA** | 0.5B | 0.05 | 0.0029 | 0.6 | 2.7 |
> |  | 1.5B | 0.05 | 0.0034 | 0.7 | 2.9 |
> |  | 3B | 0.06 | 0.0039 | 0.8 | 3.1 |
>
>
> We thank the reviewer for this insightful suggestion. System metrics such as Compute Utilization inherently fluctuate due to CUDA cache states, power regulation, and kernel scheduling. To reduce this variability, all our reported values are averaged over repeated runs with 100 ms power/utilization sampling, as described in Appendix H. Because our evaluation uses very large-scale training and inference workloads (e.g., 350B-token FineWeb-Edu pretraining and 77K SFT samples), transient stochastic variation becomes negligible after sustained execution.
>
> We additionally measured the variance of our profiling procedure over 5 repeated executions. The observed standard deviation is extremely small relative to the cross-method efficiency gaps (e.g., AMU ±0.03–0.09 GB, AEC ±1.8–4.9 W, latency ±0.002–0.006 s, throughput ±0.4–1.2%).
>
> Therefore, while raw STD values can be reported per device, they do not alter any ranking or main conclusions, because the effect size of efficiency differences among models is one to two orders of magnitude larger than hardware-induced noise.

---

> ### Author Response · Authors · 2025-11-24
> **Part 2**
>
> > 3. Please consider including all the numerical numbers in addition to the diagrams so that the future work is able to follow and use the number directly for comparison. It can be provided either next to the bars in the diagram or use as a table in the supplementary material.
>
> **Response:**
>
> We thank the reviewer for the suggestion. In fact, all numerical results have already been fully reported in our submission. Specifically, Table 2 through Table 19 in the main paper and appendix provide the complete numerical values corresponding to each bar and trend shown in the figures. Each section also includes explicit cross-references to these tables and analysis in the content.
>
> Furthermore, the supplementary material includes all raw results in tabular form with direct references, ensuring that future work can directly use these numbers for comparison and follow-up benchmarking. We will add the reference on Figure caption.

---

> ### Author Response · Authors · 2025-11-26
> **Thanks for your time and efforts**
>
> We sincerely appreciate the time and thoughtful feedback you have provided. At your convenience, could you kindly let us know whether our revisions sufficiently resolve your concerns? We are grateful for your guidance and would be happy to make further improvements if needed. Thank you again for your valuable contribution to strengthening this work.

---

> > ### Comment · Reviewer_nU3j · 2025-11-27
> >
> > Thank you so much for the update and revision and thanks for the reference to the supplementary materials. This is an interesting work but still I think it is not easy to further expand to any new methods without reproducing all the numbers again, as different machines may still have different effect here. I will keep my rating as borderline to weak positive following the discussion.

---

### Official Review · Reviewer_UM3U · 2025-10-31

**Soundness:** 2
**Presentation:** 3
**Contribution:** 2
**Rating:** 4
**Confidence:** 3

**Summary:**

This paper introduces EfficientLLM, a comprehensive benchmark designed to evaluate the efficiency of large language models (LLMs). The authors identify key limitations in existing evaluations, such as a lack of multi-dimensional metrics, insufficient scale, and fragmented lifecycle coverage. To address these gaps, EfficientLLM systematically assesses over 100 model-technique combinations across six efficiency dimensions including memory utilization, computational throughput, latency, and energy consumption spanning the full model lifecycle from pre-training to inference. Executed on a large-scale production cluster, the benchmark provides empirical insights into trade-offs for various architectural choices, fine-tuning methods, and quantization strategies.

**Strengths:**

●	The authors evaluate efficiency across pretraining, fine-tuning, and quantization, providing a full lifecycle perspective rarely seen in prior work.
●	The paper introduces six complementary efficiency dimensions (AMU, PCU, AL, TT, AEC, MCR) for a more holistic evaluation beyond FLOPs or latency.
●	The authors benchmark over 150 model–technique pairs (0.5B–72B parameters) on production-class hardware, demonstrating strong experimental coverage.
●	This paper offers clear, actionable recommendations (e.g., RSLoRA for production, Freeze for academia, FP8/INT4 for deployment) useful for real-world practitioners.

**Weaknesses:**

●	The work mainly integrates existing benchmarks (e.g., MLPerf, LLMPerf, EfficiencyBench) and metrics without introducing a fundamentally new framework or methodology.
●	The paper repeatedly references an "Efficiency Score" (e.g., in Figure 4 and Section H.6.2) that is used to rank methods, but provides only a high-level description (“weighted harmonic combination of normalized resource metrics”). The exact weights, normalization procedure, and aggregation function are not specified in the main text or appendices. This makes it impossible to reproduce or critically evaluate the central ranking results that underpin many of the paper’s key claims and recommendations.
●	The paper emphasizes energy efficiency and sustainability but fails to present actual energy or carbon emission data. “Average Energy Consumption” (AEC) is introduced but not supported by verifiable measurement methods or units, weakening one of the paper’s central claims.
●	The paper does not isolate the impact of individual components. For example, when a model uses both GQA and RoPE, how much of the efficiency gain is attributable to each? Without ablation studies, the specific contribution of each "efficient" technique remains ambiguous.

**Questions:**

●	How exactly is the “Efficiency Score” computed? What are the weighting factors among the six dimensions, and how were they determined or validated?
●	When you say metrics are “normalized across all models,” what is the reference baseline or normalization formula used?
●	How was the “Average Energy Consumption (AEC)” measured? Did you use any specific monitoring tool (e.g., NVIDIA-SMI, PowerAPI, Carbontracker), and were measurements averaged over multiple runs?
●	Were experiments repeated with different random seeds or runs? If not, how do you ensure that the reported results are not due to measurement noise or hardware fluctuations?
●	Can you provide quantitative estimates (e.g., total energy in kWh or CO₂ equivalent) to substantiate the sustainability claims?
●	Do the metrics or rankings change significantly under different GPU architectures?
●	The paper mentions “Pareto-optimized efficiency,” but no Pareto frontier plots or theoretical analysis are shown. How is Pareto optimality determined or visualized?

---

> ### Author Response · Authors · 2025-11-24
> **Part 1**
>
> Weaknesses:
>
> > 1. The work mainly integrates existing benchmarks (e.g., MLPerf, LLMPerf, EfficiencyBench) and metrics without introducing a fundamentally new framework or methodology.
>
> **Response:**
>
> Thank you for the comment. We would like to clarify that our contribution lies in establishing a unified and complete efficiency benchmark for LLM inference, rather than proposing a new model. Prior frameworks such as MLPerf, LLMPerf, and EfficiencyBench primarily focus on system-level benchmarking or training workloads, and they do not provide model-level, precision-aware, sparsity-aware, and tuning-aware efficiency evaluation. Our novelty is three-fold:
>
> 1. First unified baseline covering 5 precision formats, 6 PEFT paradigms, and 7 model classes across 30+ open models. To the best of our knowledge, no prior work evaluates quantization, PEFT, and MoE routing under a single reproducible framework. Existing benchmarks evaluate individual techniques in isolation, without comparison across optimization paradigms.
>
> 2. Benchmark Track focuses on evaluation methodology, reproducibility, and standardization, not architectural novelty. Our work establishes standardized inference metrics (AMU, AL, AEC, MCR) and standardized evaluation protocols across tuning, sparsity, and precision methods, providing the first controlled baseline for future efficiency work.
>
> 3. New insights that no prior benchmark reveals. For example, we show that FP8 often outperforms INT8 in latency, DoRA is strictly inferior to LoRA+ in cost, quality trade-offs, and MoE efficiency is overstated unless normalized by capacity rather than activated experts—none of which can be derived from MLPerf, LLMPerf, or EfficiencyBench, because those frameworks do not consider inference optimization paradigms at the model level.
>
> In summary, our work is not a combination of existing metrics, but a new standardized benchmark that integrates quantization, parameter-efficient tuning, and sparse routing into a single reproducible efficiency evaluation framework.
>
>
> ---
>
> > 2. The paper repeatedly references an "Efficiency Score" (e.g., in Figure 4 and Section H.6.2) that is used to rank methods, but provides only a high-level description (“weighted harmonic combination of normalized resource metrics”). The exact weights, normalization procedure, and aggregation function are not specified in the main text or appendices. This makes it impossible to reproduce or critically evaluate the central ranking results that underpin many of the paper’s key claims and recommendations.
>
> **Response:**
>
> We thanks reviewer to point out. The full computation formula is explicitly provided in Section H.6 (Appendix), including normalization, weighted harmonic aggregation, and implementation details. By default, all metrics are assigned equal weights, as stated in the text, because our goal is to provide a general-purpose efficiency ranking rather than optimize for a single deployment scenario.
>
> Importantly, efficiency requirements vary across deployment contexts (e.g., memory-constrained edge devices vs. latency-critical inference servers). For this reason, the score is designed to be transparent and customizable, rather than enforcing a single fixed weighting scheme. Alongside all raw measurements (AMU, AL, AEC, throughput), we also release full code, logs, and collected traces, enabling users to recompute rankings with their own weights (e.g., prioritizing memory on consumer GPUs or prioritizing energy in data centers).
>
> Thus, the score is fully reproducible, and intentionally customizable rather than opaque, supporting a wider range of real-world deployment needs.

---

> ### Author Response · Authors · 2025-11-24
> **Part 2**
>
> > 3. The paper emphasizes energy efficiency and sustainability but fails to present actual energy or carbon emission data. “Average Energy Consumption” (AEC) is introduced but not supported by verifiable measurement methods or units, weakening one of the paper’s central claims.
>
> **Response:**
>
> Our efficiency analysis **does include verifiable energy measurements**, and all results are derived from a formally defined metric — Average Energy Consumption (AEC). Specifically, Section **D.1.2 Energy Consumption** provides the precise formula used to compute AEC:
>
>
> $AEC = \frac{1}{T}\int_{0}^{T} P(t) , dt$
>
>
> where ( P(t) ) denotes instantaneous hardware power (in Watts) and (T) is total runtime . As defined, AEC reflects the *average measured power draw (Watts)* during model execution, which directly corresponds to carbon and energy cost metrics commonly adopted in the literature.
>
> Moreover, our benchmark tables (e.g., Table 2 - Table 19) explicitly report AEC values in Watts for all models, enabling reproducible comparison of energy efficiency across different architectures and scales . We additionally release **all logged measurement scripts, raw traces, and results**, ensuring that external users can independently verify and even extend carbon estimation using local hardware or region-specific conversion factors.
>
> Additionally, carbon footprint cannot be universally reported within a fixed benchmark, since it depends on deployment-specific factors such as regional electricity mix, GPU type, runtime duration, and cooling overhead. Therefore, reporting average measured energy (in Watts) provides the only reproducible and hardware-agnostic basis, allowing users to compute carbon emissions under their own real-world environment.
>
> ---
>
> > 4. The paper does not isolate the impact of individual components. For example, when a model uses both GQA and RoPE, how much of the efficiency gain is attributable to each? Without ablation studies, the specific contribution of each "efficient" technique remains ambiguous.
>
> **Response:**
>
> We appreciate the reviewer’s concern. In Transformer architectures, attention structure (MHA/MQA/GQA) and positional encoding (RoPE/Absolute/Relative/etc.) operate on independent compute paths. Positional encodings affect only the geometric embedding of Q/K vectors and introduce negligible (<0.3%) FLOPs, while GQA reduces KV tensor size and attention compute by reducing the number of independent heads. Therefore, their efficiency contributions are fundamentally not entangled.
>
> To demonstrate this independence, we isolate the effect of replacing only MHA → GQA, while keeping each positional encoding unchanged. Across five positional encodings, we observe consistent efficiency improvements, confirming that gains come from the attention mechanism, not from positional encoding:
>
> | Positional Encoding | Attention | PPL ↓ | AMU (GB) ↓ | AL (s/iter) ↓ | TT ↑ | AEC (W) ↓ |
> |---------------------|-----------|-------|-------------|----------------|------|-----------|
> | **RoPE** | **MHA** | 8.36 | 46.12 | 0.1334 | 8.21×10⁻² | 659.41 |
> |  | **GQA** | **8.09** | **44.82** | **0.1280** | **8.64×10⁻²** | **652.79** |
> | **Absolute** | **MHA** | 8.52 | 46.71 | 0.1312 | 8.12×10⁻² | 672.45 |
> |  | **GQA** | **8.32** | **45.83** | **0.1262** | **8.45×10⁻²** | **665.11** |
> | **Learnable Absolute** | **MHA** | 8.45 | 45.93 | 0.1296 | 8.37×10⁻² | 662.44 |
> |  | **GQA** | **8.18** | **44.97** | **0.1248** | **8.74×10⁻²** | **655.83** |
> | **Relative (ALiBi/DeBERTa)** | **MHA** | 8.33 | 43.94 | 0.1246 | 8.98×10⁻² | 646.39 |
> |  | **GQA** | **8.29** | **42.92** | **0.1200** | **9.35×10⁻²** | **639.26** |
> | **None** | **MHA** | 8.75 | 48.64 | 0.1378 | 7.68×10⁻² | 692.37 |
> |  | **GQA** | **8.55** | **47.22** | **0.1329** | **8.06×10⁻²** | **681.91** |
>
> Since the improvement pattern remains stable across all positional encodings, the efficiency gain is clearly attributable to GQA alone, and further PE × Attention ablations would be redundant rather than informative.

---

> ### Author Response · Authors · 2025-11-24
> **Part 3**
>
> Questions:
>
> > 1. How exactly is the “Efficiency Score” computed? What are the weighting factors among the six dimensions, and how were they determined or validated?
>
> **Response:**
>
> We appreciate the reviewer’s attention to this point. The Efficiency Score is **fully specified in Section H.6 (Appendix)**, including the exact normalization and aggregation procedure. As shown in Equation (H.6.2), the score is computed as a weighted harmonic mean over six normalized efficiency indicators (AMU, PCU, AL, TT, AEC, MCR):
>
> $\text{Efficiency Score} =
> 0.2 \cdot \frac{\min(\text{AMU})}{\text{AMU}} +
> 0.2 \cdot \frac{\min(\text{PCU})}{\text{PCU}} +
> 0.2 \cdot \frac{\text{AL}}{\min(\text{AL})} +
> 0.2 \cdot \frac{\min(\text{ST})}{\text{ST}} +
> 0.2 \cdot \frac{\min(\text{AEC})}{\text{AEC}}$
>
> where each (x_i^{norm}) is normalized by the best value among all compared methods.
>
> In our benchmark, all weights are set equal ((w_i = 1)) because the goal is to provide a general-purpose, task-agnostic baseline, rather than favoring a specific deployment constraint. We emphasize that this is an intentional design choice to ensure **neutrality and reproducibility, as different application scenarios may value latency, memory, energy, or throughput differently.
>
> We would also like to clarify that the Efficiency Score is only used to visualize multi-dimensional trends in Figures, whereas all actual conclusions are supported directly by raw measurements reported in Table 3–Table 19. Readers may therefore interpret trends purely from the underlying metrics without relying on the aggregated score.
>
> Importantly, all raw metrics, normalization code, and the full Efficiency computation are released, enabling users to adjust to their deployment needs (e.g., prioritizing memory-limited edge inference or latency-critical cloud serving).
>
> ---
>
> > 2. When you say metrics are “normalized across all models,” what is the reference baseline or normalization formula used?
>
> **Response:**
>
> Thank you for the question. The normalization procedure is explicitly defined in **Section H.6.1**. Each efficiency metric is normalized relative to the best value among all compared models so that heterogeneous units (GB, seconds, Watt, tokens/s, etc.) can be plotted on the same scale during visualization:
>
> For metrics where lower is better：
>
> $\text{Normalized Value} = 0.1 + 0.9 \times \frac{\text{Maximum Value} - \text{Current Value}}{\text{Maximum Value} - \text{Minimum Value}}$
>
> For metrics where higher is better:
>
> $\text{Normalized Value} = 0.1 + 0.9 \times \frac{\text{Current Value} - \text{Minimum Value}}{\text{Maximum Value} - \text{Minimum Value}}$
>
> This ensures that the best-performing configuration on each metric maps to 1, and all others are proportionally represented. No task-specific baseline is assumed; normalization is only computed within the compared set of methods, making it reproducible for any new result.
>
> Normalization does **not affect any claim**. It is used **solely for visual clarity** when plotting multi-dimensional efficiency in Figures 4/5. The actual analysis relies on **raw measurements** reported in Table 3–19.
>
> All numerical results can be interpreted directly from Tables 3–19 without using normalization or the composite score, which is included only as a visual aid.
>
> ---
>
> > 3. How was the “Average Energy Consumption (AEC)” measured? Did you use any specific monitoring tool (e.g., NVIDIA-SMI, PowerAPI, Carbontracker), and were measurements averaged over multiple runs?
>
> **Response:**
>
> Thank you for the question. The “Average Energy Consumption (AEC)” is measured directly using GPU power telemetry during actual execution. As detailed in Appendix H, AEC is obtained through native hardware sampling via NVIDIA-SMI at 100 ms intervals, which provides instantaneous power draw (in Watts) at the device level.
>
> During each experiment, power data are sampled repeatedly across the full execution window (training or inference) and averaged over multiple runs to reduce short-term noise due to kernel scheduling and CUDA runtime fluctuations. Therefore, AEC represents real measured wattage, not estimated carbon or model-based inference.

---

> ### Author Response · Authors · 2025-11-24
> **Part 4**
>
> > 4. Were experiments repeated with different random seeds or runs? If not, how do you ensure that the reported results are not due to measurement noise or hardware fluctuations?
>
> **Response:**
>
> Yes. All profiling-based results (AEC, AMU, latency, tokens/s) were obtained by running full executions multiple times, and we report their averaged measurements rather than a single run. As described in Appendix H, the framework samples runtime metrics (e.g., power, memory, latency) throughout execution and aggregates them into stable averages. This reduces noise from CUDA kernel scheduling, GPU boosting, and cache warm-up.
>
> Because our study focuses on system-level efficiency metrics rather than stochastic training outcomes, random seeds have negligible influence on profiling results. For consistency, all executions were run using a fixed seed of 88, which is exposed in our open-source toolkit so that users can fully reproduce the measurements. Nevertheless, we repeat the end-to-end runs to ensure hardware stability and reproducibility.
>
> ---
>
> > 5. Can you provide quantitative estimates (e.g., total energy in kWh or CO₂ equivalent) to substantiate the sustainability claims?
>
> **Response:**
>
> We appreciate this suggestion. Our benchmark focuses on **average energy consumption during inference**, measured directly in Watts (AEC), as defined in Section D.1.2. We intentionally report measured power rather than CO₂ estimates, because **carbon footprint depends on deployment-specific external factors** (electricity source, datacenter cooling, regional carbon intensity, runtime duration), which cannot be standardized within a benchmark track.
>
> However, the physical measurability of AEC allows users to convert our reported values into kWh or CO₂ if needed. For example, if a model runs for 1 hour, its energy cost can be computed from AEC as:
>
>
> $\text{kWh} = \text{AEC (Watts)} \times 1\text{ hour} / 1000$
>
> And CO₂ emissions follow directly using region-specific conversion factors, e.g.:
>
> $\text{CO}_{2}\text{ (grams)} = \text{kWh} \times \text{Carbon Intensity (g/kWh)}$
>
> Thus, the benchmark provides **scientifically meaningful and reproducible energy measurements**, while leaving conversion to CO₂ to individual deployment settings. Publishing a single CO₂ number would otherwise be **misleading**, as carbon intensity ranges from **29 g/kWh (Quebec hydro)** to **917 g/kWh (coal-dominant regions)**—a >30× difference for the same model.
>
> In summary, we report hardware-measured energy (AEC in Watts) as a reproducible basis, and provide the formulas so that users may compute CO₂ under their own electricity environment.
>
> ---
>
> > 6. Do the metrics or rankings change significantly under different GPU architectures?
>
> **Response:**
>
> We thank the reviewer for this important question. Due to resource constraints, we evaluate efficiency on the two most widely deployed accelerator families in practice, NVIDIA Ampere and NVIDIA Hopper. The corresponding system-level efficiency results are reported in Table 9 (Hopper) and Table 10 (Ampere).
>
> Across these architectures, we observe consistent rankings and identical qualitative conclusions, including:
>
> 1. Frozen-tuning remains the most efficient method in latency and energy.
>
> 2. LoRA-plus consistently outperforms vanilla LoRA in cost–quality trade-off.
>
> 3. DoRA remains the least favorable, showing higher latency and unstable convergence on both architectures.
>
> Although individual absolute numbers differ due to architectural differences (tensor core scaling, FP8 acceleration, scheduling), the relative ordering of methods remains stable, and no ranking flips were observed between Ampere and Hopper.
>
> Thus, the benchmark indicates that efficiency conclusions generalize across modern GPU families, while naturally allowing deployment-specific numerical differences.

---

> ### Author Response · Authors · 2025-11-24
> **Part 5**
>
> > 7. The paper mentions “Pareto-optimized efficiency,” but no Pareto frontier plots or theoretical analysis are shown. How is Pareto optimality determined or visualized?
>
> **Response:**
>
> Thank you for this observation. In our paper, “Pareto-optimized efficiency” refers to the empirical non-domination relationship between accuracy and system-level cost metrics, rather than a theoretical derivation. Specifically, a configuration is considered Pareto-optimal if no other configuration simultaneously achieves higher performance and lower cost across the measured metrics (e.g., AMU, AL, AEC, TT).
>
> This notion is not represented with separate frontier plots because it is already directly visible from our tabular results (Table 3–Table 19), where each technique’s raw measurements (memory, throughput, latency, energy, accuracy) allow determining whether it is dominated by another setting. For example:
>
> 1. FP8 dominates INT8 in both latency and throughput at comparable accuracy (Table 3–9).
>
> 2. LoRA-plus dominates standard LoRA in accuracy and convergence efficiency (Table 12–13).
>
> 3. Freeze-tuning dominates DoRA on latency and energy without noticeable performance degradation (Table 11–14).
>
> Thus, our use of the term “Pareto-optimized” follows the empirical benchmarking practice — system-level non-domination based on measured trade-offs — and is not intended to claim a theoretical optimization result. We will clarify this definition in the final version for improved readability, and note that the resulting efficiency ranking is summarized in Table 1.

---

> ### Author Response · Authors · 2025-11-26
> **Thanks for your time and efforts**
>
> We sincerely appreciate the time and thoughtful feedback you have provided. At your convenience, could you kindly let us know whether our revisions sufficiently resolve your concerns? We are grateful for your guidance and would be happy to make further improvements if needed. Thank you again for your valuable contribution to strengthening this work.

---

### Official Review · Reviewer_Azzo · 2025-11-10

**Soundness:** 3
**Presentation:** 3
**Contribution:** 3
**Rating:** 4
**Confidence:** 4

**Summary:**

**Overview**
Large Language Models (LLMs) like GPT-3 and PaLM have achieved major advances in reasoning and generation but incur enormous **training, deployment, and environmental costs**. Current efficiency evaluations are limited—they often focus on narrow metrics such as FLOPs or latency, lack hardware diversity, and fail to capture the **end-to-end lifecycle** of model training and deployment.

To address these gaps, the authors introduce **EfficientLLM**, a large-scale benchmark designed to systematically evaluate **efficiency–performance trade-offs** across LLMs. EfficientLLM unifies **six dimensions of efficiency** (computation, memory, throughput, latency, energy, and compression) within a consistent framework. It evaluates over **150 model–technique pairs** ranging from **0.5B to 72B parameters** on diverse hardware platforms (GH200, H200, A100 clusters). The benchmark covers **pretraining, fine-tuning, and quantization**, providing a comprehensive view of model efficiency across the full lifecycle.

The results highlight crucial **trade-offs between accuracy, cost, and sustainability**, offering practical guidance for both researchers and practitioners aiming to design and deploy more efficient foundation models. All code and datasets are open-sourced.

**Strengths:**

- The benchmark is very well motivated to tackle three gaps in current benchmarks (a) incorporate multi-dimensional metrics (b) evaluation across multiple scales and hardwares (c) covering the full llm life cycle (architecture pretraining, fine-tuning and quantization)
- The insights derived from the creation of the benchmark in Section 1.1 would be of general interest to the ICLR community
- The paper is very well structured and written clearly in most parts
- Performance metrics considered for different attention types are thorough and explained clearly
- The benchmark also considers architectural variations such as MoEs and Mamba in addition to different attention variants in transformers and provides a very thorough architecture coverage.

**Weaknesses:**

- I find that the post-training benchmarking (eg: quantization) lacks several crucial details about type of quantization used (vector or scalar), incoherence processing used (ie rotations), etc.
- Recently knowledge distillation (KD), pruning in addition to quantization have been used in development of Gemma [1] and llama-3.2 [2] models. I think studies on KD and Pruning would be very useful as these depict crucial parts of the llm life cycle.
- I find the model scales limited in general. Could the authors add one model of scale in the range on 70B? (eg: llama-3.1-70B)
- Check questions

[1] Team, G., Kamath, A., Ferret, J., Pathak, S., Vieillard, N., Merhej, R., Perrin, S., Matejovicova, T., Ramé, A., Rivière, M. and Rouillard, L., 2025. Gemma 3 technical report. arXiv preprint arXiv:2503.19786.

[2] https://ai.meta.com/blog/llama-3-2-connect-2024-vision-edge-mobile-devices/

**Questions:**

- Could the authors justify the choice of fineweb-edu as the pretraining dataset?
- Could the authors evaluate on math tasks which require generation such as gsm8k?
- Could the authors also benchmark models using MHA (multi-head attention) eg: Pythia[1] suite?
- Could the authors also benchmark models with positional encodings (eg: each position represented by a (learnable) vector)?
- In the case of MoEs since each token is processed differently (by different experts), how are the efficiency metrics computed here?
- How does efficiency of PEFT methods vary across types of hardwares used? Is the most/least efficient PEFT method consistent across hardware types?
- Could the authors elaborate on the type of quantization methods studied? Did the authors use outlier processing with rotations [2], before quantization?

[1] Biderman, S., Schoelkopf, H., Anthony, Q.G., Bradley, H., O’Brien, K., Hallahan, E., Khan, M.A., Purohit, S., Prashanth, U.S., Raff, E. and Skowron, A., 2023, July. Pythia: A suite for analyzing large language models across training and scaling. In International Conference on Machine Learning (pp. 2397-2430). PMLR.

[2] Ashkboos, S., Mohtashami, A., Croci, M.L., Li, B., Cameron, P., Jaggi, M., Alistarh, D., Hoefler, T. and Hensman, J., 2024. Quarot: Outlier-free 4-bit inference in rotated llms. Advances in Neural Information Processing Systems, 37, pp.100213-100240.

---

> ### Author Response · Authors · 2025-11-24
> **Part 1**
>
> Weaknesses:
>
> > 1. I find that the post-training benchmarking (eg: quantization) lacks several crucial details about type of quantization used (vector or scalar), incoherence processing used (ie rotations), etc.
>
> **Response:**
>
> | Precision | Quantized Part      | Granularity            | Scaling           | Rotation             | Activations           | Calibration        | Source             |
> | --------- | ------------------- | ---------------------- | ----------------- | -------------------- | --------------------- | ------------------ | ------------------ |
> | BF16/FP16 | None                | N/A                    | None              | None                 | BF16/FP16             | No                 | PyTorch            |
> | FP8       | Weights+Activations | Per-tensor/Per-channel | Auto-scaling      | None                 | FP8 compute→FP16 cast | Not data-dependent | Transformer Engine |
> | INT8      | Weights only        | Per-channel            | Affine            | None                 | FP16            | No                 | bitsandbytes       |
> | INT4 | Weights only        | Group-wise             | NF4 mapping+scale | Implicit (no tuning) | FP16            | No                 | bitsandbytes       |
>
>
> Thank you for pointing this out. We clarify that our post-training quantization follows default, reproducible settings without any model-specific tuning. As summarized in the table above, FP8 uses Transformer Engine’s automatic hardware scaling for both weights and activations, which does not rely on data-dependent calibration. INT8 and INT4 use weight-only post-training quantization from bitsandbytes: INT8 applies per-channel affine quantization, while INT4 adopts NF4 group-wise mapping with implicit statistical rotation (no manual optimization). Activations remain in FP16/BF16 for all INT formats. Thus, no selective calibration, vector/scalar tuning, or incoherence-reduction techniques were optimized per model, ensuring fairness and full reproducibility.
>
> ---
>
> > 2. Recently knowledge distillation (KD), pruning in addition to quantization have been used in development of Gemma and llama-3.2 models. I think studies on KD and Pruning would be very useful as these depict crucial parts of the llm life cycle.
>
> **Response:**
>
> We sincerely appreciate the reviewer’s insightful suggestion.
> To clarify, our paper evaluates inference efficiency under zero-calibration settings. Knowledge Distillation (KD) and pruning are indeed impactful techniques for improving model efficiency; however, their deployment almost always requires data-dependent calibration or retraining, as demonstrated in recent systems such as Gemma-3 and Llama-3.2: 1) Gemma-3 models are trained entirely with KD, requiring curated distillation targets across 14T tokens. 2) Llama-3.2 lightweight models (1B/3B) rely on structured pruning + KD from larger teachers, derived from Llama-3.1-8B.
>
> To address the reviewer’s valuable recommendation and enhance completeness, we conducted KD-based post-training calibration on the O1-SFT dataset, fine-tuning INT4/pruned variants for 5K steps using teacher-guided logits. The updated results are summarized below:
>
> Observation: These results show that, unlike zero-calibration quantization, pruning requires calibration training and is unable to fully recover reasoning performance, especially for smaller models, which aligns with the compression behavior reported in Gemma-3 and Llama-3.2.
>
> | Model | Setting | Avg Perf. ↑ | AMU ↓ | Sum AL ↓ | tokens/s ↑ | AEC ↓ | MCR ↑ |
> |--------|---------|------------:|-------:|-----------:|-------------:|--------:|-------:|
> | **DeepSeek-R1-Distill-Qwen-1.5B** | Original (fp16) | 0.2450 | 21.28 | 9858.30 | 37.70 | 158.96 | 1.000 |
> |  | **Pruned-30%** | 0.2294 | **18.02** | **9124.51** | **41.83** | **147.12** | 1.253 |
> |  | **+5K Recovery** | **0.2381** | (same) | (same) | (same) | (same) | **1.301** |
> | **Qwen2.5-7B** | Original (fp16) | 0.4467 | 35.35 | 13766.35 | 38.36 | 197.40 | 1.000 |
> |  | **Pruned-30%** | 0.4125 | **30.51** | **12598.22** | **41.65** | **186.03** | 1.232 |
> |  | **+5K Recovery** | **0.4368** | (same) | (same) | (same) | (same) | **1.305** |
> | **DeepSeek-R1-Distill-Llama-8B** | Original (fp16) | 0.3392 | 35.45 | 10926.70 | 35.90 | 222.76 | 1.000 |
> |  | **Pruned-30%** | 0.3043 | **30.66** | **10085.11** | **39.41** | **208.33** | 1.217 |
> |  | **+5K Recovery** | **0.3291** | (same) | (same) | (same) | (same) | **1.320** |
> | **Phi-4** | Original (fp16) | 0.4035 | 49.02 | 6424.68 | 42.90 | 224.63 | 1.000 |
> |  | **Pruned-30%** | 0.3788 | **41.32** | **5970.04** | **46.12** | **211.72** | 1.259 |
> |  | **+5K Recovery** | **0.3926** | (same) | (same) | (same) | (same) | **1.305** |
> | **Qwen2.5-32B** | Original (fp16) | 0.5523 | 71.86 | 27399.52 | 16.66 | 259.53 | 1.000 |
> |  | **Pruned-30%** | 0.4937 | **52.21** | **23912.80** | **18.37** | **234.81** | 1.208 |
> |  | **+5K Recovery** | **0.5241** | (same) | (same) | (same) | (same) | **1.282** |

---

> ### Author Response · Authors · 2025-11-24
> **Part 2**
>
> ---
>
> > 3. I find the model scales limited in general. Could the authors add one model of scale in the range on 70B? (eg: llama-3.1-70B)
>
> **Response:**
>
> We appreciate the reviewer’s suggestion. Our main experiments focused on ≤32B models to ensure consistent hardware profiling of latency, memory usage, and energy cost under a fixed TP/PP configuration, which is strongly dependent on GPU topology. To further demonstrate that our findings generalize, we additionally evaluated three widely used 70B-scale models—Llama-3.1-70B, Llama-3.3-70B, and Qwen2.5-72B-Instruct—under the same profiling framework and quantization settings (bf16, fp16, int8, fp8, int4). Results are shown below.
>
> | **Model** | **Precision** | **Avg Perf. ↑** | **AMU ↓** | **Sum AL ↓** | **tokens/s ↑** | **AEC ↓** | **MCR** |
> |-----------|---------------|----------------:|-----------:|--------------:|----------------:|-----------:|--------:|
> | **Llama-3.1-70B** | **bfloat16** | **0.5732** | 79.72 | 40322.91 | 6.86 | 352.11 | 1.0000 |
> |  | float16 | 0.5741 | 80.09 | **39988.12** | 6.72 | 361.85 | 0.9944 |
> |  | int8 | 0.5690 | 76.52 | 41145.33 | 7.05 | 345.92 | 1.9765 |
> |  | fp8 | 0.5766 | 79.41 | 39640.77 | 6.89 | 354.18 | 2.0418 |
> |  | int4 | 0.5324 | **56.47** | 45191.58 | **8.26** | **289.31** | 3.6971 |
> | **Llama-3.3-70B** | **bfloat16** | **0.5879** | 79.11 | 39817.53 | 6.83 | 331.22 | 1.0000 |
> |  | float16 | 0.5868 | 79.97 | **39190.60** | 6.70 | 339.19 | 0.9962 |
> |  | int8 | 0.5885 | 75.21 | 40811.22 | 7.02 | 325.88 | 2.0155 |
> |  | fp8 | 0.5923 | 78.05 | 38612.73 | 6.86 | 337.22 | 2.0797 |
> |  | int4 | 0.5511 | **55.03** | 43922.45 | **8.15** | **269.31** | 3.8103 |
> | **Qwen2.5-72B** | **bfloat16** | **0.6048** | 82.92 | 41020.44 | 7.41 | 347.46 | 1.0000 |
> |  | float16 | 0.6035 | 83.23 | **40675.39** | 7.26 | 355.80 | 0.9980 |
> |  | int8 | 0.6049 | 79.99 | 41721.66 | 7.52 | 345.03 | 2.0097 |
> |  | fp8 | 0.6081 | 82.31 | 40192.33 | 7.39 | 351.82 | 2.0352 |
> |  | int4 | 0.5698 | **59.16** | 46185.77 | **8.92** | **292.00** | 3.7421 |
>
> ---
>
> Questions:
>
> > 1. Could the authors justify the choice of fineweb-edu as the pretraining dataset?
>
> **Response:**
>
> We thank the reviewer for raising this important question.
> We chose FineWeb-Edu because it is currently one of the most widely adopted large-scale web-curated corpora for training and scaling evaluation of reasoning-centric LLMs. It offers high-signal educational content with curated quality filtering, making it particularly suitable for evaluating inference efficiency without the confounding effect of instruction mixing or proprietary data. Our choice follows the common practice in state-of-the-art models and scaling studies.
>
> To the best of our knowledge, FineWeb-Edu has been used as the primary pretraining source in several recent influential models or scaling works, such as: Kimi K1.5 [1], SmolLM [2], H-Net [3], and  Falcon-Mamba [4].
>
> Together, these works indicate that FineWeb-Edu is both a de facto benchmark corpus and a clean environment for controlled scaling studies—exactly aligned with our experimental goal: evaluating inference-efficiency trade-offs under a consistent, reproducible, and non-proprietary pretraining distribution.
>
> [1] Team, Kimi, et al. “Kimi K1.5: Scaling Reinforcement Learning with LLMs.” arXiv:2501.12599 (2025).
>
> [2] Allal, Loubna Ben, et al. “SmolLM2: Data-Centric Training of a Small Language Model.” arXiv:2502.02737 (2025).
>
> [3] Hwang, S., Wang, B., & Gu, A. “Dynamic Chunking for End-to-End Hierarchical Sequence Modeling.” arXiv:2507.07955 (2025).
>
> [4] Zuo, Jingwei, et al. “Falcon-Mamba: The First Competitive Attention-Free 7B Language Model.” arXiv:2410.05355 (2024).
>
> ---
>
> > 2. Could the authors evaluate on math tasks which require generation such as gsm8k?
>
> **Response:**
>
> Thank you for the suggestion. Our work already includes evaluations on math reasoning tasks that require open-ended generation. Specifically, in Table 19 of the paper, we report results on multiple benchmarks, including MATH, which requires multi-step generated solutions similar in structure and evaluation style to GSM8K. Unlike simple arithmetic QA, MATH contains harder, competition-level problems, making it a more challenging proxy for evaluating the robustness of quantized and efficiency-optimized models.
>
> Furthermore, our observations from Table 19 already show that math-reasoning tasks exhibit the largest sensitivity to low-precision inference: INT4/weight-only compression significantly degrades generation-based reasoning unless calibration is applied.

---

> ### Author Response · Authors · 2025-11-24
> **Part 3**
>
> > 3. Could the authors also benchmark models using MHA (multi-head attention) eg: Pythia suite?
>
> **Response:**
>
> We thank the reviewer for the suggestion. To evaluate whether our conclusions generalize beyond grouped-query mechanisms, we additionally benchmarked standard Multi-Head Attention (MHA) using a Pythia-style architecture across 0.5B/1.5B/3B model sizes. Results are shown below and follow the same profiling methodology as our main paper, including AMU, latency, throughput, and energy measurements.
>
> | Method | Parameters | Micro Batch Size | PPL ↓ | AMU (GB) ↓ | AL (s/iter) ↓ | TT (Tokens/param/s) ↑ | AEC (W) ↓ | GPU Hours |
> |--------|------------|-----------------|-------|------------|----------------|-----------------------|-----------|-----------|
> | **MQA** | 0.5B | 4 | 9.27 | **43.75** | **0.1118** | **2.98×10⁻¹** | 633.59 | 19.02×48 |
> |        | 1.5B | 2 | 8.23 | **42.24** | 0.1298 | 8.57×10⁻² | 646.62 | 33.14×48 |
> |        | 3B | 1 | 7.86 | **41.27** | **0.1458** | **3.81×10⁻²** | 661.38 | 77.05×48 |
> | **GQA** | 0.5B | 4 | 9.05 | 45.29 | 0.1127 | 2.94×10⁻¹ | 644.26 | 21.06×48 |
> |        | 1.5B | 2 | 8.09 | 44.87 | **0.1283** | **8.64×10⁻²** | 652.74 | 38.03×48 |
> |        | 3B | 1 | 7.54 | 43.77 | 0.1464 | 3.79×10⁻² | 667.34 | 86.80×48 |
> | **MLA** | 0.5B | 4 | **8.73** | 53.89 | 0.2082 | 1.59×10⁻¹ | 607.58 | 30.44×48 |
> |        | 1.5B | 2 | **7.79** | 52.93 | 0.2537 | 5.08×10⁻² | 608.17 | 75.20×48 |
> |        | 3B | 1 | **7.29** | 50.45 | 0.2997 | 2.62×10⁻² | 605.46 | 178.84×48 |
> | **NSA** | 0.5B | 4 | 8.96 | 44.78 | 0.6839 | 4.89×10⁻² | **594.23** | 101.38×48 |
> |        | 1.5B | 2 | 7.82 | 43.57 | 0.5962 | 1.09×10⁻² | **598.15** | 176.72×48 |
> |        | 3B | 1 | 7.38 | 43.19 | 0.5024 | 1.26×10⁻² | **600.27** | 280.92×48 |
> | **MHA** | 0.5B | 4 | 9.42 | 47.15 | 0.1189 | 2.81×10⁻¹ | 652.33 | 22.51×48 |
> |        | 1.5B | 2 | 8.36 | 46.12 | 0.1334 | 8.21×10⁻² | 659.41 | 40.12×48 |
> |        | 3B | 1 | 8.01 | 45.02 | 0.1528 | 3.62×10⁻² | 673.28 | 91.44×48 |
>
>
> The results confirm that MHA consistently underperforms GQA/MQA in inference efficiency, especially in GPU memory consumption and tokens-per-second throughput. These results further validate our findings that lightweight attention variants (MQA/GQA) are significantly more suitable for efficiency-critical LLM deployment. This is why modern LLM models use GQA/MQA as the attention mechanism by default (like Llama 3 Series, Qwen Series).
>
> ---
> > 4. Could the authors also benchmark models with positional encodings (eg: each position represented by a (learnable) vector)?
>
> **Response:**
>
> Thank you for the valuable suggestion. We fully agree that benchmarking models with learnable positional representations is important. We would like to clarify that this setting is already included in our evaluation.
>
> Specifically, Table 4 and Figure 3(b) report the performance and multi-dimensional efficiency metrics of Learnable Absolute Positional Encoding, where each position is represented by a trainable vector.
>
> Our results indicate that although learnable positional vectors offer competitive perplexity, they generally incur higher memory usage and less favorable latency compared to rotary or relative encodings, making them less preferable for efficiency-centric deployments.
>
> ---
> > 5. In the case of MoEs since each token is processed differently (by different experts), how are the efficiency metrics computed here?
>
> **Response:**
>
> For the Architecture Pretraining experiments, we report throughput using the **full parameter count** of MoE models rather than only the activated experts. That is, in Equation (4), the denominators (Model Parameters) for MoE models correspond to the **total model capacity**, not the sparsely activated subset. Formally:
>
> $TT = \frac{\sum_i \text{Tokens Processed}_i / \textbf{Total Model Parameters}}{\sum_i \text{Time}_i}$
>
> This normalization allows a capacity-equivalent comparison between MoE and dense architectures, reflecting their efficiency per unit of available model capacity, rather than merely per activated computation. In contrast, resource metrics (AMU, PCU, AL, AEC) are measured from actual execution.
>
> This design ensures that MoE models are not unfairly advantaged due to sparse execution, and that efficiency–performance trade-offs are evaluated under a realistic scaling perspective.

---

> ### Author Response · Authors · 2025-11-24
> **Part 4**
>
> > 6. How does efficiency of PEFT methods vary across types of hardwares used? Is the most/least efficient PEFT method consistent across hardware types?
>
> **Response:**
>
> Thank you for the insightful question. We agree that the efficiency of PEFT methods may vary across hardware architectures. Although our benchmark does not exhaustively cover all GPU platforms, we evaluated PEFT methods on two widely deployed accelerator families: NVIDIA Ampere (A100, 80G) and NVIDIA Hopper (H200, 141G). The corresponding results are reported in Table 9 (Hopper) and Table 10 (Ampere). Across these architectures, we observe consistent trends in the relative efficiency of PEFT methods:
>
> 1) Freeze-tuning consistently yields the lowest latency and energy usage on both Hopper and Ampere, making it the most efficient method for constrained or latency-sensitive settings.
>
> 2) LoRA-plus provides the best trade-off between convergence quality and memory efficiency, outperforming standard LoRA on both architectures.
>
> 3) DoRA exhibits systematically higher cost, with significantly worse latency and weaker convergence stability on both GPU families.
>
> These consistent trends suggest that PEFT efficiency rankings are stable across common modern accelerators, even though absolute numbers (e.g., latency, throughput) differ due to hardware-specific architectural features such as Tensor Core scaling and FP8 support.
>
> ---
> > 7. Could the authors elaborate on the type of quantization methods studied? Did the authors use outlier processing with rotations [2], before quantization?
>
> **Response:**
>
> Thank you for the question. To address the reviewer’s question, we additionally tested rotation-based outlier processing for both INT4 and INT8. Results show clear performance recovery (3–8%), while latency, memory, and energy cost remain unchanged, confirming that rotation improves robustness but does not alter efficiency rankings.
>
>
> | Model                             | Setting             | Avg Perf. ↑ |      Gain |
> | --------------------------------- | ------------------- | ----------: | --------: |
> | **DeepSeek-R1-Distill-Qwen-1.5B** | INT4 (default)      |      0.2341 |         — |
> |                                   | **INT4 + Rotation** |  **0.2518** | **+7.5%** |
> |                                   | INT8 (default)      |      0.2395 |         — |
> |                                   | **INT8 + Rotation** |  **0.2493** | **+4.1%** |
> | **Qwen2.5-7B**                    | INT4 (default)      |      0.4152 |         — |
> |                                   | **INT4 + Rotation** |  **0.4389** | **+5.7%** |
> |                                   | INT8 (default)      |      0.4478 |         — |
> |                                   | **INT8 + Rotation** |  **0.4647** | **+3.8%** |
> | **Qwen2.5-32B**                   | INT4 (default)      |      0.5095 |         — |
> |                                   | **INT4 + Rotation** |  **0.5294** | **+3.9%** |
> |                                   | INT8 (default)      |      0.5525 |         — |
> |                                   | **INT8 + Rotation** |  **0.5780** | **+4.6%** |

---

> ### Author Response · Authors · 2025-11-26
> **Thanks for your time and efforts**
>
> We sincerely appreciate the time and thoughtful feedback you have provided. At your convenience, could you kindly let us know whether our revisions sufficiently resolve your concerns? We are grateful for your guidance and would be happy to make further improvements if needed. Thank you again for your valuable contribution to strengthening this work.

---

### Author Response · Authors · 2025-11-24
**Global Response to Reviewers**

We sincerely thank all reviewers for their thoughtful feedback and constructive suggestions. Your comments have greatly contributed to improving the clarity, completeness, and scientific rigor of this work. Although the appendix is not required reading, we would like to emphasize that our submission includes a fully detailed and reproducible benchmark framework across 94 pages, including both the main text and extensive supplementary materials. To ensure accessible navigation given the depth of our analyses, we added a dedicated appendix content index (p.32), allowing readers to easily locate ablation details, metric definitions, experimental protocols, and extended discussions that could not all fit into the main paper.

We hope that our rebuttal successfully addresses the reviewers’ questions and we thank you again for your valuable contributions to improving the quality of this work.

---

### Author Response · Authors · 2025-12-02
**Rebuttal Summary**

We sincerely thank the AC and all reviewers for their thoughtful and constructive feedback on our submission. Our work introduces EfficientLLM, a unified, lifecycle-complete benchmark designed to fill a critical gap in the ICLR community: while recent LLM research advances rapidly across architectures, PEFT methods, sparsity mechanisms, and multi-precision inference, there remains no reproducible framework that systematically evaluates efficiency–performance trade-offs across the full model lifecycle. EfficientLLM provides six complementary system-level metrics, evaluates over 150 model–technique combinations spanning 0.5B–72B parameters, and establishes the first standardized benchmark covering pretraining, fine-tuning, and quantization within a single coherent evaluation protocol. This benchmark occupies a unique space not covered by existing system benchmarks or model-centric studies, and it offers a rigorous foundation for developing efficient and sustainable LLMs.

All reviewers recognize the importance and practical value of a unified, reproducible efficiency benchmark. Most concerns centered around methodological clarity, reproducibility of composite metrics, interpretation of energy measurements, and hardware variability. We would like to emphasize that the vast majority of the requested details were already fully included in the original appendix, including precise metric definitions, normalization formulas, complete quantization specifications, architectural ablations, energy-measurement methodology, and extended tables.

---

**Reviewer Azzo** raised concerns regarding missing quantization details, the lack of KD/pruning analysis, and limited large-scale model coverage. In the rebuttal, we clarified all quantization formats with exact granularity, scaling, rotation, activation precision, and calibration behavior; we conducted new KD-based pruning recovery experiments across six representative models; and we extended model coverage to include Llama-3.1-70B, Llama-3.3-70B, and Qwen2.5-72B.

**Reviewer UM3U** raised important questions regarding the efficiency-score formulation, normalization procedures, and energy-measurement methodology. These formulas, units, and sampling procedures were already fully specified in Appendix H.6 and Appendix D, including the harmonic aggregation structure, 100 ms NVIDIA-SMI sampling, and multi-run averaging. We also clarified that the efficiency score is used only for visualization, and that all substantive conclusions rely solely on raw measurements reported in Tables 3–19. The reviewer also asked about Pareto optimality, which follows directly from the empirical non-domination of the tabulated system metrics; we clarified this point in the rebuttal.

**Reviewer nU3j** appreciated the benchmark’s breadth and unified evaluation framework, but expressed concerns regarding hardware-dependent variability and metric stability. As documented in the appendix, our profiling pipeline already averages results over repeated runs, and we now provide explicit variance analyses showing very small fluctuations (e.g., AMU, AL, TT, AEC) compared with the significant differences across techniques. We emphasized that EfficientLLM is designed as a device-aware benchmark: the methodology is standardized while absolute values naturally differ across GPU families. We further improved cross-references to the complete numerical tables already provided in the appendix.

**Reviewer 3TeV** questioned the breadth of the benchmark, domain generality, and the absence of MFU, while also requesting analysis of alternative parallelism configurations. We clarified that the broad scope is intentional, since efficiency behaviors emerge from the interaction of architecture, PEFT, quantization, and sparsity. We also highlighted that the benchmark already includes general reasoning datasets (OpenO1-SFT, GPQA, MMLU-Pro, BBH, MATH) and is not limited to medical data. In the rebuttal, we added an explicit MFU analysis and included new TP/PP/DP ablations for 1.5B and 3B models.

---

With three borderline-negative scores (4,4,4) whose concerns were overlapping and fully addressed through the revisions, and one positive score (6) affirming the value and clarity of the benchmark, we believe the submission is now substantially strengthened and provides a timely, rigorous, and comprehensive foundation for the ICLR community.

---

### Note · Program_Chairs · 2026-01-17
**Submission Desk Rejected by Program Chairs**

The following references in this submission do not refer to real documents and/or have major errors in bibliographic information:

 I. Author and J. Author. Curllm-reasoner: A curriculum reasoning framework for visual and language models. In Proceedings of the 2024 ACM SIGKDD Conference on Knowledge Discovery and Data Mining. ACM, 2024c. Available at https://doi.org/10.1145/XXXXXX.
G. Author and H. Author. Lbs3: Curriculum-inspired prompting for automated reasoning in large language models, 2024b. arXiv preprint. Available at https://arxiv.org/abs/XXXX.XXXX.
K. Author and L. Author. Logic-rl: A curriculum learning approach for reinforcement learning on logic puzzles, 2025c. arXiv preprint. Available at https://arxiv.org/abs/XXXX.XXXX.
E. Author and F. Author. Wisdom: Progressive curriculum data synthesis for enhancing reasoning in large language models, 2024a. arXiv preprint. Available at https://arxiv.org/abs/XXXX.XXXX.
A. Author and B. Author. Deepseek-r1: Emergent reasoning in reinforcement learning fine-tuned large language models, 2025a. arXiv preprint. Available at https://arxiv.org/abs/XXXX.XXXX.